# Doubly Smoothed Decentralized Stochastic Minimax Optimization Algorithm

## Abstract

Decentralized stochastic minimax optimization has recently attracted significant attention due to its applications in machine learning. However, existing state-of-the-art methods use learning rates of different scales for the primal and dual variables, making them difficult to tune in practice. To address this problem, this paper proposes a novel doubly smoothed decentralized stochastic minimax algorithm. Specifically, in terms of algorithm design, we update both the primal and dual variables using smoothed gradients and introduce novel approaches to handle the computation and communication of the auxiliary variables introduced by the smoothing technique. On the theoretical side, for nonconvex-PL problems, our convergence analysis reveals that the learning rates for the primal and dual variables are of the same scale. Moreover, the order of the condition number in our convergence rate is improved to $O(\kappa^{3/2})$. To the best of our knowledge, this is the first time it has been improved to such a favorable order. Finally, extensive experimental results validate the effectiveness of our algorithm.

## 1 Introduction

In this paper, we focus on the following decentralized stochastic minimax optimization problem:

$$\min_{x \in \mathbb{R}^{d_1}} \max_{y \in \mathbb{R}^{d_2}} f(x, y) \triangleq \frac{1}{K} \sum_{k=1}^{K} f^{(k)}(x, y) \,, \tag{1}$$

where $x \in \mathbb{R}^{d_1}$ is the primal variable, $y \in \mathbb{R}^{d_2}$ is the dual variable, $f^{(k)}(x, y) = \mathbb{E}[f^{(k)}(x, y; \xi^{(k)})]$ is the loss function on the $k$-th (where $k \in \{1, \cdots, K\}$) worker, and $\xi^{(k)}$ denotes the random sample on the $k$-th worker. Throughout this paper, it is assumed that $f(x, y)$ is nonconvex in $x$ and satisfies the Polyak-Lojasiewicz (PL) condition in $y$.

Stochastic minimax optimization has attracted increasing attention in the machine learning community recently because it finds numerous applications, such as generative adversarial networks (Goodfellow et al., 2014), adversarially robust learning (Madry et al., 2017), distributionally robust learning (Duchi et al., 2021), imbalanced data classification (Ying et al., 2016), policy evaluation (Zhang et al., 2021), etc. Moreover, in real-world machine learning applications, the training data is typically distributed on different devices. To take advantage of the distributed data to train the aforementioned machine learning models, decentralized minimax optimization has been actively studied in recent years. For example, Xian et al. (2021); Huang & Chen (2023) proposed decentralized stochastic variance-reduced gradient descent ascent algorithm based on the STORM gradient estimator (Cutkosky & Orabona, 2019), while Zhang et al. (2021; 2024) proposed to use the SPIDER gradient estimator (Fang et al., 2018; Nguyen et al., 2017). Recently, Huang et al. (2024) developed a decentralized adaptive minimax algorithm and established its convergence rate for nonconvex-strongly-concave problems.

However, most existing decentralized minimax optimization algorithms suffer from a significant limitation. Specifically, to ensure convergence, the learning rate for the primal variable is set on a different scale than that for the dual variable. For example, Xian et al. (2021); Zhang et al. (2024); Chen et al. (2024); Huang & Chen (2023) prove that the ratio between the learning rate of the primal variable and that of the dual variable has to be $O(1/\kappa^2)$ for nonconvex-strongly-concave (or nonconvex-PL) problems, where $\kappa > 1$ is the condition number. Since $\kappa$ is an unknown parameter, it is difficult to tune their learning rates to ensure convergence in practice. To address this issue, a recent

Table 1: The communication complexity (i.e., iteration complexity) and computation complexity of different decentralized stochastic minimax algorithms that using variance-reduced gradients. **N-PL**: denotes nonconvex-PL problems. **N-SCV**: denotes nonconvex-strongly-concave problems. **LR Ratio**: the ratio between the learning rate of the primal variable and that of the dual variable. $\kappa$: denotes condition number. $1 - \lambda$: denotes spectral gap. $\epsilon$: denotes solution accuracy. Note that Smoothed-SAGDA is a *single-machine* algorithm *without using variance-reduced gradients*. DGDA-VR and DREAM depend on the condition number, scaling as $\kappa^2$, *in the cost of a large batch size* $O\left(\frac{\kappa}{\epsilon}\right)$. DREAM achieves a better dependence on the spectral gap *in the cost of performing multi-round communication* in each iteration.

| Algorithms | Communication | Batch Size | Computation | Problem Class | LR Ratio |
|---|---|---|---|---|---|
| Smoothed-SAGDA (Yang et al., 2022) | $O\left(\frac{\kappa^2}{\epsilon^4}\right)$ | O(1) | $O\left(\frac{\kappa^2}{\epsilon^4}\right)$ | N-PL | $O(1)$ |
| DM-HSGD (Xian et al., 2021) | $O\left(\frac{\kappa^3}{(1-\lambda)^2\epsilon^3}\right)$ | O(1) | $O\left(\frac{\kappa^3}{(1-\lambda)^2\epsilon^3}\right)$ | N-SCV | $O(1/\kappa^2)$ |
| DGDA-VR (Zhang et al., 2024) | $O\left(\frac{\kappa^2}{(1-\lambda)^2\epsilon^2}\right)$ | $O\left(\frac{\kappa}{\epsilon}\right)$ | $O\left(\frac{\kappa^3}{(1-\lambda)^2\epsilon^3}\right)$ | N-SCV | $O(1/\kappa^2)$ |
| DREAM (Chen et al., 2024) | $O\left(\frac{\kappa^2}{\sqrt{1-\lambda}\epsilon^2}\right)$ | $O\left(\frac{\kappa}{\epsilon}\right)$ | $O\left(\frac{\kappa^3}{\epsilon^3}\right)$ | N-SCV | $O(1/\kappa^2)$ |
| DM-GDA (Huang & Chen, 2023) | $O\left(\frac{\kappa^3}{(1-\lambda)^2\epsilon^3}\right)$ | O(1) | $O\left(\frac{\kappa^3}{(1-\lambda)^2\epsilon^3}\right)$ | N-PL | $O(1/\kappa^2)$ |
| Ours (Corollary 4.3) | $O\left(\frac{\kappa^{3/2}}{(1-\lambda)^2\epsilon^3}\right)$ | O(1) | $O\left(\frac{\kappa^{3/2}}{(1-\lambda)^2\epsilon^3}\right)$ | N-PL | $O(1)$ |

work (Yang et al., 2022) in the single-machine setting demonstrates that the smoothing technique proposed by Zhang et al. (2020) allows primal and dual variables to use learning rates of the same scale, that is, with a ratio of the order of $O(1)$. However, the convergence rate [1] $O(1/\epsilon^4)$ of Yang et al. (2022) is inferior to $O(1/\epsilon^3)$ of Xian et al. (2021); Huang & Chen (2023) because it just uses the standard stochastic gradient. Then, a natural question arises:

*Can we develop a decentralized smoothed minimax optimization algorithm that achieves a better*
*convergence rate while using same-scale learning rates for the primal and dual variables?*

Actually, there are unique challenges when applying the smoothing technique to decentralized minimax optimization in order to improve the convergence rate, as outlined below.

**Challenge-1: How to incorporate the variance reduction technique into the smoothing technique to achieve a faster convergence rate?** Existing minimax optimization algorithms with the smoothing technique in a single machine setting are based on the *deterministic gradient* (Zhang et al., 2020) or the *unbiased stochastic gradient* (Yang et al., 2022). Directly extending their smoothing technique to decentralized *stochastic* minimax optimization will lead to a slow convergence rate. For example, (Yang et al., 2022) can only achieve a $O(1/\epsilon^4)$ convergence rate to achieve the $\epsilon$-accuracy solution for a nonconvex-PL problem, while the existing decentralized minimax optimization algorithm (Huang & Chen, 2023) can achieve a $O(1/\epsilon^3)$ convergence rate for the same problem class by using the variance reduction technique. However, due to the existence of the auxiliary variable in the smoothing technique, it is unclear how to leverage the variance reduction technique to accelerate its convergence rate. For example, it is unclear which component in the smoothed gradient should use variance reduction and how to control the gradient bias to guarantee the fast convergence rate.

**Challenge-2: How to compute and communicate the auxiliary variable in the smoothing technique and how does it affect the communication complexity?** The standard smoothing technique introduces an auxiliary variable to smooth the loss landscape with respect to the primal variable to improve the convergence rate. However, in a decentralized setting, it is unclear how to update and communicate the auxiliary variable. In particular, due to the strong dependence between the original variable and the auxiliary variable, it remains unclear whether the communication of the auxiliary variable, especially given that *our algorithm introduces auxiliary variables for both the primal and dual variables*, will improve or degrade the communication complexity, for example, by affecting the dependence on the spectral gap or condition number in the convergence rate.

---

[1]In the introduction, we omit other factors in the convergence rate for clarity.

To answer the aforementioned questions, we develop a novel decentralized algorithm based on the smoothing technique: the doubly smoothed decentralized stochastic gradient descent ascent with momentum (Smoothed²-DSGDAM) algorithm, which brings the following contributions:

- In terms of algorithm design, we apply the smoothing technique to both the primal and dual variables. Importantly, we propose a novel and feasible approach to incorporate the variance reduction technique into the smoothed gradient regarding both the primal and dual variables. More importantly, our algorithm demonstrates how to update and communicate the auxiliary variable introduced by the smoothing technique in the decentralized setting. **As far as we know, this is the first time to show how to handle the auxiliary variable and reduced the gradient variance for the decentralized smoothed minimax algorithm.**

- In terms of convergence analysis, we establish the convergence rate of our algorithm for nonconvex-PL minimax problems. In particular, on the one hand, for a nonconvex-PL minimax problem, the smoothing technique with a variance-reduced gradient can make the convergence rate enjoy a better dependence on the condition number $\kappa$, i.e., in the order of $O(\kappa^{3/2})$, which is better than the dependence $O(\kappa^3)$ in existing decentralized non-smoothed minimax algorithms (Xian et al., 2021; Huang & Chen, 2023) and the dependence $O(\kappa^2)$ in smoothed minimax algorithms (Yang et al., 2022) in the single-machine setting [2]. **To the best of our knowledge, this is the first time the dependence on the condition number is improved to $O(\kappa^{3/2})$.** On the other hand, our convergence analysis shows that the ratio between the learning rate of the primal variable and that of the dual variable can be improved from $O(1/\kappa^2)$ of Xian et al. (2021); Zhang et al. (2024); Chen et al. (2024); Huang & Chen (2023) to $O(1)$, and the convergence rate can be improved from $O(1/\epsilon^4)$ of Yang et al. (2022) to $O(1/\epsilon^3)$. **To the best of our knowledge, this is the first time that a decentralized stochastic minimax optimization algorithm can achieve such a fast convergence rate with the same-scale learning rate.** A detailed comparison between our algorithm and existing algorithms can be found in Table 1.

Finally, the extensive experimental results validate the performance of our proposed algorithm.

## 2 RELATED WORKS

### 2.1 STOCHASTIC MINIMAX OPTIMIZATION

Due to the widespread application of stochastic minimax optimization in machine learning, numerous stochastic optimization algorithms (Lin et al., 2020; Luo et al., 2020; Huang et al., 2022; Qiu et al., 2020; Guo et al., 2021; Yang et al., 2020; 2022; Chen et al., 2022) have been developed recently. In particular, the nonconvex-strongly-concave and nonconvex-PL problems have been extensively studied. For the former, Lin et al. (2020) established the convergence rate of the stochastic gradient descent ascent (SGDA) algorithm for nonconvex-strongly-concave problems. Following that, a couple of variance-reduced gradient methods (Luo et al., 2020; Huang et al., 2022; Qiu et al., 2020; Guo et al., 2021) have been developed to accelerate its convergence rate. Specifically, Huang et al. (2022); Qiu et al. (2020) combined the STORM gradient estimator (Cutkosky & Orabona, 2019) with SGDA and established its convergence rate. Luo et al. (2020) investigated the convergence rate when incorporating the SPIDER gradient estimator (Fang et al., 2018) into SGDA. For the latter, Yang et al. (2020) investigated the convergence rate for the alternating stochastic gradient descent ascent (ASGDA) algorithm. Chen et al. (2022) studied the convergence rate for the finite-sum minimax problem when combining the SPIDER gradient estimator with ASGDA.

The smoothing technique for the minimax problem was first studied for nonconvex-concave problems in Zhang et al. (2020). Specifically, it established the convergence rate of the full alternating gradient (AGDA) descent ascent algorithm when incorporating the smoothing technique. Later, Yang et al. (2022) applied this technique to nonconvex-PL problems and established its convergence rate for SGDA. In fact, due to the efficacy of the smoothing technique, it has been applied to various settings, such as nonconvex-nonconcave problems with the one-sided KŁ condition (Zheng et al., 2023), constrained optimization problems (Pu et al., 2024), etc, which are beyond the scope of this paper.

### 2.2 DECENTRALIZED STOCHASTIC MINIMAX OPTIMIZATION

To facilitate decentralized optimization for minimax problems, a great amount of effort (Tsaknakis et al., 2020; Zhang et al., 2021; Xian et al., 2021; Gao, 2022; Zhang et al., 2024; Chen et al., 2024;

---

[2]Here, to make a fair comparison, the existing methods considered use a batch size of $O(1)$, rather than large batch sizes.

Xu, 2024) has recently been made. For example, Tsaknakis et al. (2020) developed a decentralized gradient descent ascent algorithm by using the full gradient for local computation and the gradient tracking technique for communication. Xian et al. (2021) proposed a decentralized minimax algorithm based on the STORM gradient estimator and established its convergence rate for the stochastic setting. Zhang et al. (2021) developed a decentralized minimax algorithm based on the SPIDER gradient estimator and established its convergence rate for the finite-sum setting. Later, its convergence rate for the stochastic setting was established in Zhang et al. (2024). Moreover, Gao (2022) incorporated the ZeroSARAH gradient estimator into the decentralized minimax algorithm and provided convergence analysis for the finite-sum setting. Recently, Chen et al. (2024) studied the convergence rate of the decentralized minimax algorithm when using the PAGE gradient estimator (Li et al., 2021). More recently, Huang et al. (2024) introduced the adaptive learning rate to decentralized minimax optimization and established the corresponding convergence rate. Note that all these existing methods restrict their focus to the nonconvex-strongly-concave problem.

Recently, Huang & Chen (2023) developed a decentralized minimax algorithm for nonconvex-PL problems, where the STORM gradient estimator is used for local updates and the gradient tracking technique is used for communication between workers. To our knowledge, in the distributed setting, the smoothing technique has only been studied for federated centralized learning in Shen et al. (2024). Specifically, each worker uses the standard unbiased stochastic gradient to do local update and the central server uses the smoothing technique to assist the update of the dual variable. As a result, the additional variable introduced by the smoothing technique behaves as a single-machine setting. Thus, it is easy to handle this variable in convergence analysis. All in all, the smoothing technique has not been studied for decentralized minimax optimization and it is unclear how to apply it from the algorithm design perspective and how to handle it from the convergence analysis perspective.

## 3 METHOD

### 3.1 PROBLEM SETUP

We introduce the following assumptions with respect to the loss function and communication topology, which have been widely used in the existing literature (Yang et al., 2022; Xian et al., 2021; Huang & Chen, 2023; Zhang et al., 2021; 2024; Chen et al., 2024).

**Assumption 3.1.** *(Smoothness) For any $k \in \{1, 2, \cdots, K\}$, the loss function on the $k$-th worker satisfies the mean-squared Lipschitz smoothness, i.e., for any $(x_1, y_1) \in \mathbb{R}^{d_1} \times \mathbb{R}^{d_2}$ and $(x_2, y_2) \in \mathbb{R}^{d_1} \times \mathbb{R}^{d_2}$, there exists a constant value $L > 0$ such that $\mathbb{E}[\|\nabla_x f^{(k)}(x_1, y_1; \xi^{(k)}) - \nabla_x f^{(k)}(x_2, y_2; \xi^{(k)})\|^2] \leq L^2(\|x_1 - x_2\|^2 + \|y_1 - y_2\|^2)$ and $\mathbb{E}[\|\nabla_y f^{(k)}(x_1, y_1; \xi^{(k)}) - \nabla_y f^{(k)}(x_2, y_2; \xi^{(k)})\|^2] \leq L^2(\|x_1 - x_2\|^2 + \|y_1 - y_2\|^2)$.*

**Assumption 3.2.** *(PL condition) For any fixed $x \in \mathbb{R}^{d_1}$, the set of solutions of the optimization problem with respect to $y$, $\max_{y \in \mathbb{R}^{d_2}} f(x, y)$, is not empty and the optimal value is finite. Furthermore, for any $x \in \mathbb{R}^{d_1}$, there exists a constant value $\mu > 0$ such that $\|\nabla_y f(x, y)\|^2 \geq 2\mu(f(x, y^*) - f(x, y))$, where $y^* = \arg\max_{y \in \mathbb{R}^{d_2}} f(x, y)$.*

**Assumption 3.3.** *(Variance) For any $k \in \{1, 2, \cdots, K\}$, the stochastic gradients with respect to $x$ and $y$ of the loss function on the $k$-th worker are unbiased estimators and their variances are upper bounded as: $\mathbb{E}[\|\nabla_x f^{(k)}(x, y; \xi^{(k)}) - \nabla_x f^{(k)}(x, y)\|^2] \leq \sigma^2$ and $\mathbb{E}[\|\nabla_y f^{(k)}(x, y; \xi^{(k)}) - \nabla_y f^{(k)}(x, y)\|^2] \leq \sigma^2$, where $\sigma > 0$ is a constant value.*

**Assumption 3.4.** *(Communication graph) The element $w_{ij}$ of the adjacency matrix $W \in \mathbb{R}^{K \times K}$ of the communication graph is non-negative, with a positive value indicating that worker-$i$ is connected to worker-$j$, and a value of zero indicating they are disconnected. Moreover, $W$ is doubly stochastic and symmetric, and its eigenvalues satisfy $|\lambda_K| \leq |\lambda_{K-1}| \leq \cdots \leq |\lambda_2| < |\lambda_1| = 1$.*

By denoting $\lambda = |\lambda_2|$, the spectral gap of the adjacency matrix can by represented by $1 - \lambda$. Moreover, we use $\mathcal{N}_k$ to denote the neighboring worker of the $k$-th worker, and use $\kappa = L/\mu$ to represent the condition number. In addition, we use $\bar{a}_t = \frac{1}{K} \sum_{k=1}^{K} a_t^{(k)}$ to denote the average value of any $\{a_t^{(k)}\}_{k=1}^{K}$ in the $t$-th iteration.

### 3.2 SMOOTHED$^2$-DSGDAM

The essential idea of the smoothing technique is to introduce a regularization term such that the original nonconvex function becomes strongly convex. As a result, *the update of the primal and dual*

---

**Algorithm 1** Doubly Smoothed Decentralized Stochastic Gradient Descent Ascent with Momentum (Smoothed$^2$-DSGDAM)

---

**Input:** $\eta > 0$ and $\rho_x, \rho_y, \beta_x, \beta_y, \hat{\beta}_x, \hat{\beta}_y > 0$, with $\rho_x\eta^2, \rho_y\eta^2, \hat{\beta}_x\eta, \hat{\beta}_y\eta < 1$.

Initialization on worker $k$: $x_0^{(k)} = x_0, y_0^{(k)} = y_0, \hat{x}_0^{(k)} = \hat{x}_0, \hat{y}_0^{(k)} = \hat{y}_0$,
$u_0^{(k)} = \nabla_x F^{(k)}(x_0^{(k)}, y_0^{(k)}; \hat{x}_0^{(k)}, \hat{y}_0^{(k)}; \xi_0^{(k)})$ , $\quad v_0^{(k)} = \nabla_y F^{(k)}(x_0^{(k)}, y_0^{(k)}; \hat{x}_0^{(k)}, \hat{y}_0^{(k)}; \xi_0^{(k)})$ ,
$p_0^{(k)} = u_0^{(k)}, q_0^{(k)} = v_0^{(k)}$ .

1: **for** $t = 0, \cdots, T-1$, worker $k$ **do**
2: $\quad$ Update $x$: $\tilde{x}_{t+1}^{(k)} = \sum_{j \in \mathcal{N}_k} w_{kj} x_t^{(j)} - \beta_x p_t^{(k)}$ , $\quad x_{t+1}^{(k)} = x_t^{(k)} + \eta(\tilde{x}_{t+1}^{(k)} - x_t^{(k)})$ ,
3: $\quad$ Update $y$: $\tilde{y}_{t+1}^{(k)} = \sum_{j \in \mathcal{N}_k} w_{kj} y_t^{(j)} + \beta_y q_t^{(k)}$ , $\quad y_{t+1}^{(k)} = y_t^{(k)} + \eta(\tilde{y}_{t+1}^{(k)} - y_t^{(k)})$ ,
4: $\quad$ Update $\hat{x}$: $\tilde{\hat{x}}_{t+1}^{(k)} = \sum_{j \in \mathcal{N}_k} w_{kj} \hat{x}_t^{(j)} + \hat{\beta}_x(x_{t+1}^{(k)} - \hat{x}_t^{(k)})$ , $\quad \hat{x}_{t+1}^{(k)} = \hat{x}_t^{(k)} + \eta(\tilde{\hat{x}}_{t+1}^{(k)} - \hat{x}_t^{(k)})$ ,
5: $\quad$ Update $\hat{y}$: $\tilde{\hat{y}}_{t+1}^{(k)} = \sum_{j \in \mathcal{N}_k} w_{kj} \hat{y}_t^{(j)} + \hat{\beta}_y(y_{t+1}^{(k)} - \hat{y}_t^{(k)})$ , $\quad \hat{y}_{t+1}^{(k)} = \hat{y}_t^{(k)} + \eta(\tilde{\hat{y}}_{t+1}^{(k)} - \hat{y}_t^{(k)})$ ,
6: $\quad$ Compute variance-reduced gradient $u_t^{(k)}$:
$\quad u_{t+1}^{(k)} = (1 - \rho_x\eta^2)(u_t^{(k)} - \nabla_x F^{(k)}(x_t^{(k)}, y_t^{(k)}; \hat{x}_t^{(k)}, \hat{y}_t^{(k)}; \xi_{t+1}^{(k)})) + \nabla_x F^{(k)}(x_{t+1}^{(k)}, y_{t+1}^{(k)}; \hat{x}_{t+1}^{(k)}, \hat{y}_{t+1}^{(k)}; \xi_{t+1}^{(k)})$ ,
7: $\quad$ Compute variance-reduced gradient $v_t^{(k)}$:
$\quad v_{t+1}^{(k)} = (1 - \rho_y\eta^2)(v_t^{(k)} - \nabla_y F^{(k)}(x_t^{(k)}, y_t^{(k)}; \hat{x}_t^{(k)}, \hat{y}_t^{(k)}; \xi_{t+1}^{(k)})) + \nabla_y F^{(k)}(x_{t+1}^{(k)}, y_{t+1}^{(k)}; \hat{x}_{t+1}^{(k)}, \hat{y}_{t+1}^{(k)}; \xi_{t+1}^{(k)})$ ,
8: $\quad$ Gradient tracking:
$\quad p_{t+1}^{(k)} = \sum_{j \in \mathcal{N}_k} w_{kj} p_t^{(k)} + u_{t+1}^{(k)} - u_t^{(k)}$ , $\quad q_{t+1}^{(k)} = \sum_{j \in \mathcal{N}_k} w_{kj} q_t^{(k)} + v_{t+1}^{(k)} - v_t^{(k)}$ ,
9: **end for**

---

*variables can be well coordinated to avoid divergence.* Inspired by this, we introduce the doubly smoothed loss function, which adds the regularization term to both the primal and dual variables such that the nonconvex-PL loss function becomes strongly-convex-strongly-concave. Specifically, the doubly smoothed loss function is defined as follows:

$$F(x, y; \hat{x}, \hat{y}) = \frac{1}{K} \sum_{k=1}^K \underbrace{f^{(k)}(x, y) + \frac{\gamma_1}{2}\|x - \hat{x}\|^2 - \frac{\gamma_2}{2}\|y - \hat{y}\|^2}_{F^{(k)}(x, y; \hat{x}, \hat{y})} , \qquad (2)$$

where $\gamma_1 > 0$ and $\gamma_2 > 0$ are hyperparameters, and $\hat{x} \in \mathbb{R}^{d_1}, \hat{y} \in \mathbb{R}^{d_2}$ **are the auxiliary variables for the primal and dual variables, respectively**. Here, $\gamma_1$ and $\gamma_2$ are set such that $F(x, y; \hat{x}, \hat{y})$ is strongly convex with respect to $x$ and strongly concave with respect to $y$. For example, we can set $\gamma_1 = 2L$ and $\gamma_2 = 2L$. Note that most existing works in the single-machine setting, such as (Zhang et al., 2020; Yang et al., 2022) apply the smoothing technique to a single variable. Only a recent work (Zheng et al., 2023) uses it for both variables for nonconvex-nonconcave problems. However, it focuses on the deterministic setting, failing to handle the biased stochastic gradient estimator and the decentralized communication. Hence, a new algorithm design and convergence analysis are required to address the challenges caused by them.

Based on the smoothed loss function in Eq. (2), the $k$-th worker can compute the stochastic gradient with respect to the primal and dual variables in the $t$-th iteration as follows:

$$\nabla_x F^{(k)}(x_t^{(k)}, y_t^{(k)}; \hat{x}_t^{(k)}, \hat{y}_t^{(k)}; \xi_t^{(k)}) = \nabla_x f^{(k)}(x_t^{(k)}, y_t^{(k)}; \xi_t^{(k)}) + \gamma_1(x_t^{(k)} - \hat{x}_t^{(k)}) ,$$
$$\nabla_y F^{(k)}(x_t^{(k)}, y_t^{(k)}; \hat{x}_t^{(k)}, \hat{y}_t^{(k)}; \xi_t^{(k)}) = \nabla_y f^{(k)}(x_t^{(k)}, y_t^{(k)}; \xi_t^{(k)}) - \gamma_2(y_t^{(k)} - \hat{y}_t^{(k)}) . \qquad (3)$$

In terms of the smoothed loss function in Eq. (2) and the stochastic gradients in Eq. (3), we develop a novel doubly smoothed decentralized stochastic gradient descent ascent with momentum (**Smoothed$^2$-DSGDAM**) algorithm in Algorithm 1. Generally speaking, we apply the variance reduction technique, STORM (Cutkosky & Orabona, 2019), to the stochastic gradient on each worker to update the primal and dual variables, and use the gradient tracking technique to conduct communication between different workers. However, there are two unique challenges when designing our Smoothed$^2$-DSGDAM algorithm: **1) How to apply the variance reduction technique in the presence of the smoothing term? 2) How to update and communicate the auxiliary variables $\hat{x}$ and $\hat{y}$ to guarantee convergence in the decentralized setting?**

As for the first challenge regarding variance reduction, there are actually two ways to apply variance reduction. Specifically, we can apply it to the original stochastic gradient $\nabla_x f^{(k)}(x_t^{(k)}, y_t^{(k)}; \xi_t^{(k)})$ or to the smoothed gradient $\nabla_x F^{(k)}(x_t^{(k)}, y_t^{(k)}; \hat{x}_t^{(k)}, \hat{y}_t^{(k)}; \xi_t^{(k)})$. However, computing the variance-reduced gradient $u_t^{(k)}$ for the original stochastic gradient $\nabla_x f^{(k)}(x_t^{(k)}, y_t^{(k)}; \xi_t^{(k)})$ will complicate the convergence analysis, when bounding a critical term $\langle \nabla_x F(\bar{x}_t, \bar{y}_t; \hat{\bar{x}}_t, \hat{\bar{y}}_t), \bar{x}_{t+1} - \bar{x}_t \rangle$, where $\bar{x}_t, \bar{y}_t, \hat{\bar{x}}_t$, and $\hat{\bar{y}}_t$ denote the averaged variable across workers.

Specifically, when computing the STORM gradient estimator $u_t^{(k)}$ for the smoothed gradient $\nabla_x F^{(k)}(x_t^{(k)}, y_t^{(k)}; \hat{x}_t^{(k)}, \hat{y}_t^{(k)}; \xi_t^{(k)})$, we can bound it as follows:

$$\langle \nabla_x F(\bar{x}_t, \bar{y}_t; \hat{\bar{x}}_t, \hat{\bar{y}}_t), \bar{x}_{t+1} - \bar{x}_t \rangle = -\eta\beta_x \langle \nabla_x F(\bar{x}_t, \bar{y}_t; \hat{\bar{x}}_t, \hat{\bar{y}}_t), \bar{u}_t \rangle$$

$$= -\frac{\eta\beta_x}{2}\|\bar{u}_t\|^2 - \frac{\eta\beta_x}{2}\|\nabla_x F(\bar{x}_t, \bar{y}_t; \hat{\bar{x}}_t, \hat{\bar{y}}_t)\|^2 + \frac{\eta\beta_x}{2}\|\nabla_x F(\bar{x}_t, \bar{y}_t; \hat{\bar{x}}_t, \hat{\bar{y}}_t) - \bar{u}_t\|^2 . \quad (4)$$

All three terms in the last step are straightforward to handle.

However, when computing the STORM gradient estimator $u_t^{(k)}$ for the original stochastic gradient $\nabla_x f^{(k)}(x_t^{(k)}, y_t^{(k)}; \xi_t^{(k)})$, we have

$$\langle \nabla_x F(\bar{x}_t, \bar{y}_t; \hat{\bar{x}}_t, \hat{\bar{y}}_t), \bar{x}_{t+1} - \bar{x}_t \rangle$$

$$= -\eta\beta_x \langle \nabla_x F(\bar{x}_t, \bar{y}_t; \hat{\bar{x}}_t, \hat{\bar{y}}_t), \gamma_1(\bar{x}_t - \hat{\bar{x}}_t) \rangle - \eta\beta_x \langle \nabla_x F(\bar{x}_t, \bar{y}_t; \hat{\bar{x}}_t, \hat{\bar{y}}_t), \bar{u}_t \rangle$$

$$\leq \frac{\eta\beta_x}{4}\|\nabla_x F(\bar{x}_t, \bar{y}_t; \hat{\bar{x}}_t, \hat{\bar{y}}_t)\|^2 + 4\eta\beta_x \gamma_1^2 \|\bar{x}_t - \hat{\bar{x}}_t\|^2$$

$$- \frac{\eta\beta_x}{2}\|\bar{u}_t\|^2 - \frac{\eta\beta_x}{2}\|\nabla_x F(\bar{x}_t, \bar{y}_t; \hat{\bar{x}}_t, \hat{\bar{y}}_t)\|^2 + \frac{\eta\beta_x}{2}\|\nabla_x F(\bar{x}_t, \bar{y}_t; \hat{\bar{x}}_t, \hat{\bar{y}}_t) - \bar{u}_t\|^2 . \quad (5)$$

Here, the last term should be further handled by $\|\nabla_x F(\bar{x}_t, \bar{y}_t; \hat{\bar{x}}_t, \hat{\bar{y}}_t) - \bar{u}_t\|^2 \leq 2\|\nabla_x f(\bar{x}_t, \bar{y}_t) - \bar{u}_t\|^2 + 2\gamma_1^2 \|\bar{x}_t - \hat{\bar{x}}_t\|^2$. Then, **it can be seen that this approach introduces an addition term (marked in blue), which can make it more challenging to select hyperparameters to handle $\|\bar{x}_t - \hat{\bar{x}}_t\|^2$.**

In addition to this problem, if STORM is applied to the original stochastic gradient, whenever $\|\bar{x}_{t+1} - \bar{x}_t\|^2$ appears, it should be decomposed into two terms: $\|\bar{u}_t\|^2$ and $\|\bar{x}_t - \hat{\bar{x}}_t\|^2$. In contrast, the smoothed one only needs to replace $\|\bar{x}_{t+1} - \bar{x}_t\|^2$ with $\eta^2 \beta_x^2 \|\bar{u}_t\|^2$, which is much easier for the downstream proof. All in all, applying the variance reduction technique to the original stochastic gradient could significantly complicate the proof. Therefore, we apply it to the smoothed gradient $\nabla_x F^{(k)}(x_t^{(k)}, y_t^{(k)}; \hat{x}_t^{(k)}, \hat{y}_t^{(k)}; \xi_t^{(k)})$, which is shown in Step 6 of Algorithm 1.

Regarding the update and communication of the variable $\hat{x}$ and $\hat{y}$, it has not been studied in the existing decentralized optimization literature. A straightforward approach is to update $\hat{x}$ (and $\hat{y}$) locally without communication. However, in convergence analysis, we have to handle the negative term, $-\|\hat{\bar{x}}_{t+1} - \hat{\bar{x}}_t\|^2$ (See Lemma C.1), and positive term, $\|\hat{x}_{t+1}^{(k)} - \hat{x}_t^{(k)}\|^2$ (See Lemma D.1), simultaneously. Specifically, we need to convert $\|\hat{x}_{t+1}^{(k)} - \hat{x}_t^{(k)}\|^2$ to $\|\hat{\bar{x}}_{t+1} - \hat{\bar{x}}_t\|^2$ based on the consensus error $\|\hat{x}_t^{(k)} - \hat{\bar{x}}_t\|^2$. If there is no communication operation for $\hat{x}$, it is difficult to control the consensus error. In fact, it may be exploding. To address this challenge, we propose the following approach for the update and communication of $\hat{x}$ (and $\hat{y}$):

$$\tilde{\hat{x}}_{t+1}^{(k)} = \sum_{j \in \mathcal{N}_k} w_{kj} \hat{x}_t^{(j)} + \hat{\beta}_x(x_{t+1}^{(k)} - \hat{x}_t^{(k)}), \quad \hat{x}_{t+1}^{(k)} = \hat{x}_t^{(k)} + \eta(\tilde{\hat{x}}_{t+1}^{(k)} - \hat{x}_t^{(k)}), \quad (6)$$

where $\hat{\beta}_x > 0$ and $\eta > 0$ are hyperparameters. The first step in Eq. (6) can be viewed as the update of the communicated variable $\sum_{j \in \mathcal{N}_k} w_{kj} \hat{x}_t^{(j)}$ with the local gradient $x_{t+1}^{(k)} - \hat{x}_t^{(k)}$, and the second step is a convex combination between the intermediate variable $\tilde{\hat{x}}_{t+1}^{(k)}$ and the local variable $\hat{x}_t^{(k)}$. With such an update and communication strategy, we can bound the consensus error regarding the auxiliary variable as shown in our Lemma D.3, where the coefficient $1 - \frac{\eta(1-\lambda^2)}{4}$ is important to shrink the consensus error.

In summary, the smoothing technique brings new challenges for algorithm design in the decentralized setting. In our algorithm, we develop novel strategies to handle variance reduction and the update and communication of the auxiliary variables. Therefore, our algorithm design is novel.

## 4 CONVERGENCE ANALYSIS

Before presenting the convergence rate of our algorithm, we introduce the following stationary measures, which were introduced in (Yang et al., 2022).

**Definition 4.1.** *A solution $(x, y)$ is termed the $(\epsilon_1, \epsilon_2)$-stationary solution if $\|\nabla_x f(x,y)\| \leq \epsilon_1$ and $\|\nabla_y f(x,y)\| \leq \epsilon_2$. A solution $x$ is termed the $\epsilon$-stationary solution if $\|\nabla\Phi(x)\| \leq \epsilon$, where $\Phi(x) = f(x, y^*(x))$ and $y^*(x) = \arg\max_{y \in \mathbb{R}^{d_2}} f(x,y)$.*

Based on the assumptions in Section 3, we establish the convergence rate of our Algorithm 1 in the following theorem.

**Theorem 4.2.** *Given Assumptions 3.1-3.4, when $\rho_x > 0$, $\rho_y > 0$, $\gamma = O(L)$, the condition about $\eta$ and $\beta_x$ in Eq. (165), and those about $\beta_y$, $\hat{\beta}_x$, $\hat{\beta}_y$ in Eq. (55) hold, Algorithm 1 has the following convergence upper bound:*

$$\frac{1}{T}\sum_{t=0}^{T-1}\left(\mathbb{E}[\|\nabla_x f(\bar{x}_t, \bar{y}_t)\|^2] + \kappa\mathbb{E}[\|\nabla_y f(\bar{x}_t, \bar{y}_t)\|^2]\right) \leq O(\kappa\rho_x^2\eta^4\sigma^2) + O(\kappa\rho_y^2\eta^4\sigma^2)$$

$$+ O\left(\frac{\kappa\mathcal{P}_0}{\beta_x\eta T}\right) + O\left(\frac{\kappa}{T}\frac{1}{K}\sum_{k=1}^{K}\mathbb{E}[\|\nabla_x f^{(k)}(x_0, y_0)\|^2]\right) + O\left(\frac{\kappa}{T}\frac{1}{K}\sum_{k=1}^{K}\mathbb{E}[\|\nabla_y f^{(k)}(x_0, y_0)\|^2]\right)$$

$$+ O\left(\frac{\kappa\sigma^2}{\rho_x\eta^2 TB}\right) + O\left(\frac{\kappa\sigma^2}{\rho_y\eta^2 TB}\right) + O\left(\frac{\kappa\sigma^2}{T}\right) + O\left(\frac{\kappa\rho_x\eta^2\sigma^2}{K}\right) + O\left(\frac{\kappa\rho_y\eta^2\sigma^2}{K}\right). \tag{7}$$

*where $\mathcal{P}_0 = F(x_0, y_0; \hat{x}_0, \hat{y}_0) - 2F_d(y_0; \hat{x}_0, \hat{y}_0) + 2q(\hat{x}_0)$, whose definitions can be found in Eq. (12).*

**Corollary 4.3.** *Given Assumptions 3.1-3.4, by setting $\beta_x = O((1-\lambda)^2)$, $\beta_y = O((1-\lambda)^2)$, $\hat{\beta}_x = O\left(\frac{(1-\lambda)^2}{\kappa}\right)$, $\hat{\beta}_y = O((1-\lambda)^2)$, $\eta = O\left(\frac{K\epsilon}{\kappa^{1/2}}\right)$, $\rho_x = O\left(\frac{1}{K}\right)$, $\rho_y = O\left(\frac{1}{K}\right)$, $B = O\left(\frac{\kappa^{1/2}}{\epsilon}\right)$, $T = O\left(\frac{\kappa^{3/2}}{K(1-\lambda)^2\epsilon^3}\right)$, Algorithm 1 can achieve the $O(\epsilon, \epsilon/\sqrt{\kappa})$-stationary solution, where $\epsilon > 0$ denotes the solution accuracy, and $B$ is the batch size in the initial iteration.*

**Remark 4.4.** *The actual learning rate of the primal variable is $\beta_x\eta = O\left(\frac{K(1-\lambda)^2\epsilon}{\kappa^{1/2}}\right)$, and that of the dual variable is $\beta_y\eta = O\left(\frac{K(1-\lambda)^2\epsilon}{\kappa^{1/2}}\right)$. Obviously, they are on the same scale in terms of condition number $\kappa$, solution accuracy $\epsilon$, and spectral gap $1 - \lambda$. In addition, the constant batch size based methods, including DM-HSGD (Xian et al., 2021) and DM-GDA (Huang & Chen, 2023), use the learning rate for the primal variable in the order of $O\left(\frac{K(1-\lambda)^2\epsilon}{\kappa^3}\right)$ and that for the dual variable is $O\left(\frac{K(1-\lambda)^2\epsilon}{\kappa^1}\right)$. Apparently, our algorithm can allow a larger learning rate. Moreover, when the number of workers $K = 1$, the spectral gap becomes $1 - \lambda = 1$. Our learning rates $O\left(\frac{\epsilon}{\kappa^{1/2}}\right)$ are larger than $O(\frac{\epsilon^2}{\kappa^1})$ of the single-machine method, Smoothed-SAGDA (Yang et al., 2022).*

The primal–dual learning rate ratio is important because the loss function often exhibits distinct properties for the two variables. When the loss function is nonconvex in the primal variable but satisfies the PL condition in the dual variable, optimizing the primal variable becomes significantly more challenging, and a smaller learning rate is commonly used (See Table 1) to maintain stability and prevent divergence. In contrast, with the smoothing technique, the loss function becomes strongly convex in the primal variable and strongly concave in the dual variable, resulting a well-behaved loss landscape that permits a larger primal learning rate.

**Remark 4.5.** *To compare the convergence rate of our algorithm in Corollary 4.3 with existing algorithms in Table 1, we need to translate the $O(\epsilon, \epsilon/\sqrt{\kappa})$-stationary solution to the $O(\epsilon)$-stationary solution. In particular, (Yang et al., 2022) shows that we can apply stochastic gradient descent ascent algorithm to the optimization problem: $\min_{x \in \mathbb{R}^{d_1}} \max_{y \in \mathbb{R}^{d_2}} f(x,y) + L\|x - x'\|^2$, where $x'$ is the output of our Algorithm 1. Since this problem satisfies the PL condition in both $x$ and $y$, the iteration complexity for the translation is in the order of $\tilde{O}(\frac{1}{\epsilon^2})$, which is apparently dominated by $T = O\left(\frac{\kappa^{3/2}}{K(1-\lambda)^2\epsilon^3}\right)$. Therefore, the iteration complexity to find the $O(\epsilon)$-stationary solution is still $T = O\left(\frac{\kappa^{3/2}}{K(1-\lambda)^2\epsilon^3}\right)$.*

The proof structure and all technical details is provided in Appendix B-E.

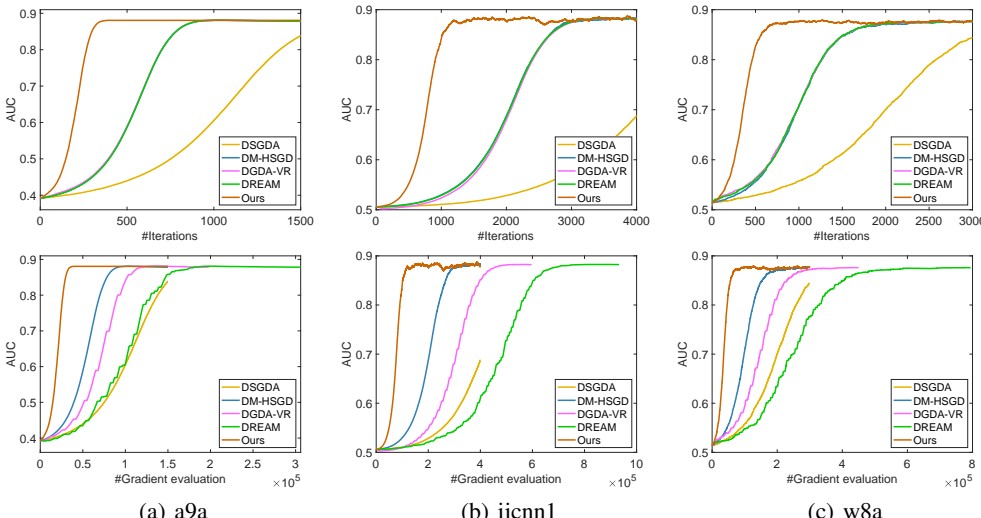

Figure 1: Test AUC vs. Iterations and Gradient Evaluations (Random Graph).

# 5 EXPERIMENTS

In this section, we conduct extensive experiments on AUC maximization, which is defined in Appendix A, to verify the performance of our Algorithm 1.

## 5.1 AUC MAXIMIZATION

**Experimental Settings** We employ three benchmark datasets: a9a, w8a, and ijcnn1, which can be found from LIBSVM Data website [3]. In our experiments, $80\%$ of samples are used as the training set, while the remaining $20\%$ are used for testing. The training samples are randomly distributed across ten workers, where $K = 10$ in our experiment. To evaluate the performance of our algorithm, we compare it with the state-of-the-art decentralized optimization algorithms: DSGDA (Tsaknakis et al., 2020), DM-HSGD [4] (Xian et al., 2021), DGDA-VR (Zhang et al., 2024), and DREAM (Chen et al., 2024). Notably, for DSGDA, we use the stochastic gradient descent ascent instead of the full gradient as described in their paper. For DM-HSGD, the STORM gradient estimator is employed. DGDA-VR leverages the SPIDER gradient estimator in the stochastic setting, while DREAM utilizes the PAGE estimator.

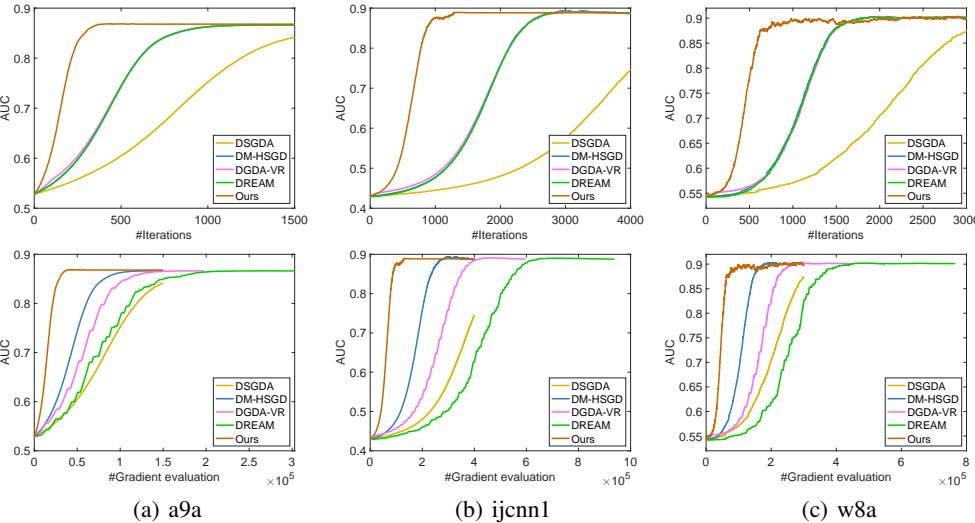

Figure 2: Test AUC vs Iterations and Gradient Evaluations (Line Graph).

[3] https://www.csie.ntu.edu.tw/~cjlin/libsvmtools/datasets/

[4] Note that DM-GDA is the same as DM-HSGD; they differ only in their convergence analysis under different assumptions.

Specifically, we consider two types of communication networks: 1) an Erdos-Renyi random graph with an edge probability of 0.5, and 2) a line communication network where each worker is connected to only two neighboring workers. Throughout all experiments, we fix the solution accuracy $\epsilon$ at 0.01 and use a batch size $b$ of 100. For the a9a and ijcnn1 datasets, the step size of all methods is set to 0.01. Specifically, in our method, $\beta_x$, $\beta_y$, $\hat{\beta}_x$, and $\hat{\beta}_y$ are each set to 0.1, while $\eta$ is set to 0.1, ensuring that their product equals 0.01. For the w8a dataset, the step size of all methods is set to 0.05. In this case, $\beta_x$, $\beta_y$, $\hat{\beta}_x$, and $\hat{\beta}_y$ are each set to 0.5, while $\eta$ remains 0.1, ensuring that their product equals 0.05. Moreover, according to the theoretical results of the baseline methods, the learning rate of the dual variable in DSGDA, DM-HSGD, and DGDA-VR is scaled by $1/\kappa$, while the learning rate of the primal variable is scaled by $1/\kappa^3$. For DREAM, scaling is 1 for the dual variable and $1/\kappa^2$ for the primal variable. Both learning rates in our method are scaled by $1/\kappa^{1/2}$. In our experiments, we assume $\kappa = 1.5$. Additionally, in our method, $\gamma_1$ and $\gamma_2$ are assigned a value of 0.01. For DM-HSGD, the coefficient of the STORM estimator is set to 0.01. Additionally, DGDA-VR computes the full gradient every 100 iterations, while for DREAM, the probability of the PAGE estimator is set to $\frac{\sqrt{b}}{b\sqrt{K}}$.

**Experimental Results** For the random communication graph, we present test AUC versus the number of iterations and gradient evaluations in Figure 1. As shown in Figure 1, our algorithm achieves significantly faster convergence than all baseline methods in terms of the number of iterations, demonstrating its superior efficiency. Furthermore, Figure 1 also indicates that our method also converges more quickly when measured by the number of gradient evaluations, highlighting its lower sample complexity. Notably, DGDA-VR and DREAM incur significantly higher computational cost due to periodic full-gradient computation. These results underscore the efficacy of our algorithm in optimizing performance while maintaining computational efficiency. For the line communication graph, we also present test AUC versus the number of iterations and gradient evaluations in Figure 2. Our method continues to exhibit faster convergence compared to the baseline methods, further validating its effectiveness.

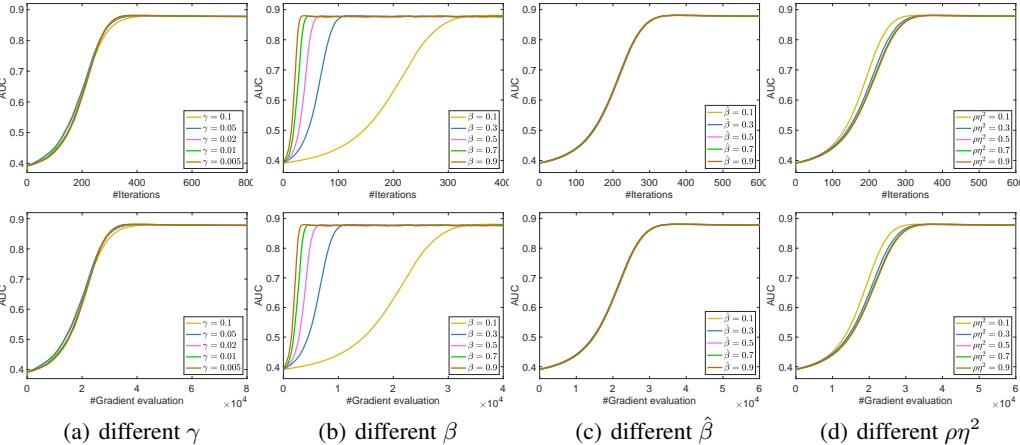

Figure 3: Test AUC under hyperparameters (Random Graph, a9a).

Finally, we evaluate the performance of our method under different values of $\gamma$, $\beta$, $\hat{\beta}$, and $\rho\eta$ in Figure 3, where we set $\gamma_x = \gamma_y = \gamma$, $\beta_x = \beta_y = \beta$, $\hat{\beta}_x = \hat{\beta}_y = \hat{\beta}$, and $\rho_x = \rho_y = \rho$. Our method is robust to all hyperparameters except $\beta$, so they do not require fine-tuning. Since $\beta$ only scales the learning rate, we fix its value, leaving the learning rate $\eta$ as the only hyperparameter to tune.

## 5.2 FAIR CLASSIFICATION

We consider the following nonconvex-PL minimax optimization problem (Nouiehed et al., 2019):

$$\min_x \max_{y\in\mathcal{Y}} \frac{1}{K}\sum_{k=1}^{K}\sum_{c=1}^{C} y_c \mathcal{L}_c^{(k)}(x) - \frac{\lambda}{2}\|y\|^2 \qquad s.t.\ \mathcal{Y} = \{y \in \mathbb{R}^C | y_c \geq 0, \sum_{c=1}^{C} y_c = 1\}. \quad (8)$$

This task serves as a standard benchmark for reweighting classes to improve worst-class performance and has been widely evaluated in federated learning algorithms (Sharma et al., 2022). We conduct

the evaluation in a decentralized setting with eight workers on CIFAR-10 using ResNet-18 (He et al., 2016). In this setup, $\mathcal{L}_c^{(k)}$ represents the cross-entropy loss functions corresponding to class $c$ on worker $k$ for the $C = 10$ classes, and $x$ denotes the model parameters of ResNet-18. We consider three types of communication networks: a random graph, a ring graph, and a torus graph. The learning rate is tuned via grid search and fixed at 0.1, the per-worker batch size is set to 32, and all other hyperparameters and baseline settings remain consistent with the earlier experiments.

Figure 4 reports the test accuracy versus the number of iterations and gradient evaluations. Our algorithm achieves the best overall test accuracy among all baselines and converges faster in terms of gradient evaluations. Although DREAM appears slightly faster in early iterations, its periodic large-batch updates substantially increase the total number of gradient computations, leading to a higher overall computational cost. Overall, our algorithm achieves both superior accuracy and more efficient convergence.

In addition, we evaluate the sensitivity of our method to the hyperparameters $\gamma$ and $\hat{\beta}$, which are the only new hyperparameters introduced compared with existing baselines. As shown in Figure 5, our method remains robust even under this more challenging task, further demonstrating that it does not require complicated hyperparameter tuning.

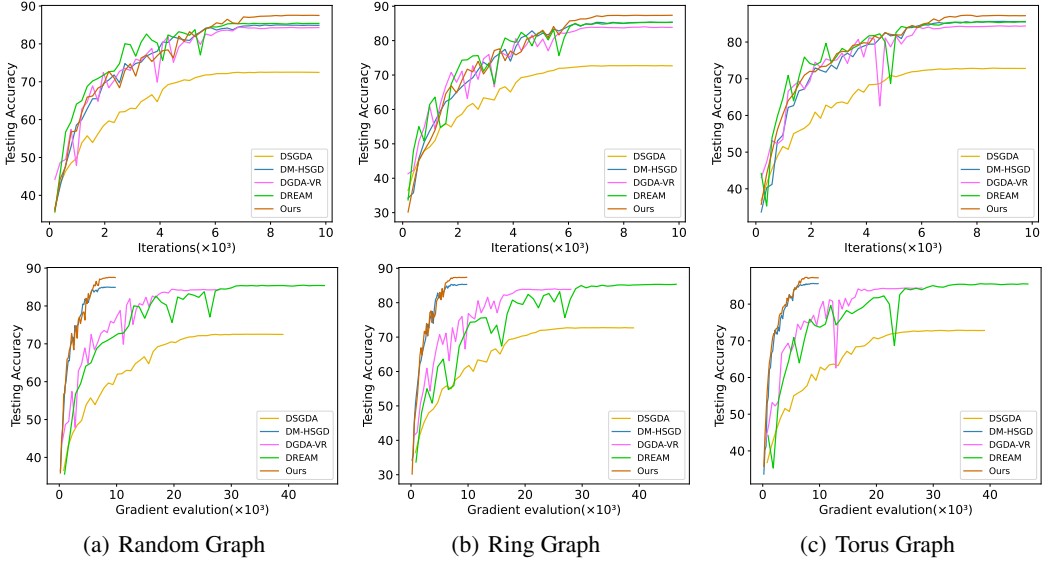

(a) Random Graph        (b) Ring Graph        (c) Torus Graph

Figure 4: Testing Accuracy vs Iterations and Gradient Evaluations on CIFAR-10.

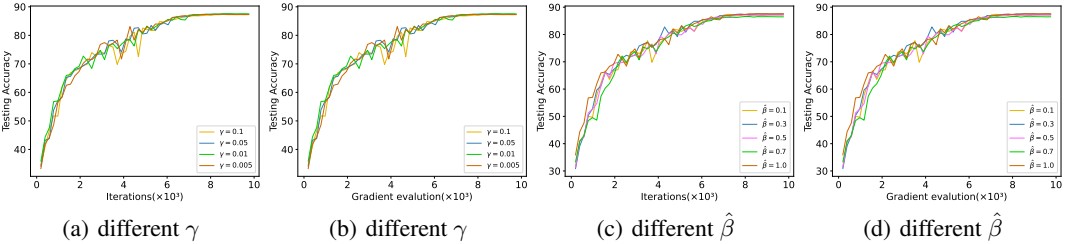

(a) different $\gamma$     (b) different $\gamma$     (c) different $\hat{\beta}$     (d) different $\hat{\beta}$

Figure 5: Testing Accuracy under hyperparameters (Random Graph, CIFAR-10).

## 6 CONCLUSION

In this paper, we developed a novel decentralized minimax optimization algorithm based on the smoothing technique. In particular, our algorithm demonstrates how to incorporate the variance-reduced gradient in the presence of the auxiliary variable and how to perform communication for the auxiliary variable. Moreover, our algorithm can achieve a better dependence on the condition number than all existing methods, which confirms the significance of our algorithm. Finally, experimental results confirm the effectiveness of our algorithm.

**Ethics statement** This research complies with the ICLR Code of Ethics. The study is purely theoretical and methodological, and it does not involve human participants or personally identifiable information. The datasets used in this paper are publicly accessible sources. Our algorithm is designed for advancing machine learning research and does not raise foreseeable risks regarding safety, fairness, privacy, or security.

**Reproducibility statement** To facilitate reproducibility, we provide a comprehensive description of the algorithm and its underlying assumptions in Section 3, with complete theoretical analyses and proofs in Section 4 and Appendix B- E. Experimental details, including datasets, hyperparameter settings, and the communication graph used in decentralized network, are presented in Section 5. Upon acceptance, we will release the full source code to ensure that all reported results can be reliably reproduced.

**The Use of Large Language Models (LLMs)** LLMs were used only to aid in polishing the writing and improving readability of the manuscript. No part of the research ideation, algorithm design, analysis, or experimental results relied on LLMs.

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

## A    AUC MAXIMIZATION

Specifically, we focus on the AUC maximization problem (Ying et al., 2016) for the binary classification task, which is formulated as the following minimax optimization problem (Note that we have included the smoothed term $\gamma_1/2\|x - \hat{x}\|^2$, $\gamma_2/2\|y - \hat{y}\|^2$):

$$
\begin{aligned}
\min_{x,\tilde{x}_1,\tilde{x}_2} \max_{y} \frac{1}{K} \sum_{k=1}^{K} \frac{1}{n} \sum_{i=1}^{n} &\Big( (1-p)(x^T a_i^{(k)} - \tilde{x}_1)^2 \mathbb{I}_{[b_i^{(k)}=1]} \\
&+ 2(1+y)\big( p x^T a_i^{(k)} \mathbb{I}_{[b_i^{(k)}=-1]} - (1-p)x^T a_i^{(k)} \mathbb{I}_{[b_i^{(k)}=1]} \big) \\
&+ p(x^T a_i^{(k)} - \tilde{x}_2)^2 \mathbb{I}_{[b_i^{(k)}=-1]} - p(1-p)y^2 \\
&+ \rho \sum_{j=1}^{d} \frac{x_j^2}{1+x_j^2} + \frac{\gamma_1}{2}\|x - \hat{x}\|^2 - \frac{\gamma_2}{2}\|y - \hat{y}\|^2 \Big) ,
\end{aligned}
\tag{9}
$$

where $x \in \mathbb{R}^d$ is the classifier's parameter, $\tilde{x}_1 \in \mathbb{R}$, $\tilde{x}_2 \in \mathbb{R}$, $y \in \mathbb{R}$ are the parameters to compute the AUC loss, $\hat{x}$ and $\hat{y}$ are the auxiliary variables. $(a_i^{(k)}, b_i^{(k)})$ is the $i$-th sample's feature and label on the $k$-th worker, $p$ is the prior probability of positive class, $\mathbb{I}$ is an indicator function, $\rho$ is a hyperparameter for the regularization term, and $\gamma_1 > 0$, $\gamma_2 > 0$ are hyperparameters for the auxiliary variable. In our experiments, we set $\rho$ to 0.001. Notably, this optimization problem satisfies the nonconvex-PL optimization problem, which can be efficiently solved using our proposed Algorithm 1.

To further evaluate the sensitivity of our method to the hyperparameters $\gamma$ and $\hat{\beta}$, we added on ablation study under the line graph topology, as shown in Figure 6.

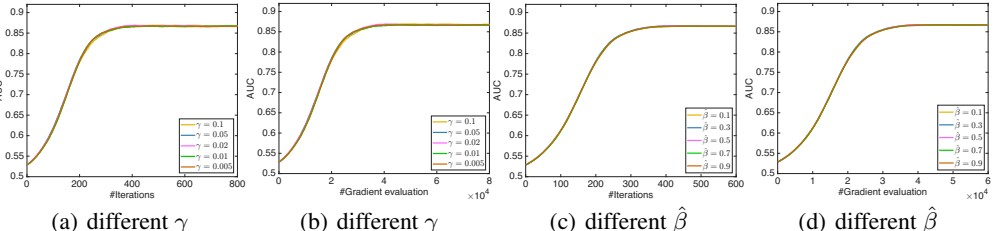

| (a) different $\gamma$ | (b) different $\gamma$ | (c) different $\hat{\beta}$ | (d) different $\hat{\beta}$ |

Figure 6: Test AUC under hyperparameters (Line Graph, a9a).

## B    THE STRUCTURE OF THE PROOF FOR THEOREM 4.2

To make our proof easy to follow, we provide an overview diagram in Figure 7.

It is worth noting that the STORM gradient estimator is a biased gradient estimator, so existing convergence analyzes based on the deterministic gradient (Zhang et al., 2020; Zheng et al., 2023) and the unbiased gradient estimator (Yang et al., 2022) cannot be applied directly to our algorithm. Moreover, most existing stochastic smoothing methods typically apply smoothing only to the primal variable, which makes their analysis insufficient for our algorithm.

In Figure 7, there are actually two key components in our proof: 1) **the optimization error related to doubly smoothing**, 2) **the consensus error and the gradient estimation error related to the decentralized setting**. In Section C, we provide the lemmas for bounding the optimization error. This includes:

- descent-ascent update lemmas (Lemma C.1, Lemma C.2, Lemma C.3);
- optimal solution mappings (Lemma C.4, Lemma C.5);
- auxiliary sequences (Lemma C.6, Lemma C.7).

These results are used in a potential function as Eq.(52):

$$
\mathcal{P}_t = \mathbb{E}[F(\bar{x}_t, \bar{y}_t; \bar{\hat{x}}_t, \bar{\hat{y}}_t)] - 2\mathbb{E}[F_d(\bar{y}_t; \bar{\hat{x}}_t, \bar{\hat{y}}_t)] + 2\mathbb{E}[q(\bar{\hat{x}}_t)] ,
$$

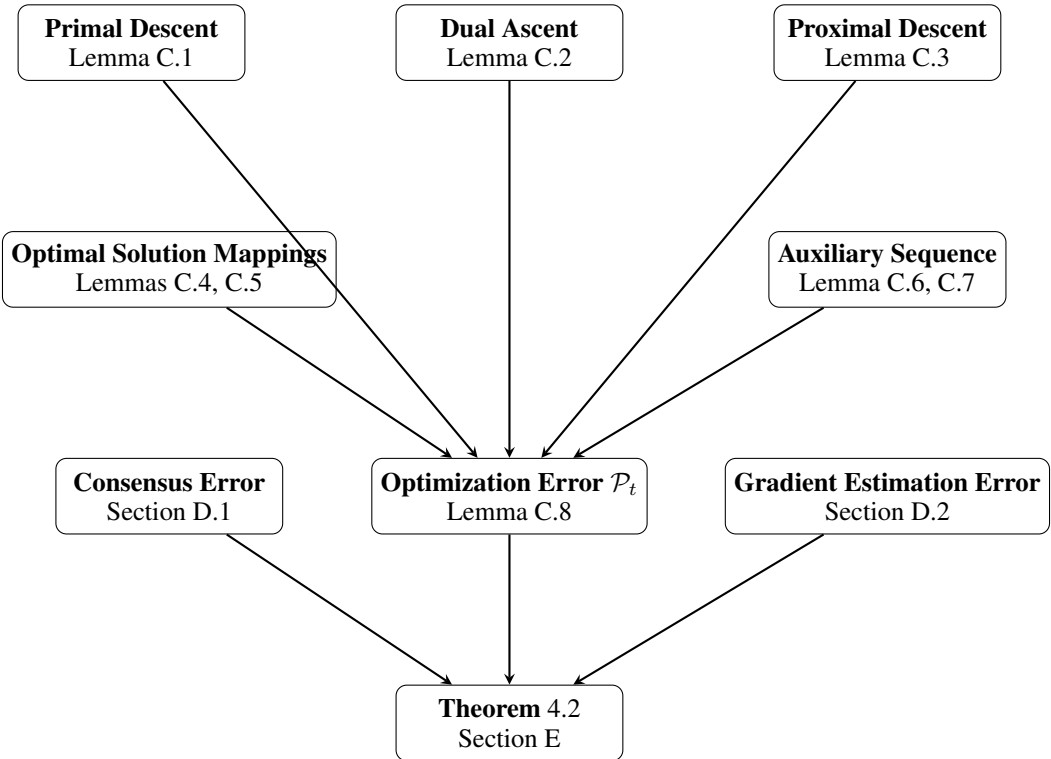

Figure 7: The structure of the proof for Theorem 4.2

to establish the overall optimization error bound $\mathcal{P}_{t+1} - \mathcal{P}_t$ in Lemma C.8. It is worth noted that Lemma C.8 demonstrates that optimization error is affected by the consensus error caused by the decentralized setting and gradient estimation errors. Therefore, in Section D, we address two types of error in the decentralized setting:

- the consensus error, including that of auxiliary variables introduced by smoothing (Section D.1);
- the gradient estimation error from the STORM update (Section D.2).

After establishing all supporting lemmas, we proceed to derive the convergence rate through a novel potential function $\mathcal{L}_t$, which intergrates the optimization error in Lemma C.8 and the consensus error and gradient estimation error together as follows:

$$\mathcal{L}_t = \underbrace{\mathcal{P}_t}_{\text{optimization error}} + c_1 \underbrace{\mathbb{E}[\|\frac{1}{K}\sum_{k=1}^{K} u_t^{(k)} - \frac{1}{K}\sum_{k=1}^{K}\nabla_x F^{(k)}(x_t^{(k)}, y_t^{(k)}; \hat{x}_t^{(k)}, \hat{y}_t^{(k)})\|^2]}_{\text{gradient estimation error}}$$

$$+ c_2 \underbrace{\mathbb{E}[\|\frac{1}{K}\sum_{k=1}^{K} v_t^{(k)} - \frac{1}{K}\sum_{k=1}^{K}\nabla_y F^{(k)}(x_t^{(k)}, y_t^{(k)}; \hat{x}_t^{(k)}, \hat{y}_t^{(k)})\|^2]}_{\text{gradient estimation error}}$$

$$+ \underbrace{c_3 \frac{1}{K}\sum_{k=1}^{K} \mathbb{E}[\|\bar{x}_t - x_t^{(k)}\|^2] + c_4 \frac{1}{K}\sum_{k=1}^{K} \mathbb{E}[\|\bar{y}_t - y_t^{(k)}\|^2] + c_5 \frac{1}{K}\sum_{k=1}^{K} \mathbb{E}[\|\bar{\hat{x}}_t - \hat{x}_t^{(k)}\|^2]}_{\text{consensus error}}$$

$$+ \underbrace{c_{10} \frac{1}{K}\sum_{k=1}^{K} \mathbb{E}[\|\bar{\hat{y}}_t - \hat{y}_t^{(k)}\|^2] + c_6 \frac{1}{K}\sum_{k=1}^{K} \mathbb{E}[\|\bar{p}_t - p_t^{(k)}\|^2] + c_7 \frac{1}{K}\sum_{k=1}^{K} \mathbb{E}[\|\bar{q}_t - q_t^{(k)}\|^2]}_{\text{consensus error}}$$

$$+ c_8 \frac{1}{K} \underbrace{\sum_{k=1}^{K} \mathbb{E}[\|u_t^{(k)} - \nabla_x F^{(k)}(x_t^{(k)}, y_t^{(k)}; \hat{x}_t^{(k)}, \hat{y}_t^{(k)})\|^2]}_{\text{gradient estimation error}}$$

$$+ c_9 \frac{1}{K} \underbrace{\sum_{k=1}^{K} \mathbb{E}[\|v_t^{(k)} - \nabla_y F^{(k)}(x_t^{(k)}, y_t^{(k)}; \hat{x}_t^{(k)}, \hat{y}_t^{(k)})\|^2]}_{\text{gradient estimation error}} .$$

By selecting appropriate hyperparameters, as detailed in Section E, we establish the convergence guarantee stated in Theorem 4.2. The construction of this proof framework is both technically intricate and conceptually non-trivial, underscoring the novelty and difficulty of our analysis.

## B.1 TERMINOLOGIES

To establish the convergence rate of Algorithm 1, we introduce the following symbols:

$$X_t = [x_t^{(1)}, x_t^{(2)}, \cdots, x_t^{(K)}] \in \mathbb{R}^{d_1 \times K} , \ \tilde{X}_t = [\tilde{x}_t^{(1)}, \tilde{x}_t^{(2)}, \cdots, \tilde{x}_t^{(K)}] \in \mathbb{R}^{d_1 \times K} ,$$

$$Y_t = [y_t^{(1)}, y_t^{(2)}, \cdots, y_t^{(K)}] \in \mathbb{R}^{d_2 \times K} , \ \tilde{Y}_t = [\tilde{y}_t^{(1)}, \tilde{y}_t^{(2)}, \cdots, \tilde{y}_t^{(K)}] \in \mathbb{R}^{d_2 \times K} ,$$

$$\hat{X}_t = [\hat{x}_t^{(1)}, \hat{x}_t^{(2)}, \cdots, \hat{x}_t^{(K)}] \in \mathbb{R}^{d_1 \times K} , \ \hat{\tilde{X}}_t = [\hat{\tilde{x}}_t^{(1)}, \hat{\tilde{x}}_t^{(2)}, \cdots, \hat{\tilde{x}}_t^{(K)}] \in \mathbb{R}^{d_1 \times K} ,$$

$$\hat{Y}_t = [\hat{y}_t^{(1)}, \hat{y}_t^{(2)}, \cdots, \hat{y}_t^{(K)}] \in \mathbb{R}^{d_2 \times K} , \ \hat{\tilde{Y}}_t = [\hat{\tilde{y}}_t^{(1)}, \hat{\tilde{y}}_t^{(2)}, \cdots, \hat{\tilde{y}}_t^{(K)}] \in \mathbb{R}^{d_2 \times K} ,$$

$$U_t = [u_t^{(1)}, u_t^{(2)}, \cdots, u_t^{(K)}] \in \mathbb{R}^{d_1 \times K} , \ V_t = [v_t^{(1)}, v_t^{(2)}, \cdots, v_t^{(K)}] \in \mathbb{R}^{d_2 \times K} ,$$

$$P_t = [p_t^{(1)}, p_t^{(2)}, \cdots, p_t^{(K)}] \in \mathbb{R}^{d_1 \times K} , \ Q_t = [q_t^{(1)}, q_t^{(2)}, \cdots, q_t^{(K)}] \in \mathbb{R}^{d_2 \times K} ,$$

$$\bar{X}_t = \frac{1}{K} X_t \mathbf{1} \mathbf{1}^T , \ \bar{Y}_t = \frac{1}{K} Y_t \mathbf{1} \mathbf{1}^T , \ \bar{\hat{X}}_t = \frac{1}{K} \hat{X}_t \mathbf{1} \mathbf{1}^T , \ \bar{\hat{Y}}_t = \frac{1}{K} \hat{Y}_t \mathbf{1} \mathbf{1}^T ,$$

$$\bar{U}_t = \frac{1}{K} U_t \mathbf{1} \mathbf{1}^T , \ \bar{V}_t = \frac{1}{K} V_t \mathbf{1} \mathbf{1}^T , \ \bar{P}_t = \frac{1}{K} P_t \mathbf{1} \mathbf{1}^T , \ \bar{Q}_t = \frac{1}{K} Q_t \mathbf{1} \mathbf{1}^T , \quad (10)$$

where $\mathbf{1} = [1, 1, \cdots, 1]^T \in \mathbb{R}^K$. Based on these terminologies, the update of $x$, $y$, $\hat{x}$, $\hat{y}$, $p$, and $q$ in Algorithm 1 is represented as follows:

$$\tilde{X}_{t+1} = X_t W - \beta_x P_t , X_{t+1} = X_t + \eta(\tilde{X}_{t+1} - X_t) ,$$

$$\tilde{Y}_{t+1} = Y_t W + \beta_y Q_t , Y_{t+1} = Y_t + \eta(\tilde{Y}_{t+1} - Y_t) ,$$

$$\hat{\tilde{X}}_{t+1} = \hat{X}_t W + \hat{\beta}_x(X_{t+1} - \hat{X}_t) , \ \hat{X}_{t+1} = \hat{X}_t + \eta(\hat{\tilde{X}}_{t+1} - \hat{X}_t) ,$$

$$\hat{\tilde{Y}}_{t+1} = \hat{Y}_t W + \hat{\beta}_y(Y_{t+1} - \hat{Y}_t) , \ \hat{Y}_{t+1} = \hat{Y}_t + \eta(\hat{\tilde{Y}}_{t+1} - \hat{Y}_t) ,$$

$$P_{t+1} = P_t W + U_{t+1} - U_t , Q_{t+1} = Q_t W + V_{t+1} - V_t ,$$

$$\bar{X}_{t+1} = \bar{X}_t - \beta_x \eta \bar{U}_t , \bar{Y}_{t+1} = \bar{Y}_t + \beta_y \eta \bar{V}_t ,$$

$$\bar{\hat{X}}_{t+1} = \bar{\hat{X}}_t + \hat{\beta}_x \eta(\bar{X}_{t+1} - \bar{\hat{X}}_t) , \bar{\hat{Y}}_{t+1} = \bar{\hat{Y}}_t + \hat{\beta}_y \eta(\bar{Y}_{t+1} - \bar{\hat{Y}}_t) . \quad (11)$$

Note that $\bar{P}_t = \bar{U}_t$ and $\bar{Q}_t = \bar{V}_t$.

Moreover, following (Yang et al., 2022; Zheng et al., 2023), we introduce the following auxiliary functions and variables for convergence analysis:

$$F_d(y; \hat{x}, \hat{y}) = \min_{x \in \mathbb{R}^{d_1}} F(x, y; \hat{x}, \hat{y}) , \quad \text{dual function}$$

$$F_p(x; \hat{x}, \hat{y}) = \max_{y \in \mathbb{R}^{d_2}} F(x, y; \hat{x}, \hat{y}) , \quad \text{primal function}$$

$$g(\hat{x}, \hat{y}) = \min_{x \in \mathbb{R}^{d_1}} \max_{y \in \mathbb{R}^{d_2}} F(x, y; \hat{x}, \hat{y}) ,$$

$$p(\hat{y}) = \min_{\hat{x} \in \mathbb{R}^{d_1}} g(\hat{x}, \hat{y}) , \quad q(\hat{x}) = \max_{\hat{y} \in \mathbb{R}^{d_2}} g(\hat{x}, \hat{y}) ,$$

$$x^*(y; \hat{x}, \hat{y}) = \arg\min_{x \in \mathbb{R}^{d_1}} F(x, y; \hat{x}, \hat{y}) ,$$

$$y^*(x; \hat{x}, \hat{y}) = \arg\max_{y \in \mathbb{R}^{d_2}} F(x, y; \hat{x}, \hat{y}) \,,$$

$$x^*(\hat{x}, \hat{y}) \triangleq x^*(y^*(\hat{x}, \hat{y}); \hat{x}, \hat{y}) = \arg\min_{x \in \mathbb{R}^{d_1}} F_p(x; \hat{x}, \hat{y}) \,,$$

$$y^*(\hat{x}, \hat{y}) \triangleq y^*(x^*(\hat{x}, \hat{y}); \hat{x}, \hat{y}) = \arg\max_{y \in \mathbb{R}^{d_2}} F_d(y; \hat{x}, \hat{y}) \,,$$

$$\hat{x}^*(\hat{y}) = \arg\min_{\hat{x} \in \mathbb{R}^{d_1}} g(\hat{x}, \hat{y}) \,, \qquad \hat{y}^*(\hat{x}) = \arg\max_{\hat{y} \in \mathbb{R}^{d_2}} g(\hat{x}, \hat{y}) \,,$$

$$y^+(\hat{x}_t, \hat{y}_t) = y_t + \beta_y \eta \nabla_y F_d(y_t; \hat{x}_t, \hat{y}_t) \,,$$

$$\hat{y}^+(\hat{x}_{t+1}) = \hat{y}_t + \hat{\beta}_y \eta (y^*(\hat{x}_t, \hat{y}_t) - \hat{y}_t) \,. \tag{12}$$

## B.2 FUNCTION PROPERTIES

**Lemma B.1.** *(Zheng et al., 2023) Given Assumptions 3.1-3.4, then $F(x, y; \hat{x}, \hat{y})$ is $(\gamma_1 + L)$-smooth and $(\gamma_1 - L)$-strongly convex with respect to $x$. $F(x, y; \hat{x}, \hat{y})$ is $(\gamma_2 + L)$-smooth and $(\gamma_2 - L)$-strongly concave with respect to $y$.*

**Lemma B.2.** *(Zheng et al., 2023) Given Assumptions 3.1-3.4, the following inequality holds:*

$$\|x^*(y_1; \hat{x}, \hat{y}) - x^*(y_2; \hat{x}, \hat{y})\| \leq C_{x^1_{y\hat{x}\hat{y}}} \|y_1 - y_2\| \,,$$

$$\|x^*(y; \hat{x}_1, \hat{y}) - x^*(y; \hat{x}_2, \hat{y})\| \leq C_{x^2_{y\hat{x}\hat{y}}} \|\hat{x}_1 - \hat{x}_2\| \,,$$

$$\|x^*(\hat{x}_1, \hat{y}) - x^*(\hat{x}_2, \hat{y})\| \leq C_{x^1_{\hat{x}\hat{y}}} \|\hat{x}_1 - \hat{x}_2\| \,,$$

$$\|y^*(x_1; \hat{x}, \hat{y}) - y^*(x_2; \hat{x}, \hat{y})\| \leq C_{y^1_{x\hat{x}\hat{y}}} \|x_1 - x_2\| \,,$$

$$\|y^*(x; \hat{x}, \hat{y}_1) - y^*(x; \hat{x}, \hat{y}_2)\| \leq C_{y^3_{x\hat{x}\hat{y}}} \|\hat{y}_1 - \hat{y}_2\| \,,$$

$$\|y^*(\hat{x}_1, \hat{y}) - y^*(\hat{x}_2, \hat{y})\| \leq C_{y^1_{\hat{x}\hat{y}}} \|\hat{x}_1 - \hat{x}_2\| \,,$$

$$\|y^*(\hat{x}, \hat{y}_1) - y^*(\hat{x}, \hat{y}_2)\| \leq C_{y^2_{\hat{x}\hat{y}}} \|\hat{y}_1 - \hat{y}_2\| \,, \tag{13}$$

*where*

$$C_{x^1_{y\hat{x}\hat{y}}} = \frac{\gamma_1}{\gamma_1 - L} \,, \quad C_{x^2_{y\hat{x}\hat{y}}} = \frac{\gamma_1}{\gamma_1 - L} \,, \quad C_{x^1_{\hat{x}\hat{y}}} = \frac{\gamma_1}{\gamma_1 - L} \,,$$

$$C_{y^1_{x\hat{x}\hat{y}}} = \frac{\gamma_2}{\gamma_2 - L} \,, \quad C_{y^3_{x\hat{x}\hat{y}}} = \frac{\gamma_2}{\gamma_2 - L} \,, \quad C_{y^1_{\hat{x}\hat{y}}} = \frac{\gamma_1}{\gamma_2 - L} C_{x^1_{y\hat{x}\hat{y}}} + 1 \,, C_{y^2_{\hat{x}\hat{y}}} = \frac{\gamma_2}{\gamma_2 - L} \,. \tag{14}$$

**Lemma B.3.** *(Zheng et al., 2023) Given Assumptions 3.1-3.4, then $F_d(y; \hat{x}, \hat{y})$ is $L_d$-smooth, where $L_d = LC_{x^1_{y\hat{x}\hat{y}}} + L + \gamma_2$.*

**Lemma B.4.** *Given Assumptions 3.1-3.4, by defining $y^+(\hat{x}_t, \hat{y}_t) = y_t + \beta_y \eta \nabla_y F_d(y_t; \hat{x}_t, \hat{y}_t)$, the following inequality holds:*

$$\|y_t - y^*(\hat{x}_t, \hat{y}_t)\| \leq \frac{1}{\beta_y \eta (\gamma_2 - L)} \|y_t - y^+(\hat{x}_t, \hat{y}_t)\| \,. \tag{15}$$

*Proof.* Due to $y^*(\hat{x}_t, \hat{y}_t) = \arg\max_{y \in \mathbb{R}^{d_2}} F_d(y; \hat{x}_t, \hat{y}_t)$, for any $y \in \mathbb{R}^{d_2}$, we have

$$\langle y - y^*(\hat{x}_t, \hat{y}_t), \nabla_y F_d(y^*(\hat{x}_t, \hat{y}_t); \hat{x}_t, \hat{y}_t) \rangle \leq 0. \tag{16}$$

By taking $y = y_t$, we have

$$\langle y_t - y^*(\hat{x}_t, \hat{y}_t), \nabla_y F_d(y^*(\hat{x}_t, \hat{y}_t); \hat{x}_t, \hat{y}_t) \rangle \leq 0 \,. \tag{17}$$

In addition, because $F_d(y; \hat{x}, \hat{y})$ is $(\gamma_2 - L)$-strongly concave with respect to $y$, we have

$$\langle y_t - y^*(\hat{x}_t, \hat{y}_t), \nabla_y F_d(y_t; \hat{x}_t, \hat{y}_t) - \nabla_y F_d(y^*(\hat{x}_t, \hat{y}_t); \hat{x}_t, \hat{y}_t) \rangle + (\gamma_2 - L)\|y_t - y^*(\hat{x}_t, \hat{y}_t)\|^2 \leq 0 \,. \tag{18}$$

By combining the above two inequalities, we have

$$\langle y_t - y^*(\hat{x}_t, \hat{y}_t), \nabla_y F_d(y_t; \hat{x}_t, \hat{y}_t) \rangle + (\gamma_2 - L)\|y_t - y^*(\hat{x}_t, \hat{y}_t)\|^2 \leq 0 \,. \tag{19}$$

Then, we can obtain

$$(\gamma_2 - L)\|y_t - y^*(\hat{x}_t, \hat{y}_t)\|^2 \leq \langle y^*(\hat{x}_t, \hat{y}_t) - y_t, \nabla_y F_d(y_t; \hat{x}_t, \hat{y}_t)\rangle$$

$$\leq \|y_t - y^*(\hat{x}_t, \hat{y}_t)\|\|\nabla_y F_d(y_t; \hat{x}_t, \hat{y}_t)\| = \|y_t - y^*(\hat{x}_t, \hat{y}_t)\|\|\frac{y^+(\hat{x}_t, \hat{y}_t) - y_t}{\beta_y \eta}\| . \quad (20)$$

As a result, we have

$$\|y_t - y^*(\hat{x}_t, \hat{y}_t)\| \leq \frac{1}{\beta_y \eta(\gamma_2 - L)}\|y^+(\hat{x}_t, \hat{y}_t) - y_t\| . \quad (21)$$

$\square$

**Lemma B.5.** *Given Assumptions 3.1-3.4, then*

$$\|x_t - x^*(y_t; \hat{x}_t, \hat{y}_t)\| \leq \frac{1}{\gamma_1 - L}\|\nabla_x F(x_t, y_t; \hat{x}_t, \hat{y}_t)\|. \quad (22)$$

*Proof.* Due to $x^*(y_t; \hat{x}_t, \hat{y}_t) = \arg\min_{x \in \mathbb{R}^{d_1}} F(x, y_t; \hat{x}_t, \hat{y}_t)$, for any $x \in \mathbb{R}^{d_1}$, we have

$$\langle x - x^*(y_t; \hat{x}_t, \hat{y}_t), -\nabla_x F(x^*(y_t; \hat{x}_t, \hat{y}_t), y_t; \hat{x}_t, \hat{y}_t)\rangle \leq 0. \quad (23)$$

By taking $x = x_t$, we have

$$\langle x_t - x^*(y_t; \hat{x}_t, \hat{y}_t), \nabla_x F(x^*(y_t; \hat{x}_t, \hat{y}_t), y_t; \hat{x}_t, \hat{y}_t)\rangle \geq 0. \quad (24)$$

In addition, because $F(x, y; \hat{x}, \hat{y})$ is $(\gamma_1 - L)$-strongly convex with respect to $x$, we have

$$\langle x_t - x^*(y_t; \hat{x}_t, \hat{y}_t), \nabla_x F(x_t, y_t; \hat{x}_t, \hat{y}_t) - \nabla_x F(x^*(y_t; \hat{x}_t, \hat{y}_t), y_t; \hat{x}_t, \hat{y}_t)\rangle$$

$$\geq (\gamma_1 - L)\|x_t - x^*(y_t; \hat{x}_t, \hat{y}_t)\|^2 . \quad (25)$$

By combing the above two inequalities, we have

$$(\gamma_1 - L)\|x_t - x^*(y_t; \hat{x}_t, \hat{y}_t)\|^2 \leq \langle x_t - x^*(y_t; \hat{x}_t, \hat{y}_t), \nabla_x F(x_t, y_t; \hat{x}_t, \hat{y}_t)\rangle$$

$$\leq \|x_t - x^*(y_t; \hat{x}_t, \hat{y}_t)\|\|\nabla_x F(x_t, y_t; \hat{x}_t, \hat{y}_t)\| . \quad (26)$$

As a result, we have

$$\|x_t - x^*(y_t; \hat{x}_t, \hat{y}_t)\| \leq \frac{1}{\gamma_1 - L}\|\nabla_x F(x_t, y_t; \hat{x}_t, \hat{y}_t)\|. \quad (27)$$

$\square$

## C  OPTIMIZATION ERRORS

**Lemma C.1.** *Given Assumptions 3.1-3.4 and $\eta \leq \frac{1}{2\beta_x(\gamma_1+L)}$, the following inequality holds:*

$$\mathbb{E}[F(\bar{x}_{t+1}, \bar{y}_{t+1}; \bar{\hat{x}}_{t+1}, \bar{\hat{y}}_{t+1})] - \mathbb{E}[F(\bar{x}_t, \bar{y}_t; \bar{\hat{x}}_t, \bar{\hat{y}}_t)]$$

$$\leq -\frac{\beta_x \eta}{2}\mathbb{E}[\|\nabla_x F(\bar{x}_t, \bar{y}_t; \bar{\hat{x}}_t, \bar{\hat{y}}_t)\|^2] + \frac{\beta_y \eta}{2}\mathbb{E}[\|\nabla_y F(\bar{x}_t, \bar{y}_t; \bar{\hat{x}}_t, \bar{\hat{y}}_t)\|^2] + (4\beta_y \eta \beta_x^2 \eta^2 L^2 - \frac{\beta_x \eta}{4})\mathbb{E}[\|\bar{u}_t\|^2]$$

$$+ (\frac{3\beta_y \eta}{4} + \frac{\beta_y^2 \eta^2(\gamma_2 + L)}{2})\mathbb{E}[\|\bar{v}_t\|^2] + \frac{\beta_x \eta}{2}\mathbb{E}[\|\nabla_x F(\bar{x}_t, \bar{y}_t; \bar{\hat{x}}_t, \bar{\hat{y}}_t) - \bar{u}_t\|^2]$$

$$- \frac{\gamma_1(2 - \hat{\beta}_x \eta)}{2\hat{\beta}_x \eta}\mathbb{E}[\|\bar{\hat{x}}_{t+1} - \bar{\hat{x}}_t\|^2] - \frac{\gamma_2(\hat{\beta}_y \eta - 2)}{2\hat{\beta}_y \eta}\mathbb{E}[\|\bar{\hat{y}}_{t+1} - \bar{\hat{y}}_t\|^2] . \quad (28)$$

*Proof.* Because $F(x, y; \hat{x}, \hat{y})$ is $(L + \gamma_1)$-smooth with respect to $x$, we have

$$\mathbb{E}[F(\bar{x}_{t+1}, \bar{y}_t; \bar{\hat{x}}_t, \bar{\hat{y}}_t)]$$

$$\leq \mathbb{E}[F(\bar{x}_t, \bar{y}_t; \bar{\hat{x}}_t, \bar{\hat{y}}_t)] + \mathbb{E}[\langle \nabla_x F(\bar{x}_t, \bar{y}_t; \bar{\hat{x}}_t, \bar{\hat{y}}_t), \bar{x}_{t+1} - \bar{x}_t \rangle] + \frac{L + \gamma_1}{2} \mathbb{E}[\|\bar{x}_{t+1} - \bar{x}_t\|^2]$$

$$= \mathbb{E}[F(\bar{x}_t, \bar{y}_t; \bar{\hat{x}}_t, \bar{\hat{y}}_t)] - \beta_x \eta \mathbb{E}[\langle \nabla_x F(\bar{x}_t, \bar{y}_t; \bar{\hat{x}}_t, \bar{\hat{y}}_t), \bar{u}_t \rangle] + \frac{\beta_x^2 \eta^2 (L + \gamma_1)}{2} \mathbb{E}[\|\bar{u}_t\|^2]$$

$$= \mathbb{E}[F(\bar{x}_t, \bar{y}_t; \bar{\hat{x}}_t, \bar{\hat{y}}_t)] - \frac{\beta_x \eta}{2} \mathbb{E}[\|\nabla_x F(\bar{x}_t, \bar{y}_t; \bar{\hat{x}}_t, \bar{\hat{y}}_t)\|^2] - \frac{\beta_x \eta}{2} \mathbb{E}[\|\bar{u}_t\|^2]$$

$$+ \frac{\beta_x \eta}{2} \mathbb{E}[\|\nabla_x F(\bar{x}_t, \bar{y}_t; \bar{\hat{x}}_t, \bar{\hat{y}}_t) - \bar{u}_t\|^2] + \frac{\beta_x^2 \eta^2 (L + \gamma_1)}{2} \mathbb{E}[\|\bar{u}_t\|^2]$$

$$\leq \mathbb{E}[F(\bar{x}_t, \bar{y}_t; \bar{\hat{x}}_t, \bar{\hat{y}}_t)] - \frac{\beta_x \eta}{2} \mathbb{E}[\|\nabla_x F(\bar{x}_t, \bar{y}_t; \bar{\hat{x}}_t, \bar{\hat{y}}_t)\|^2] - \frac{\beta_x \eta}{4} \mathbb{E}[\|\bar{u}_t\|^2]$$

$$+ \frac{\beta_x \eta}{2} \mathbb{E}[\|\nabla_x F(\bar{x}_t, \bar{y}_t; \bar{\hat{x}}_t, \bar{\hat{y}}_t) - \bar{u}_t\|^2] , \tag{29}$$

where the last step holds due to $\eta \leq \frac{1}{2\beta_x(\gamma_1 + L)}$.

In addition, because $F(x, y; \hat{x}, \hat{y})$ is $(L + \gamma_1)$-smooth with respect to $y$, we have

$$\mathbb{E}[F(\bar{x}_{t+1}, \bar{y}_{t+1}; \bar{\hat{x}}_t, \bar{\hat{y}}_t)]$$

$$\leq \mathbb{E}[F(\bar{x}_{t+1}, \bar{y}_t; \bar{\hat{x}}_t, \bar{\hat{y}}_t)] + \mathbb{E}[\langle \nabla_y F(\bar{x}_{t+1}, \bar{y}_t; \bar{\hat{x}}_t, \bar{\hat{y}}_t), \bar{y}_{t+1} - \bar{y}_t \rangle] + \frac{\gamma_2 + L}{2} \mathbb{E}[\|\bar{y}_{t+1} - \bar{y}_t\|^2]$$

$$= \mathbb{E}[F(\bar{x}_{t+1}, \bar{y}_t; \bar{\hat{x}}_t, \bar{\hat{y}}_t)] + \beta_y \eta \mathbb{E}[\langle \nabla_y F(\bar{x}_{t+1}, \bar{y}_t; \bar{\hat{x}}_t, \bar{\hat{y}}_t) - \nabla_y F(\bar{x}_t, \bar{y}_t; \bar{\hat{x}}_t, \bar{\hat{y}}_t), \bar{v}_t \rangle]$$

$$+ \beta_y \eta \mathbb{E}[\langle \nabla_y F(\bar{x}_t, \bar{y}_t; \bar{\hat{x}}_t, \bar{\hat{y}}_t), \bar{v}_t \rangle] + \frac{\beta_y^2 \eta^2 (\gamma_2 + L)}{2} \mathbb{E}[\|\bar{v}_t\|^2]$$

$$\leq \mathbb{E}[F(\bar{x}_{t+1}, \bar{y}_t; \bar{\hat{x}}_t, \bar{\hat{y}}_t)] + 4\beta_y \eta \mathbb{E}[\|\nabla_y F(\bar{x}_{t+1}, \bar{y}_t; \bar{\hat{x}}_t, \bar{\hat{y}}_t) - \nabla_y F(\bar{x}_t, \bar{y}_t; \bar{\hat{x}}_t, \bar{\hat{y}}_t)\|^2] + \frac{\beta_y \eta}{4} \mathbb{E}[\|\bar{v}_t\|^2]$$

$$+ \frac{\beta_y \eta}{2} \mathbb{E}[\|\nabla_y F(\bar{x}_t, \bar{y}_t; \bar{\hat{x}}_t, \bar{\hat{y}}_t)\|^2] + \frac{\beta_y \eta}{2} \mathbb{E}[\|\bar{v}_t\|^2] + \frac{\beta_y^2 \eta^2 (\gamma_2 + L)}{2} \mathbb{E}[\|\bar{v}_t\|^2]$$

$$\leq \mathbb{E}[F(\bar{x}_{t+1}, \bar{y}_t; \bar{\hat{x}}_t, \bar{\hat{y}}_t)] + \frac{\beta_y \eta}{2} \mathbb{E}[\|\nabla_y F(\bar{x}_t, \bar{y}_t; \bar{\hat{x}}_t, \bar{\hat{y}}_t)\|^2]$$

$$+ 4\beta_y \eta \beta_x^2 \eta^2 L^2 \mathbb{E}[\|\bar{u}_t\|^2] + \left( \frac{3\beta_y \eta}{4} + \frac{\beta_y^2 \eta^2 (\gamma_2 + L)}{2} \right) \mathbb{E}[\|\bar{v}_t\|^2] , \tag{30}$$

where the last step holds due to the following inequality.

$$\mathbb{E}[\|\nabla_y F(\bar{x}_{t+1}, \bar{y}_t; \bar{\hat{x}}_t, \bar{\hat{y}}_t) - \nabla_y F(\bar{x}_t, \bar{y}_t; \bar{\hat{x}}_t, \bar{\hat{y}}_t)\|^2]$$

$$= \mathbb{E}[\|\nabla_y f(\bar{x}_{t+1}, \bar{y}_t) + \gamma_2 (\bar{y}_t - \bar{\hat{y}}_t) - \nabla_y f(\bar{x}_t, \bar{y}_t) - \gamma_2 (\bar{y}_t - \bar{\hat{y}}_t)\|^2]$$

$$\leq L^2 \mathbb{E}[\|\bar{x}_{t+1} - \bar{x}_t\|^2] \leq \beta_x^2 \eta^2 L^2 \mathbb{E}[\|\bar{u}_t\|^2] . \tag{31}$$

By combining Eq. (29) and Eq. (30), we have

$$\mathbb{E}[F(\bar{x}_{t+1}, \bar{y}_{t+1}; \bar{\hat{x}}_t, \bar{\hat{y}}_t)]$$

$$\leq \mathbb{E}[F(\bar{x}_t, \bar{y}_t; \bar{\hat{x}}_t, \bar{\hat{y}}_t)] - \frac{\beta_x \eta}{2} \mathbb{E}[\|\nabla_x F(\bar{x}_t, \bar{y}_t; \bar{\hat{x}}_t, \bar{\hat{y}}_t)\|^2] + \frac{\beta_y \eta}{2} \mathbb{E}[\|\nabla_y F(\bar{x}_t, \bar{y}_t; \bar{\hat{x}}_t, \bar{\hat{y}}_t)\|^2]$$

$$+ \frac{\beta_x \eta}{2} \mathbb{E}[\|\nabla_x F(\bar{x}_t, \bar{y}_t; \bar{\hat{x}}_t, \bar{\hat{y}}_t) - \bar{u}_t\|^2]$$

$$+ (4\beta_y \eta \beta_x^2 \eta^2 L^2 - \frac{\beta_x \eta}{4}) \mathbb{E}[\|\bar{u}_t\|^2] + (\frac{3\beta_y \eta}{4} + \frac{\beta_y^2 \eta^2 (\gamma_2 + L)}{2}) \mathbb{E}[\|\bar{v}_t\|^2] . \tag{32}$$

Moreover, according to the definition of $F(x, y; \hat{x}, \hat{y})$, we have

$$F(\bar{x}_{t+1}, \bar{y}_{t+1}; \bar{\hat{x}}_t, \bar{\hat{y}}_t) - F(\bar{x}_{t+1}, \bar{y}_{t+1}; \bar{\hat{x}}_{t+1}, \bar{\hat{y}}_t)$$

$$= f(\bar{x}_{t+1}, \bar{y}_{t+1}) + \frac{\gamma_1}{2} \|\bar{x}_{t+1} - \bar{\hat{x}}_t\|^2 - \frac{\gamma_2}{2} \|\bar{y}_{t+1} - \bar{\hat{y}}_t\|^2$$

$$- f(\bar{x}_{t+1}, \bar{y}_{t+1}) - \frac{\gamma_1}{2} \|\bar{x}_{t+1} - \bar{\hat{x}}_{t+1}\|^2 + \frac{\gamma_2}{2} \|\bar{y}_{t+1} - \bar{\hat{y}}_t\|^2$$

$$= \frac{\gamma_1}{2} \left( \|\bar{x}_{t+1} - \bar{\hat{x}}_t\|^2 - \|\bar{x}_{t+1} - \bar{\hat{x}}_{t+1}\|^2 \right)$$

$$= \frac{\gamma_1}{2} \left( \|\bar{x}_{t+1} - \hat{\bar{x}}_t\|^2 - \|(1 - \hat{\beta}_x \eta)(\bar{x}_{t+1} - \hat{\bar{x}}_t)\|^2 \right)$$

$$= \frac{\gamma_1(1 - (1 - \hat{\beta}_x \eta)^2)}{2} \|\bar{x}_{t+1} - \hat{\bar{x}}_t\|^2$$

$$= \frac{\gamma_1(1 - (1 - \hat{\beta}_x \eta)^2)}{2\hat{\beta}_x^2 \eta^2} \|\hat{\bar{x}}_{t+1} - \hat{\bar{x}}_t\|^2$$

$$= \frac{\gamma_1(2 - \hat{\beta}_x \eta)}{2\hat{\beta}_x \eta} \|\hat{\bar{x}}_{t+1} - \hat{\bar{x}}_t\|^2 , \tag{33}$$

where the third and fifth steps hold due to $\hat{\bar{x}}_{t+1} = \hat{\bar{x}}_t + \hat{\beta}_x \eta(\bar{x}_{t+1} - \hat{\bar{x}}_t)$.

Similarly, we have

$$F(\bar{x}_{t+1}, \bar{y}_{t+1}; \hat{\bar{x}}_{t+1}, \hat{\bar{y}}_t) - F(\bar{x}_{t+1}, \bar{y}_{t+1}; \hat{\bar{x}}_{t+1}, \hat{\bar{y}}_{t+1})$$

$$= f(\bar{x}_{t+1}, \bar{y}_{t+1}) + \frac{\gamma_1}{2} \|\bar{x}_{t+1} - \hat{\bar{x}}_{t+1}\|^2 - \frac{\gamma_2}{2} \|\bar{y}_{t+1} - \hat{\bar{y}}_t\|^2$$

$$- f(\bar{x}_{t+1}, \bar{y}_{t+1}) - \frac{\gamma_1}{2} \|\bar{x}_{t+1} - \hat{\bar{x}}_{t+1}\|^2 + \frac{\gamma_2}{2} \|\bar{y}_{t+1} - \hat{\bar{y}}_{t+1}\|^2$$

$$= \frac{\gamma_2}{2} \|\bar{y}_{t+1} - \hat{\bar{y}}_{t+1}\|^2 - \frac{\gamma_2}{2} \|\bar{y}_{t+1} - \hat{\bar{y}}_t\|^2$$

$$= \frac{\gamma_2(\hat{\beta}_y \eta - 2)}{2\hat{\beta}_y \eta} \|\hat{\bar{y}}_{t+1} - \hat{\bar{y}}_t\|^2 . \tag{34}$$

By combining the above three inequalities, we have

$$\mathbb{E}[F(\bar{x}_{t+1}, \bar{y}_{t+1}; \hat{\bar{x}}_{t+1}, \hat{\bar{y}}_{t+1})] - \mathbb{E}[F(\bar{x}_t, \bar{y}_t; \hat{\bar{x}}_t, \hat{\bar{y}}_t)]$$

$$= \mathbb{E}[F(\bar{x}_{t+1}, \bar{y}_{t+1}; \hat{\bar{x}}_{t+1}, \hat{\bar{y}}_{t+1}) - F(\bar{x}_{t+1}, \bar{y}_{t+1}; \hat{\bar{x}}_{t+1}, \hat{\bar{y}}_t)$$

$$+ \left( F(\bar{x}_{t+1}, \bar{y}_{t+1}; \hat{\bar{x}}_{t+1}, \hat{\bar{y}}_t) - F(\bar{x}_{t+1}, \bar{y}_{t+1}; \hat{\bar{x}}_t, \hat{\bar{y}}_t) \right) + \left( F(\bar{x}_{t+1}, \bar{y}_{t+1}; \hat{\bar{x}}_t, \hat{\bar{y}}_t) - F(\bar{x}_t, \bar{y}_t; \hat{\bar{x}}_t, \hat{\bar{y}}_t) \right)]$$

$$\leq -\frac{\beta_x \eta}{2} \mathbb{E}[\|\nabla_x F(\bar{x}_t, \bar{y}_t; \hat{\bar{x}}_t, \hat{\bar{y}}_t)\|^2] + \frac{\beta_y \eta}{2} \mathbb{E}[\|\nabla_y F(\bar{x}_t, \bar{y}_t; \hat{\bar{x}}_t, \hat{\bar{y}}_t)\|^2]$$

$$+ \frac{\beta_x \eta}{2} \mathbb{E}[\|\nabla_x F(\bar{x}_t, \bar{y}_t; \hat{\bar{x}}_t, \hat{\bar{y}}_t) - \bar{u}_t\|^2]$$

$$+ (4\beta_y \eta \beta_x^2 \eta^2 L^2 - \frac{\beta_x \eta}{4}) \mathbb{E}[\|\bar{u}_t\|^2] + (\frac{3\beta_y \eta}{4} + \frac{\beta_y^2 \eta^2(\gamma_2 + L)}{2}) \mathbb{E}[\|\bar{v}_t\|^2]$$

$$- \frac{\gamma_1(2 - \hat{\beta}_x \eta)}{2\hat{\beta}_x \eta} \mathbb{E}[\|\hat{\bar{x}}_{t+1} - \hat{\bar{x}}_t\|^2] - \frac{\gamma_2(\hat{\beta}_y \eta - 2)}{2\hat{\beta}_y \eta} \mathbb{E}[\|\hat{\bar{y}}_{t+1} - \hat{\bar{y}}_t\|^2] . \tag{35}$$

$\square$

**Lemma C.2.** *Given Assumptions 3.1-3.4, the following inequality holds:*

$$\mathbb{E}[F_d(\bar{y}_{t+1}; \hat{\bar{x}}_{t+1}, \hat{\bar{y}}_{t+1})] - \mathbb{E}[F_d(\bar{y}_t; \hat{\bar{x}}_t, \hat{\bar{y}}_t)]$$

$$\geq \beta_y \eta \mathbb{E}[\langle \nabla_y F_d(\bar{y}_t; \hat{\bar{x}}_t, \hat{\bar{y}}_t), \bar{v}_t \rangle] - \frac{\beta_y^2 \eta^2 L_d}{2} \mathbb{E}[\|\bar{v}_t\|^2] + \frac{\gamma_2(2 - \hat{\beta}_y \eta)}{2\hat{\beta}_y \eta} \mathbb{E}[\|\hat{\bar{y}}_{t+1} - \hat{\bar{y}}_t\|^2]$$

$$+ \frac{\gamma_1}{2} \mathbb{E}[\langle \hat{\bar{x}}_{t+1} - \hat{\bar{x}}_t, \hat{\bar{x}}_{t+1} + \hat{\bar{x}}_t - 2x^*(\bar{y}_{t+1}; \hat{\bar{x}}_{t+1}, \hat{\bar{y}}_t) \rangle] . \tag{36}$$

*Proof.* According to the definition of $F_d(y; \hat{x}, \hat{y})$, we have

$$F_d(\bar{y}_{t+1}; \hat{\bar{x}}_{t+1}, \hat{\bar{y}}_{t+1}) - F_d(\bar{y}_{t+1}; \hat{\bar{x}}_{t+1}, \hat{\bar{y}}_t)$$

$$= F(x^*(\bar{y}_{t+1}; \hat{\bar{x}}_{t+1}, \hat{\bar{y}}_{t+1}), \bar{y}_{t+1}; \hat{\bar{x}}_{t+1}, \hat{\bar{y}}_{t+1}) - F(x^*(\bar{y}_{t+1}; \hat{\bar{x}}_{t+1}, \hat{\bar{y}}_t), \bar{y}_{t+1}; \hat{\bar{x}}_{t+1}, \hat{\bar{y}}_t)$$

$$\geq F(x^*(\bar{y}_{t+1}; \hat{\bar{x}}_{t+1}, \hat{\bar{y}}_{t+1}), \bar{y}_{t+1}; \hat{\bar{x}}_{t+1}, \hat{\bar{y}}_{t+1}) - F(x^*(\bar{y}_{t+1}; \hat{\bar{x}}_{t+1}, \hat{\bar{y}}_{t+1}), \bar{y}_{t+1}; \hat{\bar{x}}_{t+1}, \hat{\bar{y}}_t)$$

$$= \frac{\gamma_2}{2}(\|\bar{y}_{t+1} - \hat{\bar{y}}_t\|^2 - \|\bar{y}_{t+1} - \hat{\bar{y}}_{t+1}\|^2)$$

$$= \frac{\gamma_2(2 - \hat{\beta}_y\eta)}{2\hat{\beta}_y\eta}\|\bar{\hat{y}}_{t+1} - \bar{\hat{y}}_t\|^2 \,, \tag{37}$$

where the second step holds due to $x^*(\bar{y}_{t+1}; \bar{\hat{x}}_{t+1}, \bar{\hat{y}}_t) = \arg\min_{x\in\mathbb{R}^{d_1}} F(x, \bar{y}_{t+1}; \bar{\hat{x}}_{t+1}, \bar{\hat{y}}_t)$, the last step holds as Eq. (34).

In addition, according to the definition of $F_d(y; \hat{x}, \hat{y})$, we have

$$
\begin{aligned}
&F_d(\bar{y}_{t+1}; \bar{\hat{x}}_{t+1}, \bar{\hat{y}}_t) - F_d(\bar{y}_{t+1}; \bar{\hat{x}}_t, \bar{\hat{y}}_t) \\
&= F(x^*(\bar{y}_{t+1}; \bar{\hat{x}}_{t+1}, \bar{\hat{y}}_t), \bar{y}_{t+1}; \bar{\hat{x}}_{t+1}, \bar{\hat{y}}_t) - F(x^*(\bar{y}_{t+1}; \bar{\hat{x}}_t, \bar{\hat{y}}_t), \bar{y}_{t+1}; \bar{\hat{x}}_t, \bar{\hat{y}}_t) \\
&\geq F(x^*(\bar{y}_{t+1}; \bar{\hat{x}}_{t+1}, \bar{\hat{y}}_t), \bar{y}_{t+1}; \bar{\hat{x}}_{t+1}, \bar{\hat{y}}_t) - F(x^*(\bar{y}_{t+1}; \bar{\hat{x}}_{t+1}, \bar{\hat{y}}_t), \bar{y}_{t+1}; \bar{\hat{x}}_t, \bar{\hat{y}}_t) \\
&= \frac{\gamma_1}{2}(\|x^*(\bar{y}_{t+1}; \bar{\hat{x}}_{t+1}, \bar{\hat{y}}_t) - \bar{\hat{x}}_{t+1}\|^2 - \|x^*(\bar{y}_{t+1}; \bar{\hat{x}}_{t+1}, \bar{\hat{y}}_t) - \bar{\hat{x}}_t\|^2) \\
&= \frac{\gamma_1}{2}\langle x^*(\bar{y}_{t+1}; \bar{\hat{x}}_{t+1}, \bar{\hat{y}}_t) - \bar{\hat{x}}_{t+1} - (x^*(\bar{y}_{t+1}; \bar{\hat{x}}_{t+1}, \bar{\hat{y}}_t) - \bar{\hat{x}}_t), \\
&\qquad x^*(\bar{y}_{t+1}; \bar{\hat{x}}_{t+1}, \bar{\hat{y}}_t) - \bar{\hat{x}}_{t+1} + (x^*(\bar{y}_{t+1}; \bar{\hat{x}}_{t+1}, \bar{\hat{y}}_t) - \bar{\hat{x}}_t)\rangle \\
&= \frac{\gamma_1}{2}\langle \bar{\hat{x}}_{t+1} - \bar{\hat{x}}_t, \bar{\hat{x}}_{t+1} + \bar{\hat{x}}_t - 2x^*(\bar{y}_{t+1}; \bar{\hat{x}}_{t+1}, \bar{\hat{y}}_t)\rangle \,,
\end{aligned}
\tag{38}
$$

where the second step holds due to $x^*(\bar{y}_{t+1}; \bar{\hat{x}}_t, \bar{\hat{y}}_t) = \arg\min_{x\in\mathbb{R}^{d_1}} F(x, \bar{y}_{t+1}; \bar{\hat{x}}_t, \bar{\hat{y}}_t)$, the fourth step holds due to the fact $a^2 - b^2 = (a - b)(a + b)$.

Moreover, because $F_d(y; \hat{x}, \hat{y})$ is $L_d$-smooth, we have

$$
\begin{aligned}
F_d(\bar{y}_{t+1}; \bar{\hat{x}}_t, \bar{\hat{y}}_t) &\geq F_d(\bar{y}_t; \bar{\hat{x}}_t, \bar{\hat{y}}_t) + \langle\nabla_y F_d(\bar{y}_t; \bar{\hat{x}}_t, \bar{\hat{y}}_t), \bar{y}_{t+1} - \bar{y}_t\rangle - \frac{L_d}{2}\|\bar{y}_{t+1} - \bar{y}_t\|^2 \\
&= F_d(\bar{y}_t; \bar{\hat{x}}_t, \bar{\hat{y}}_t) + \beta_y\eta\langle\nabla_y F_d(\bar{y}_t; \bar{\hat{x}}_t, \bar{\hat{y}}_t), \bar{v}_t\rangle - \frac{\beta_y^2\eta^2 L_d}{2}\|\bar{v}_t\|^2 \,.
\end{aligned}
\tag{39}
$$

By combining the above three inequalities, we have

$$
\begin{aligned}
&\mathbb{E}[F_d(\bar{y}_{t+1}; \bar{\hat{x}}_{t+1}, \bar{\hat{y}}_{t+1})] - \mathbb{E}[F_d(\bar{y}_t; \bar{\hat{x}}_t, \bar{\hat{y}}_t)] \\
&= \mathbb{E}[F_d(\bar{y}_{t+1}; \bar{\hat{x}}_{t+1}, \bar{\hat{y}}_{t+1}) - F_d(\bar{y}_{t+1}; \bar{\hat{x}}_{t+1}, \bar{\hat{y}}_t) \\
&\quad + \Big(F_d(\bar{y}_{t+1}; \bar{\hat{x}}_{t+1}, \bar{\hat{y}}_t) - F_d(\bar{y}_{t+1}; \bar{\hat{x}}_t, \bar{\hat{y}}_t)\Big) + \Big(F_d(\bar{y}_{t+1}; \bar{\hat{x}}_t, \bar{\hat{y}}_t) - F_d(\bar{y}_t; \bar{\hat{x}}_t, \bar{\hat{y}}_t)]\Big) \\
&\geq \beta_y\eta\mathbb{E}[\langle\nabla_y F_d(\bar{y}_t; \bar{\hat{x}}_t, \bar{\hat{y}}_t), \bar{v}_t\rangle] - \frac{\beta_y^2\eta^2 L_d}{2}\mathbb{E}[\|\bar{v}_t\|^2] + \frac{\gamma_2(2 - \hat{\beta}_y\eta)}{2\hat{\beta}_y\eta}\mathbb{E}[\|\bar{\hat{y}}_{t+1} - \bar{\hat{y}}_t\|^2] \\
&\quad + \frac{\gamma_1}{2}\mathbb{E}[\langle\bar{\hat{x}}_{t+1} - \bar{\hat{x}}_t, \bar{\hat{x}}_{t+1} + \bar{\hat{x}}_t - 2x^*(\bar{y}_{t+1}; \bar{\hat{x}}_{t+1}, \bar{\hat{y}}_t)\rangle] \,.
\end{aligned}
\tag{40}
$$

$\square$

**Lemma C.3.** *Given Assumptions 3.1-3.4, the following inequality holds:*

$$q(\bar{\hat{x}}_{t+1}) - q(\bar{\hat{x}}_t) \leq \frac{\gamma_1}{2}\langle\bar{\hat{x}}_{t+1} - \bar{\hat{x}}_t, \bar{\hat{x}}_{t+1} + \bar{\hat{x}}_t - 2x^*(\bar{\hat{x}}_t, \hat{y}^*(\bar{\hat{x}}_{t+1}))\rangle \,. \tag{41}$$

*Proof.*

$$
\begin{aligned}
&q(\bar{\hat{x}}_{t+1}) - q(\bar{\hat{x}}_t) \\
&= g(\bar{\hat{x}}_{t+1}, \hat{y}^*(\bar{\hat{x}}_{t+1})) - g(\bar{\hat{x}}_t, \hat{y}^*(\bar{\hat{x}}_t)) \\
&= F_p(x^*(\bar{\hat{x}}_{t+1}, \hat{y}^*(\bar{\hat{x}}_{t+1})); \bar{\hat{x}}_{t+1}, \hat{y}^*(\bar{\hat{x}}_{t+1})) - F_p(x^*(\bar{\hat{x}}_t, \hat{y}^*(\bar{\hat{x}}_t)); \bar{\hat{x}}_t, \hat{y}^*(\bar{\hat{x}}_t)) \\
&\leq F_p(x^*(\bar{\hat{x}}_{t+1}, \hat{y}^*(\bar{\hat{x}}_{t+1})); \bar{\hat{x}}_{t+1}, \hat{y}^*(\bar{\hat{x}}_{t+1})) - F_p(x^*(\bar{\hat{x}}_t, \hat{y}^*(\bar{\hat{x}}_{t+1})); \bar{\hat{x}}_t, \hat{y}^*(\bar{\hat{x}}_{t+1})) \\
&\leq F_p(x^*(\bar{\hat{x}}_t, \hat{y}^*(\bar{\hat{x}}_{t+1})); \bar{\hat{x}}_{t+1}, \hat{y}^*(\bar{\hat{x}}_{t+1})) - F_p(x^*(\bar{\hat{x}}_t, \hat{y}^*(\bar{\hat{x}}_{t+1})); \bar{\hat{x}}_t, \hat{y}^*(\bar{\hat{x}}_{t+1})) \\
&= F(x^*(\bar{\hat{x}}_t, \hat{y}^*(\bar{\hat{x}}_{t+1})), y^*(x^*(\bar{\hat{x}}_t, \hat{y}^*(\bar{\hat{x}}_{t+1})); \bar{\hat{x}}_{t+1}, \hat{y}^*(\bar{\hat{x}}_{t+1})); \bar{\hat{x}}_{t+1}, \hat{y}^*(\bar{\hat{x}}_{t+1})) \\
&\quad - F(x^*(\bar{\hat{x}}_t, \hat{y}^*(\bar{\hat{x}}_{t+1})), y^*(x^*(\bar{\hat{x}}_t, \hat{y}^*(\bar{\hat{x}}_{t+1})); \bar{\hat{x}}_t, \hat{y}^*(\bar{\hat{x}}_{t+1})); \bar{\hat{x}}_t, \hat{y}^*(\bar{\hat{x}}_{t+1})) \\
&\leq F(x^*(\bar{\hat{x}}_t, \hat{y}^*(\bar{\hat{x}}_{t+1})), y^*(x^*(\bar{\hat{x}}_t, \hat{y}^*(\bar{\hat{x}}_{t+1})); \bar{\hat{x}}_{t+1}, \hat{y}^*(\bar{\hat{x}}_{t+1})); \bar{\hat{x}}_{t+1}, \hat{y}^*(\bar{\hat{x}}_{t+1}))
\end{aligned}
$$

$$- F(x^*(\bar{\hat{x}}_t, \hat{y}^*(\bar{\hat{x}}_{t+1})), y^*(x^*(\bar{\hat{x}}_t, \hat{y}^*(\bar{\hat{x}}_{t+1})); \bar{\hat{x}}_{t+1}, \hat{y}^*(\bar{\hat{x}}_{t+1})); \bar{\hat{x}}_t, \hat{y}^*(\bar{\hat{x}}_{t+1}))$$

$$= \frac{\gamma_1}{2}(\|x^*(\bar{\hat{x}}_t, \hat{y}^*(\bar{\hat{x}}_{t+1})) - \bar{\hat{x}}_{t+1}\|^2 - \|x^*(\bar{\hat{x}}_t, \hat{y}^*(\bar{\hat{x}}_{t+1})) - \bar{\hat{x}}_t\|^2)$$

$$= \frac{\gamma_1}{2}\langle x^*(\bar{\hat{x}}_t, \hat{y}^*(\bar{\hat{x}}_{t+1})) - \bar{\hat{x}}_{t+1} - (x^*(\bar{\hat{x}}_t, \hat{y}^*(\bar{\hat{x}}_{t+1})) - \bar{\hat{x}}_t),$$

$$x^*(\bar{\hat{x}}_t, \hat{y}^*(\bar{\hat{x}}_{t+1})) - \bar{\hat{x}}_{t+1} + (x^*(\bar{\hat{x}}_t, \hat{y}^*(\bar{\hat{x}}_{t+1})) - \bar{\hat{x}}_t)\rangle$$

$$= \frac{\gamma_1}{2}\langle \bar{\hat{x}}_{t+1} - \bar{\hat{x}}_t, \bar{\hat{x}}_{t+1} + \bar{\hat{x}}_t - 2x^*(\bar{\hat{x}}_t, \hat{y}^*(\bar{\hat{x}}_{t+1}))\rangle \,, \tag{42}$$

where the second step holds due to $g(\hat{x}, \hat{y}) = \min_{x \in \mathbb{R}^{d_1}} F_p(x; \hat{x}, \hat{y})$, the three inequalities hold due to $y^*(x; \hat{x}, \hat{y}) = \arg\max_y F(x, y; \hat{x}, \hat{y})$ and $F_p(x; \hat{x}, \hat{y}) = F(x, y^*(x; \hat{x}, \hat{y}); \hat{x}, \hat{y})$, the second to last step holds due to the fact $a^2 - b^2 = (a - b)(a + b)$.

$\square$

**Lemma C.4.** *Given Assumptions 3.1-3.4, the following inequality holds:*

$$\|x^*(\bar{\hat{x}}_{t+1}, \hat{y}^+(\bar{\hat{x}}_{t+1})) - x^*(\bar{\hat{x}}_{t+1}, \hat{y}^*(\bar{\hat{x}}_{t+1}))\|^2$$

$$\leq \frac{2}{\gamma_1 - L}\frac{2\gamma_2^2 C_{y_{\hat{x}\hat{y}}^1}^2}{\mu}\|\bar{\hat{x}}_{t+1} - \bar{\hat{x}}_t\|^2 + \frac{2}{\gamma_1 - L}\frac{2\gamma_2^2}{\mu}\left(C_{y_{\hat{x}\hat{y}}^2}^2 + \frac{(1 - \hat{\beta}_y\eta)^2}{\hat{\beta}_y^2\eta^2}\right)\|\bar{\hat{y}}_t - \hat{y}^+(\bar{\hat{x}}_{t+1})\|^2 \,. \tag{43}$$

*Proof.*

$$\frac{\gamma_1 - L}{2}\|x^*(\bar{\hat{x}}_{t+1}, \hat{y}^+(\bar{\hat{x}}_{t+1})) - x^*(\bar{\hat{x}}_{t+1}, \hat{y}^*(\bar{\hat{x}}_{t+1}))\|^2$$

$$\leq F_p(x^*(\bar{\hat{x}}_{t+1}, \hat{y}^+(\bar{\hat{x}}_{t+1})); \bar{\hat{x}}_{t+1}, \hat{y}^*(\bar{\hat{x}}_{t+1})) - F_p(x^*(\bar{\hat{x}}_{t+1}, \hat{y}^*(\bar{\hat{x}}_{t+1})); \bar{\hat{x}}_{t+1}, \hat{y}^*(\bar{\hat{x}}_{t+1}))$$

$$\leq \max_{\hat{y} \in \mathbb{R}^{d_2}} F_p(x^*(\bar{\hat{x}}_{t+1}, \hat{y}^+(\bar{\hat{x}}_{t+1})); \bar{\hat{x}}_{t+1}, \hat{y}) - F_p(x^*(\bar{\hat{x}}_{t+1}, \hat{y}^*(\bar{\hat{x}}_{t+1})); \bar{\hat{x}}_{t+1}, \hat{y}^*(\bar{\hat{x}}_{t+1}))$$

$$\leq \max_{\hat{y} \in \mathbb{R}^{d_2}} F_p(x^*(\bar{\hat{x}}_{t+1}, \hat{y}^+(\bar{\hat{x}}_{t+1})); \bar{\hat{x}}_{t+1}, \hat{y}) - F_p(x^*(\bar{\hat{x}}_{t+1}, \hat{y}^+(\bar{\hat{x}}_{t+1})); \bar{\hat{x}}_{t+1}, \hat{y}^+(\bar{\hat{x}}_{t+1}))$$

$$\leq \frac{1}{2\mu}\|\nabla_{\hat{y}} F_p(x^*(\bar{\hat{x}}_{t+1}, \hat{y}^+(\bar{\hat{x}}_{t+1})); \bar{\hat{x}}_{t+1}, \hat{y}^+(\bar{\hat{x}}_{t+1}))\|^2$$

$$= \frac{\gamma_2^2}{2\mu}\|y^*(x^*(\bar{\hat{x}}_{t+1}, \hat{y}^+(\bar{\hat{x}}_{t+1})); \bar{\hat{x}}_{t+1}, \hat{y}^+(\bar{\hat{x}}_{t+1})) - \hat{y}^+(\bar{\hat{x}}_{t+1})\|^2$$

$$= \frac{\gamma_2^2}{2\mu}\|y^*(x^*(\bar{\hat{x}}_{t+1}, \hat{y}^+(\bar{\hat{x}}_{t+1})); \bar{\hat{x}}_{t+1}, \hat{y}^+(\bar{\hat{x}}_{t+1})) - \bar{\hat{y}}_t - \hat{\beta}_y\eta(y^*(\bar{\hat{x}}_t, \bar{\hat{y}}_t) - \bar{\hat{y}}_t)\|^2$$

$$\leq \frac{\gamma_2^2}{\mu}\|y^*(\bar{\hat{x}}_{t+1}, \hat{y}^+(\bar{\hat{x}}_{t+1})) - y^*(\bar{\hat{x}}_t, \bar{\hat{y}}_t)\|^2 + (1 - \hat{\beta}_y\eta)^2\frac{\gamma_2^2}{\mu}\|\bar{\hat{y}}_t - y^*(\bar{\hat{x}}_t, \bar{\hat{y}}_t)\|^2$$

$$\leq \frac{2\gamma_2^2}{\mu}\|y^*(\bar{\hat{x}}_{t+1}, \hat{y}^+(\bar{\hat{x}}_{t+1})) - y^*(\bar{\hat{x}}_t, \hat{y}^+(\bar{\hat{x}}_{t+1}))\|^2 + \frac{2\gamma_2^2}{\mu}\|y^*(\bar{\hat{x}}_t, \hat{y}^+(\bar{\hat{x}}_{t+1})) - y^*(\bar{\hat{x}}_t, \bar{\hat{y}}_t)\|^2$$

$$+ (1 - \hat{\beta}_y\eta)^2\frac{\gamma_2^2}{\mu}\|\bar{\hat{y}}_t - y^*(\bar{\hat{x}}_t, \bar{\hat{y}}_t)\|^2$$

$$\leq \frac{2\gamma_2^2 C_{y_{\hat{x}\hat{y}}^1}^2}{\mu}\|\bar{\hat{x}}_{t+1} - \bar{\hat{x}}_t\|^2 + \frac{2\gamma_2^2 C_{y_{\hat{x}\hat{y}}^2}^2}{\mu}\|\hat{y}^+(\bar{\hat{x}}_{t+1}) - \bar{\hat{y}}_t\|^2 + \frac{\gamma_2^2}{\mu}\frac{(1 - \hat{\beta}_y\eta)^2}{\hat{\beta}_y^2\eta^2}\|\hat{y}^+(\bar{\hat{x}}_{t+1}) - \bar{\hat{y}}_t\|^2$$

$$\leq \frac{2\gamma_2^2 C_{y_{\hat{x}\hat{y}}^1}^2}{\mu}\|\bar{\hat{x}}_{t+1} - \bar{\hat{x}}_t\|^2 + \frac{2\gamma_2^2}{\mu}\left(C_{y_{\hat{x}\hat{y}}^2}^2 + \frac{(1 - \hat{\beta}_y\eta)^2}{\hat{\beta}_y^2\eta^2}\right)\|\bar{\hat{y}}_t - \hat{y}^+(\bar{\hat{x}}_{t+1})\|^2 \,, \tag{44}$$

where the first step holds because $F_p(x; \hat{x}, \hat{y})$ is $(\gamma_1 - L)$-strongly convex with respect to $x$, the fourth step holds due to Theorem 5.2 of (Yu et al., 2022) with PL property being a special KL property, the fifth step holds due to the definition of $F_p$, the sixth step and the last step hold due to the definition $\hat{y}^+(\bar{\hat{x}}_{t+1}) = \bar{\hat{y}}_t + \hat{\beta}_y\eta(y^*(\bar{\hat{x}}_t, \bar{\hat{y}}_t) - \bar{\hat{y}}_t)$.

$\square$

**Lemma C.5.** *Given Assumptions 3.1-3.4, the following inequality holds:*

$$\mathbb{E}[\|x^*(\bar{y}_{t+1}; \bar{\hat{x}}_{t+1}, \bar{\hat{y}}_t) - x^*(\bar{\hat{x}}_{t+1}, \hat{y}^+(\bar{\hat{x}}_{t+1}))\|^2]$$

$$\leq 10\beta_y^2\eta^2 C_{x^1_{y\hat{x}\hat{y}}}^2 \mathbb{E}[\|\bar{v}_t - \nabla_y F(\bar{x}_t, \bar{y}_t; \bar{\hat{x}}_t, \bar{\hat{y}}_t)\|^2] + 10\beta_y^2\eta^2 L^2 C_{x^1_{y\hat{x}\hat{y}}}^2 \mathbb{E}[\|\bar{x}_t - x^*(\bar{y}_t; \bar{\hat{x}}_t, \bar{\hat{y}}_t)\|^2]$$

$$+ 5C_{x^1_{y\hat{x}\hat{y}}}^2 \left(1 + \frac{1}{\beta_y^2\eta^2(\gamma_2 - L)^2}\right) \mathbb{E}[\|y^+(\bar{\hat{x}}_t, \bar{\hat{y}}_t) - \bar{y}_t\|^2]$$

$$+ 5C_{x^1_{y\hat{x}\hat{y}}}^2 C_{y^1_{\hat{x}\hat{y}}}^2 \mathbb{E}[\|\bar{\hat{x}}_t - \bar{\hat{x}}_{t+1}\|^2] + 5C_{x^1_{y\hat{x}\hat{y}}}^2 C_{y^2_{\hat{x}\hat{y}}}^2 \mathbb{E}[\|\bar{\hat{y}}_t - \hat{y}^+(\bar{\hat{x}}_{t+1})\|^2] . \tag{45}$$

*Proof.*

$$\mathbb{E}[\|x^*(\bar{y}_{t+1}; \bar{\hat{x}}_{t+1}, \bar{\hat{y}}_t) - x^*(\bar{\hat{x}}_{t+1}, \hat{y}^+(\bar{\hat{x}}_{t+1}))\|^2]$$

$$\leq C_{x^1_{y\hat{x}\hat{y}}}^2 \mathbb{E}[\|\bar{y}_{t+1} - y^*(\bar{\hat{x}}_{t+1}, \hat{y}^+(\bar{\hat{x}}_{t+1}))\|^2]$$

$$= C_{x^1_{y\hat{x}\hat{y}}}^2 \mathbb{E}[\|\bar{y}_{t+1} - y^+(\bar{\hat{x}}_t, \bar{\hat{y}}_t) + y^+(\bar{\hat{x}}_t, \bar{\hat{y}}_t) - \bar{y}_t + \bar{y}_t - y^*(\bar{\hat{x}}_t, \bar{\hat{y}}_t)$$

$$+ y^*(\bar{\hat{x}}_t, \bar{\hat{y}}_t) - y^*(\bar{\hat{x}}_{t+1}, \bar{\hat{y}}_t) + y^*(\bar{\hat{x}}_{t+1}, \bar{\hat{y}}_t) - y^*(\bar{\hat{x}}_{t+1}, \hat{y}^+(\bar{\hat{x}}_{t+1}))\|^2]$$

$$\leq 5C_{x^1_{y\hat{x}\hat{y}}}^2 \mathbb{E}[\|\bar{y}_{t+1} - y^+(\bar{\hat{x}}_t, \bar{\hat{y}}_t)\|^2] + 5C_{x^1_{y\hat{x}\hat{y}}}^2 \mathbb{E}[\|y^+(\bar{\hat{x}}_t, \bar{\hat{y}}_t) - \bar{y}_t\|^2] + 5C_{x^1_{y\hat{x}\hat{y}}}^2 \mathbb{E}[\|\bar{y}_t - y^*(\bar{\hat{x}}_t, \bar{\hat{y}}_t)\|^2]$$

$$+ 5C_{x^1_{y\hat{x}\hat{y}}}^2 \mathbb{E}[\|y^*(\bar{\hat{x}}_t, \bar{\hat{y}}_t) - y^*(\bar{\hat{x}}_{t+1}, \bar{\hat{y}}_t)\|^2] + 5C_{x^1_{y\hat{x}\hat{y}}}^2 \mathbb{E}[\|y^*(\bar{\hat{x}}_{t+1}, \bar{\hat{y}}_t) - y^*(\bar{\hat{x}}_{t+1}, \hat{y}^+(\bar{\hat{x}}_{t+1}))\|^2]$$

$$\leq 10\beta_y^2\eta^2 C_{x^1_{y\hat{x}\hat{y}}}^2 \mathbb{E}[\|\bar{v}_t - \nabla_y F(\bar{x}_t, \bar{y}_t; \bar{\hat{x}}_t, \bar{\hat{y}}_t)\|^2] + 10\beta_y^2\eta^2 L^2 C_{x^1_{y\hat{x}\hat{y}}}^2 \mathbb{E}[\|\bar{x}_t - x^*(\bar{y}_t; \bar{\hat{x}}_t, \bar{\hat{y}}_t)\|^2]$$

$$+ 5C_{x^1_{y\hat{x}\hat{y}}}^2 \left(1 + \frac{1}{\beta_y^2\eta^2(\gamma_2 - L)^2}\right) \mathbb{E}[\|y^+(\bar{\hat{x}}_t, \bar{\hat{y}}_t) - \bar{y}_t\|^2]$$

$$+ 5C_{x^1_{y\hat{x}\hat{y}}}^2 C_{y^1_{\hat{x}\hat{y}}}^2 \mathbb{E}[\|\bar{\hat{x}}_t - \bar{\hat{x}}_{t+1}\|^2] + 5C_{x^1_{y\hat{x}\hat{y}}}^2 C_{y^2_{\hat{x}\hat{y}}}^2 \mathbb{E}[\|\bar{\hat{y}}_t - \hat{y}^+(\bar{\hat{x}}_{t+1})\|^2] , \tag{46}$$

where the last step holds due to the following inequality:

$$\mathbb{E}[\|\bar{y}_{t+1} - y^+(\bar{\hat{x}}_t, \bar{\hat{y}}_t)\|^2]$$

$$= \mathbb{E}[\|\bar{y}_t + \beta_y\eta\bar{v}_t - \bar{y}_t - \beta_y\eta\nabla_y F(x^*(\bar{y}_t; \bar{\hat{x}}_t, \bar{\hat{y}}_t), \bar{y}_t; \bar{\hat{x}}_t, \bar{\hat{y}}_t)\|^2]$$

$$= \beta_y^2\eta^2 \mathbb{E}[\|\bar{v}_t - \nabla_y F(x^*(\bar{y}_t; \bar{\hat{x}}_t, \bar{\hat{y}}_t), \bar{y}_t; \bar{\hat{x}}_t, \bar{\hat{y}}_t)\|^2]$$

$$\leq 2\beta_y^2\eta^2 \mathbb{E}[\|\bar{v}_t - \nabla_y F(\bar{x}_t, \bar{y}_t; \bar{\hat{x}}_t, \bar{\hat{y}}_t)\|^2]$$

$$+ 2\beta_y^2\eta^2 \mathbb{E}[\|\nabla_y F(\bar{x}_t, \bar{y}_t; \bar{\hat{x}}_t, \bar{\hat{y}}_t) - \nabla_y F(x^*(\bar{y}_t; \bar{\hat{x}}_t, \bar{\hat{y}}_t), \bar{y}_t; \bar{\hat{x}}_t, \bar{\hat{y}}_t)\|^2]$$

$$\leq 2\beta_y^2\eta^2 \mathbb{E}[\|\bar{v}_t - \nabla_y F(\bar{x}_t, \bar{y}_t; \bar{\hat{x}}_t, \bar{\hat{y}}_t)\|^2] + 2\beta_y^2\eta^2 L^2 \mathbb{E}[\|\bar{x}_t - x^*(\bar{y}_t; \bar{\hat{x}}_t, \bar{\hat{y}}_t)\|^2] . \tag{47}$$

$\square$

**Lemma C.6.** *Given Assumptions 3.1-3.4, the following inequality holds:*

$$\mathbb{E}[\|\bar{\hat{y}}_t - \hat{y}^+(\bar{\hat{x}}_{t+1})\|^2] \leq 2\mathbb{E}[\|\bar{\hat{y}}_{t+1} - \bar{\hat{y}}_t\|^2] + 4\hat{\beta}_y^2\eta^2\beta_y^2\eta^2 \mathbb{E}[\|\bar{v}_t\|^2]$$

$$+ \frac{4\hat{\beta}_y^2}{\beta_y^2(\gamma_2 - L)^2} \mathbb{E}[\|\bar{y}_t - y^+(\bar{\hat{x}}_t, \bar{\hat{y}}_t)\|^2] . \tag{48}$$

*Proof.*

$$\frac{1}{2}\mathbb{E}[\|\bar{\hat{y}}_t - \hat{y}^+(\bar{\hat{x}}_{t+1})\|^2]$$

$$\leq \mathbb{E}[\|\bar{\hat{y}}_{t+1} - \bar{\hat{y}}_t\|^2] + \mathbb{E}[\|\bar{\hat{y}}_{t+1} - \hat{y}^+(\bar{\hat{x}}_{t+1})\|^2]$$

$$\leq \mathbb{E}[\|\bar{\hat{y}}_{t+1} - \bar{\hat{y}}_t\|^2] + \mathbb{E}[\|\bar{\hat{y}}_t + \hat{\beta}_y\eta(\bar{y}_{t+1} - \bar{\hat{y}}_t) - \bar{\hat{y}}_t - \hat{\beta}_y\eta(y^*(\bar{\hat{x}}_t, \bar{\hat{y}}_t) - \bar{\hat{y}}_t)\|^2]$$

$$= \mathbb{E}[\|\bar{\hat{y}}_{t+1} - \bar{\hat{y}}_t\|^2] + \hat{\beta}_y^2\eta^2 \mathbb{E}[\|\bar{y}_{t+1} - y^*(\bar{\hat{x}}_t, \bar{\hat{y}}_t)\|^2]$$

$$\leq \mathbb{E}[\|\bar{\hat{y}}_{t+1} - \bar{\hat{y}}_t\|^2] + 2\hat{\beta}_y^2\eta^2 \mathbb{E}[\|\bar{y}_{t+1} - \bar{y}_t\|^2] + 2\hat{\beta}_y^2\eta^2 \mathbb{E}[\|\bar{y}_t - y^*(\bar{\hat{x}}_t, \bar{\hat{y}}_t)\|^2]$$

$$\leq \mathbb{E}[\|\bar{\hat{y}}_{t+1} - \bar{\hat{y}}_t\|^2] + 2\hat{\beta}_y^2\eta^2\beta_y^2\eta^2 \mathbb{E}[\|\bar{v}_t\|^2] + \frac{2\hat{\beta}_y^2}{\beta_y^2(\gamma_2 - L)^2} \mathbb{E}[\|\bar{y}_t - y^+(\bar{\hat{x}}_t, \bar{\hat{y}}_t)\|^2] . \tag{49}$$

$\square$

**Lemma C.7.** *Given Assumptions 3.1-3.4, the following inequality holds:*

$$\mathbb{E}[\|\bar{y}_t - y^+(\bar{\hat{x}}_t, \bar{\hat{y}}_t)\|^2] \leq 4\beta_y^2\eta^2 L^2 \mathbb{E}[\|x^*(\bar{y}_t; \bar{\hat{x}}_t, \bar{\hat{y}}_t) - \bar{x}_t\|^2]$$
$$+ 4\beta_y^2\eta^2 \mathbb{E}[\|\nabla_y F(\bar{x}_t, \bar{y}_t; \bar{\hat{x}}_t, \bar{\hat{y}}_t) - \bar{v}_t\|^2] + 2\beta_y^2\eta^2 \mathbb{E}[\|\bar{v}_t\|^2] . \tag{50}$$

*Proof.*

$$\mathbb{E}[\|y^+(\bar{\hat{x}}_t, \bar{\hat{y}}_t) - \bar{y}_t\|^2]$$
$$\leq 2\mathbb{E}[\|y^+(\bar{\hat{x}}_t, \bar{\hat{y}}_t) - \bar{y}_{t+1}\|^2] + 2\mathbb{E}[\|\bar{y}_{t+1} - \bar{y}_t\|^2]$$
$$= 2\mathbb{E}[\|\bar{y}_t + \beta_y\eta\nabla_y F_d(\bar{y}_t; \bar{\hat{x}}_t, \bar{\hat{y}}_t) - \bar{y}_t - \beta_y\eta\bar{v}_t\|^2] + 2\beta_y^2\eta^2\mathbb{E}[\|\bar{v}_t\|^2]$$
$$= 2\beta_y^2\eta^2\mathbb{E}[\|\nabla_y F_d(\bar{y}_t; \bar{\hat{x}}_t, \bar{\hat{y}}_t) - \bar{v}_t\|^2] + 2\beta_y^2\eta^2\mathbb{E}[\|\bar{v}_t\|^2]$$
$$\leq 4\beta_y^2\eta^2\mathbb{E}[\|\nabla_y F_d(\bar{y}_t; \bar{\hat{x}}_t, \bar{\hat{y}}_t) - \nabla_y F(\bar{x}_t, \bar{y}_t; \bar{\hat{x}}_t, \bar{\hat{y}}_t)\|^2]$$
$$+ 4\beta_y^2\eta^2\mathbb{E}[\|\nabla_y F(\bar{x}_t, \bar{y}_t; \bar{\hat{x}}_t, \bar{\hat{y}}_t) - \bar{v}_t\|^2] + 2\beta_y^2\eta^2\mathbb{E}[\|\bar{v}_t\|^2] \tag{51}$$
$$\leq 4\beta_y^2\eta^2 L^2 \mathbb{E}[\|x^*(\bar{y}_t; \bar{\hat{x}}_t, \bar{\hat{y}}_t) - \bar{x}_t\|^2] + 4\beta_y^2\eta^2\mathbb{E}[\|\nabla_y F(\bar{x}_t, \bar{y}_t; \bar{\hat{x}}_t, \bar{\hat{y}}_t) - \bar{v}_t\|^2] + 2\beta_y^2\eta^2\mathbb{E}[\|\bar{v}_t\|^2] .$$

$\square$

**Lemma C.8.** *Given Assumptions 3.1-3.4, by defining*

$$\mathcal{P}_t = \mathbb{E}[F(\bar{x}_t, \bar{y}_t; \bar{\hat{x}}_t, \bar{\hat{y}}_t)] - 2\mathbb{E}[F_d(\bar{y}_t; \bar{\hat{x}}_t, \bar{\hat{y}}_t)] + 2\mathbb{E}[q(\bar{\hat{x}}_t)] , \tag{52}$$

*by setting $\eta \leq \frac{1}{\hat{\beta}_x}$, $\eta \leq \frac{1}{\hat{\beta}_y}$, and $\beta_x \leq \min\{\frac{L^2}{120\gamma_1^3}, \frac{\sqrt{\mu(\gamma_1-L)^3(\gamma_2-L)^2}}{512\sqrt{6\gamma_1 c_{\hat{\beta}_x}}\gamma_2 c_{\hat{\beta}_y}}\}$, then the following inequality holds:*

$$\mathcal{P}_{t+1} - \mathcal{P}_t \leq -\frac{\beta_x\eta}{4}\mathbb{E}[\|\nabla_x F(\bar{x}_t, \bar{y}_t; \bar{\hat{x}}_t, \bar{\hat{y}}_t)\|^2] - \frac{\beta_y\eta}{2}\mathbb{E}[\|\nabla_y F(\bar{x}_t, \bar{y}_t; \bar{\hat{x}}_t, \bar{\hat{y}}_t)\|^2]$$

$$+ \frac{\beta_x\eta}{2}\mathbb{E}[\|\nabla_x F(\bar{x}_t, \bar{y}_t; \bar{\hat{x}}_t, \bar{\hat{y}}_t) - \bar{u}_t\|^2] + A_3\mathbb{E}[\|\nabla_y F(\bar{x}_t, \bar{y}_t; \bar{\hat{x}}_t, \bar{\hat{y}}_t) - \bar{v}_t\|^2]$$

$$+ \left(4\beta_y\eta\beta_x^2\eta^2 L^2 - \frac{\beta_x\eta}{4}\right)\mathbb{E}[\|\bar{u}_t\|^2]$$

$$+ \left(\beta_y^2\eta^2 L_d + \frac{3\beta_y\eta}{4} + \frac{\beta_y^2\eta^2(\gamma_2+L)}{2} + 4A_1\hat{\beta}_y^2\eta^2\beta_y^2\eta^2 + 2A_2\beta_y^2\eta^2 - \frac{7}{8}\beta_y\eta\right)\mathbb{E}[\|\bar{v}_t\|^2]$$

$$+ \left(2\gamma_1 C_{x_{\hat{x}\hat{y}}^1} + \frac{\gamma_1}{6\hat{\beta}_x} + 6\gamma_1\hat{\beta}_x\eta\left(10C_{x_{y\hat{x}\hat{y}}^1}^2 C_{y_{\hat{x}\hat{y}}^1}^2 + \frac{4}{\gamma_1-L}\frac{2\gamma_2^2 C_{y_{\hat{x}\hat{y}}^1}^2}{\mu}\right) - \frac{\gamma_1(2-\hat{\beta}_x\eta)}{2\hat{\beta}_x\eta}\right)\mathbb{E}[\|\bar{\hat{x}}_{t+1} - \bar{\hat{x}}_t\|^2]$$

$$+ \left(2A_1 - \frac{\gamma_2(2-\hat{\beta}_y\eta)}{2\hat{\beta}_y\eta}\right)\mathbb{E}[\|\bar{\hat{y}}_{t+1} - \bar{\hat{y}}_t\|^2] , \tag{53}$$

*where*

$$A_1 = 6\gamma_1\hat{\beta}_x\eta\left(10C_{x_{y\hat{x}\hat{y}}^1}^2 C_{y_{\hat{x}\hat{y}}^2}^2 + \frac{4}{\gamma_1-L}\frac{2\gamma_2^2}{\mu}\left(C_{y_{\hat{x}\hat{y}}^2}^2 + \frac{(1-\hat{\beta}_y\eta)^2}{\hat{\beta}_y^2\eta^2}\right)\right) ,$$

$$A_2 = 60\gamma_1\hat{\beta}_x\eta C_{x_{y\hat{x}\hat{y}}^1}^2\left(1 + \frac{1}{\beta_y^2\eta^2(\gamma_2-L)^2}\right) + A_1\frac{4\hat{\beta}_y^2}{\beta_y^2(\gamma_2-L)^2} ,$$

$$A_3 = \beta_y\eta + 120\gamma_1\hat{\beta}_x\eta\beta_y^2\eta^2 C_{x_{y\hat{x}\hat{y}}^1}^2 + 4A_2\beta_y^2\eta^2 . \tag{54}$$

*and*

$$\beta_y = \beta_x \underbrace{\frac{(\gamma_1-L)^2}{64L^2}}_{c_{\beta_y}=O(1)} , \quad \hat{\beta}_x = \beta_x \underbrace{\frac{(\gamma_1-L)^4(\gamma_2-L)^2\mu}{24\times 64^2\gamma_1 L^2\left(5\gamma_1^2\mu + 16\gamma_2^2(\gamma_1-L)\right)}}_{c_{\hat{\beta}_x}=O(1/\kappa)} ,$$

$$\hat{\beta}_y = \beta_x \underbrace{\frac{(\gamma_1-L)^4(\gamma_2-L)^4}{64^2\times 480\gamma_1^3\gamma_2^2 L^2}}_{c_{\hat{\beta}_y}=O(1)} . \tag{55}$$

*Proof.* Based on Lemmas C.1, C.2, C.3, we have

$$
\begin{aligned}
\mathcal{P}_{t+1} - \mathcal{P}_t \leq {} & -\frac{\beta_x \eta}{2} \mathbb{E}[\|\nabla_x F(\bar{x}_t, \bar{y}_t; \bar{\hat{x}}_t, \bar{\hat{y}}_t)\|^2] + \frac{\beta_y \eta}{2} \mathbb{E}[\|\nabla_y F(\bar{x}_t, \bar{y}_t; \bar{\hat{x}}_t, \bar{\hat{y}}_t)\|^2] \\
& + \frac{\beta_x \eta}{2} \mathbb{E}[\|\nabla_x F(\bar{x}_t, \bar{y}_t; \bar{\hat{x}}_t, \bar{\hat{y}}_t) - \bar{u}_t\|^2] + \left(4\beta_y \eta \beta_x^2 \eta^2 L^2 - \frac{\beta_x \eta}{4}\right) \mathbb{E}[\|\bar{u}_t\|^2] \\
& + \left(\beta_y^2 \eta^2 L_d + \frac{3\beta_y \eta}{4} + \frac{\beta_y^2 \eta^2 (\gamma_2 + L)}{2}\right) \mathbb{E}[\|\bar{v}_t\|^2] + \left(\frac{-\gamma_1(2 - \hat{\beta}_x \eta)}{2\hat{\beta}_x \eta}\right) \mathbb{E}[\|\bar{\hat{x}}_{t+1} - \bar{\hat{x}}_t\|^2] \\
& + \left(\frac{-\gamma_2(2 - \hat{\beta}_y \eta)}{2\hat{\beta}_y \eta}\right) \mathbb{E}[\|\bar{\hat{y}}_{t+1} - \bar{\hat{y}}_t\|^2] - 2\beta_y \eta \mathbb{E}[\langle \nabla_y F_d(\bar{y}_t; \bar{\hat{x}}_t, \bar{\hat{y}}_t), \bar{v}_t \rangle] \\
& + 2\gamma_1 \mathbb{E}[\langle \bar{\hat{x}}_{t+1} - \bar{\hat{x}}_t, x^*(\bar{y}_{t+1}; \bar{\hat{x}}_{t+1}, \bar{\hat{y}}_t) - x^*(\bar{\hat{x}}_t, \hat{y}^*(\bar{\hat{x}}_{t+1}))\rangle].
\end{aligned}
\tag{56}
$$

For $-2\beta_y \eta \mathbb{E}[\langle \nabla_y F_d(\bar{y}_t; \bar{\hat{x}}_t, \bar{\hat{y}}_t), \bar{v}_t \rangle]$, we have

$$
\begin{aligned}
& -2\beta_y \eta \mathbb{E}[\langle \nabla_y F_d(\bar{y}_t; \bar{\hat{x}}_t, \bar{\hat{y}}_t), \bar{v}_t \rangle] \\
={} & -2\beta_y \eta \mathbb{E}[\langle \nabla_y F_d(\bar{y}_t; \bar{\hat{x}}_t, \bar{\hat{y}}_t) - \nabla_y F(\bar{x}_t, \bar{y}_t; \bar{\hat{x}}_t, \bar{\hat{y}}_t), \bar{v}_t \rangle] - 2\beta_y \eta \mathbb{E}[\langle \nabla_y F(\bar{x}_t, \bar{y}_t; \bar{\hat{x}}_t, \bar{\hat{y}}_t), \bar{v}_t \rangle] \\
={} & -2\beta_y \eta \mathbb{E}[\langle \nabla_y F_d(\bar{y}_t; \bar{\hat{x}}_t, \bar{\hat{y}}_t) - \nabla_y F(\bar{x}_t, \bar{y}_t; \bar{\hat{x}}_t, \bar{\hat{y}}_t), \bar{v}_t \rangle] \\
& - \beta_y \eta \mathbb{E}[\|\nabla_y F(\bar{x}_t, \bar{y}_t; \bar{\hat{x}}_t, \bar{\hat{y}}_t)\|^2] - \beta_y \eta \mathbb{E}[\|\bar{v}_t\|^2] + \beta_y \eta \mathbb{E}[\|\nabla_y F(\bar{x}_t, \bar{y}_t; \bar{\hat{x}}_t, \bar{\hat{y}}_t) - \bar{v}_t\|^2] \\
\leq{} & \beta_y \eta \frac{1}{\nu} \mathbb{E}[\|\nabla_y F_d(\bar{y}_t; \bar{\hat{x}}_t, \bar{\hat{y}}_t) - \nabla_y F(\bar{x}_t, \bar{y}_t; \bar{\hat{x}}_t, \bar{\hat{y}}_t)\|^2] + \nu \beta_y \eta \mathbb{E}[\|\bar{v}_t\|^2] \\
& - \beta_y \eta \mathbb{E}[\|\nabla_y F(\bar{x}_t, \bar{y}_t; \bar{\hat{x}}_t, \bar{\hat{y}}_t)\|^2] - \beta_y \eta \mathbb{E}[\|\bar{v}_t\|^2] + \beta_y \eta \mathbb{E}[\|\nabla_y F(\bar{x}_t, \bar{y}_t; \bar{\hat{x}}_t, \bar{\hat{y}}_t) - \bar{v}_t\|^2] \\
={} & \beta_y \eta \frac{1}{\nu} \mathbb{E}[\|\nabla_y F_d(\bar{y}_t; \bar{\hat{x}}_t, \bar{\hat{y}}_t) - \nabla_y F(\bar{x}_t, \bar{y}_t; \bar{\hat{x}}_t, \bar{\hat{y}}_t)\|^2] \\
& - \beta_y \eta \mathbb{E}[\|\nabla_y F(\bar{x}_t, \bar{y}_t; \bar{\hat{x}}_t, \bar{\hat{y}}_t)\|^2] - (1 - \nu)\beta_y \eta \mathbb{E}[\|\bar{v}_t\|^2] + \beta_y \eta \mathbb{E}[\|\nabla_y F(\bar{x}_t, \bar{y}_t; \bar{\hat{x}}_t, \bar{\hat{y}}_t) - \bar{v}_t\|^2] \\
={} & \beta_y \eta \frac{1}{\nu} \mathbb{E}[\|\nabla_y F(x^*(\bar{y}_t; \bar{\hat{x}}_t, \bar{\hat{y}}_t), \bar{y}_t; \bar{\hat{x}}_t, \bar{\hat{y}}_t) - \nabla_y F(\bar{x}_t, \bar{y}_t; \bar{\hat{x}}_t, \bar{\hat{y}}_t)\|^2] \\
& - \beta_y \eta \mathbb{E}[\|\nabla_y F(\bar{x}_t, \bar{y}_t; \bar{\hat{x}}_t, \bar{\hat{y}}_t)\|^2] - (1 - \nu)\beta_y \eta \mathbb{E}[\|\bar{v}_t\|^2] + \beta_y \eta \mathbb{E}[\|\nabla_y F(\bar{x}_t, \bar{y}_t; \bar{\hat{x}}_t, \bar{\hat{y}}_t) - \bar{v}_t\|^2] \\
\leq{} & \beta_y \eta L^2 \frac{1}{\nu} \mathbb{E}[\|x^*(\bar{y}_t; \bar{\hat{x}}_t, \bar{\hat{y}}_t) - \bar{x}_t\|^2] \\
& - \beta_y \eta \mathbb{E}[\|\nabla_y F(\bar{x}_t, \bar{y}_t; \bar{\hat{x}}_t, \bar{\hat{y}}_t)\|^2] - (1 - \nu)\beta_y \eta \mathbb{E}[\|\bar{v}_t\|^2] + \beta_y \eta \mathbb{E}[\|\nabla_y F(\bar{x}_t, \bar{y}_t; \bar{\hat{x}}_t, \bar{\hat{y}}_t) - \bar{v}_t\|^2],
\end{aligned}
\tag{57}
$$

where the third step holds due to Young's inequality $2a^T b \leq \frac{1}{\nu}\|a\|^2 + \nu\|b\|^2$ with $\nu > 0$ being a constant, and the last step holds due to the following inequality:

$$
\begin{aligned}
& \mathbb{E}[\|\nabla_y F(x^*(\bar{y}_t; \bar{\hat{x}}_t, \bar{\hat{y}}_t), \bar{y}_t; \bar{\hat{x}}_t, \bar{\hat{y}}_t) - \nabla_y F(\bar{x}_t, \bar{y}_t; \bar{\hat{x}}_t, \bar{\hat{y}}_t)\|^2] \\
={} & \mathbb{E}[\|\nabla_y f(x^*(\bar{y}_t; \bar{\hat{x}}_t, \bar{\hat{y}}_t), \bar{y}_t) - \nabla_y f(\bar{x}_t, \bar{y}_t)\|^2] \leq L^2 \mathbb{E}[\|x^*(\bar{y}_t; \bar{\hat{x}}_t, \bar{\hat{y}}_t) - \bar{x}_t\|^2].
\end{aligned}
\tag{58}
$$

For $2\gamma_1 \mathbb{E}[\langle \bar{\hat{x}}_{t+1} - \bar{\hat{x}}_t, x^*(\bar{y}_{t+1}; \bar{\hat{x}}_{t+1}, \bar{\hat{y}}_t) - x^*(\bar{\hat{x}}_t, \hat{y}^*(\bar{\hat{x}}_{t+1}))\rangle]$, we have

$$
\begin{aligned}
& 2\gamma_1 \mathbb{E}[\langle \bar{\hat{x}}_{t+1} - \bar{\hat{x}}_t, x^*(\bar{y}_{t+1}; \bar{\hat{x}}_{t+1}, \bar{\hat{y}}_t) - x^*(\bar{\hat{x}}_t, \hat{y}^*(\bar{\hat{x}}_{t+1}))\rangle] \\
={} & 2\gamma_1 \mathbb{E}[\langle \bar{\hat{x}}_{t+1} - \bar{\hat{x}}_t, x^*(\bar{y}_{t+1}; \bar{\hat{x}}_{t+1}, \bar{\hat{y}}_t) - x^*(\bar{\hat{x}}_{t+1}, \hat{y}^*(\bar{\hat{x}}_{t+1}))\rangle] \\
& + 2\gamma_1 \mathbb{E}[\langle \bar{\hat{x}}_{t+1} - \bar{\hat{x}}_t, x^*(\bar{\hat{x}}_{t+1}, \hat{y}^*(\bar{\hat{x}}_{t+1})) - x^*(\bar{\hat{x}}_t, \hat{y}^*(\bar{\hat{x}}_{t+1}))\rangle] \\
\leq{} & \frac{\gamma_1}{6\hat{\beta}_x \eta} \mathbb{E}[\|\bar{\hat{x}}_{t+1} - \bar{\hat{x}}_t\|^2] + 6\gamma_1 \hat{\beta}_x \eta \mathbb{E}[\|x^*(\bar{y}_{t+1}; \bar{\hat{x}}_{t+1}, \bar{\hat{y}}_t) - x^*(\bar{\hat{x}}_{t+1}, \hat{y}^*(\bar{\hat{x}}_{t+1}))\|^2] \\
& + 2\gamma_1 \mathbb{E}[\|\bar{\hat{x}}_{t+1} - \bar{\hat{x}}_t\| \|x^*(\bar{\hat{x}}_{t+1}, \hat{y}^*(\bar{\hat{x}}_{t+1})) - x^*(\bar{\hat{x}}_t, \hat{y}^*(\bar{\hat{x}}_{t+1}))\|] \\
\leq{} & \frac{\gamma_1}{6\hat{\beta}_x \eta} \mathbb{E}[\|\bar{\hat{x}}_{t+1} - \bar{\hat{x}}_t\|^2] + 6\gamma_1 \hat{\beta}_x \eta \mathbb{E}[\|x^*(\bar{y}_{t+1}; \bar{\hat{x}}_{t+1}, \bar{\hat{y}}_t) - x^*(\bar{\hat{x}}_{t+1}, \hat{y}^*(\bar{\hat{x}}_{t+1}))\|^2] \\
& + 2\gamma_1 C_{x_{\hat{x}\hat{y}}^1} \mathbb{E}[\|\bar{\hat{x}}_{t+1} - \bar{\hat{x}}_t\|^2],
\end{aligned}
\tag{59}
$$

where the second step holds due to Young's inequality $2a^T b \leq \frac{1}{\nu}\|a\|^2 + \nu\|b\|^2$ with $\nu = 6\hat{\beta}_x \eta$ and $a^T b \leq \|a\|\|b\|$, and the last step holds due to Lemma B.2.

Then, by plugging Eq. (57) and Eq. (59) into Eq. (56) with $\nu = \frac{1}{8}$, we have

$$
\mathcal{P}_{t+1} - \mathcal{P}_t \leq -\frac{\beta_x \eta}{2} \mathbb{E}[\|\nabla_x F(\bar{x}_t, \bar{y}_t; \bar{\hat{x}}_t, \bar{\hat{y}}_t)\|^2] - \frac{\beta_y \eta}{2} \mathbb{E}[\|\nabla_y F(\bar{x}_t, \bar{y}_t; \bar{\hat{x}}_t, \bar{\hat{y}}_t)\|^2]
$$

$$
+ \frac{\beta_x \eta}{2} \mathbb{E}[\|\nabla_x F(\bar{x}_t, \bar{y}_t; \bar{\hat{x}}_t, \bar{\hat{y}}_t) - \bar{u}_t\|^2] + \beta_y \eta \mathbb{E}[\|\nabla_y F(\bar{x}_t, \bar{y}_t; \bar{\hat{x}}_t, \bar{\hat{y}}_t) - \bar{v}_t\|^2]
$$

$$
+ \left( 4\beta_y \eta \beta_x^2 \eta^2 L^2 - \frac{\beta_x \eta}{4} \right) \mathbb{E}[\|\bar{u}_t\|^2] + \left( \beta_y^2 \eta^2 L_d + \frac{3\beta_y \eta}{4} + \frac{\beta_y^2 \eta^2 (\gamma_2 + L)}{2} - \frac{7}{8} \beta_y \eta \right) \mathbb{E}[\|\bar{v}_t\|^2]
$$

$$
+ \left( 2\gamma_1 C_{x_{\hat{x}\hat{y}}^1} + \frac{\gamma_1}{6\hat{\beta}_x \eta} - \frac{\gamma_1 (2 - \hat{\beta}_x \eta)}{2\hat{\beta}_x \eta} \right) \mathbb{E}[\|\bar{\hat{x}}_{t+1} - \bar{\hat{x}}_t\|^2] + \left( -\frac{\gamma_2 (2 - \hat{\beta}_y \eta)}{2\hat{\beta}_y \eta} \right) \mathbb{E}[\|\bar{\hat{y}}_{t+1} - \bar{\hat{y}}_t\|^2]
$$

$$
+ 8\beta_y \eta L^2 \mathbb{E}[\|x^*(\bar{y}_t; \bar{\hat{x}}_t, \bar{\hat{y}}_t) - \bar{x}_t\|^2] + 6\gamma_1 \hat{\beta}_x \eta \mathbb{E}[\|x^*(\bar{y}_{t+1}; \bar{\hat{x}}_{t+1}, \bar{\hat{y}}_t) - x^*(\bar{\hat{x}}_{t+1}, \hat{y}^*(\bar{\hat{x}}_{t+1}))\|^2] .
$$
(60)

For $\mathbb{E}[\|x^*(\bar{y}_{t+1}; \bar{\hat{x}}_{t+1}, \bar{\hat{y}}_t) - x^*(\bar{\hat{x}}_{t+1}, \hat{y}^*(\bar{\hat{x}}_{t+1}))\|^2]$, we have

$$
\mathbb{E}[\|x^*(\bar{y}_{t+1}; \bar{\hat{x}}_{t+1}, \bar{\hat{y}}_t) - x^*(\bar{\hat{x}}_{t+1}, \hat{y}^*(\bar{\hat{x}}_{t+1}))\|^2]
$$

$$
\leq 2\mathbb{E}[\|x^*(\bar{y}_{t+1}; \bar{\hat{x}}_{t+1}, \bar{\hat{y}}_t) - x^*(\bar{\hat{x}}_{t+1}, \hat{y}^+(\bar{\hat{x}}_{t+1}))\|^2]
$$

$$
+ 2\mathbb{E}[\|x^*(\bar{\hat{x}}_{t+1}, \hat{y}^+(\bar{\hat{x}}_{t+1})) - x^*(\bar{\hat{x}}_{t+1}, \hat{y}^*(\bar{\hat{x}}_{t+1}))\|^2]
$$

$$
\leq 20\beta_y^2 \eta^2 C_{x_{y\hat{x}\hat{y}}^1}^2 \mathbb{E}[\|\bar{v}_t - \nabla_y F(\bar{x}_t, \bar{y}_t; \bar{\hat{x}}_t, \bar{\hat{y}}_t)\|^2] + 20\beta_y^2 \eta^2 L^2 C_{x_{y\hat{x}\hat{y}}^1}^2 \mathbb{E}[\|\bar{x}_t - x^*(\bar{y}_t; \bar{\hat{x}}_t, \bar{\hat{y}}_t)\|^2]
$$

$$
+ 10 C_{x_{y\hat{x}\hat{y}}^1}^2 \left( 1 + \frac{1}{\beta_y^2 \eta^2 (\gamma_2 - L)^2} \right) \mathbb{E}[\|y^+(\bar{\hat{x}}_t, \bar{\hat{y}}_t) - \bar{y}_t\|^2]
$$

$$
+ 10 C_{x_{y\hat{x}\hat{y}}^1}^2 C_{y_{\hat{x}\hat{y}}^1}^2 \mathbb{E}[\|\bar{\hat{x}}_t - \bar{\hat{x}}_{t+1}\|^2] + 10 C_{x_{y\hat{x}\hat{y}}^1}^2 C_{y_{\hat{x}\hat{y}}^2}^2 \mathbb{E}[\|\bar{\hat{y}}_t - \hat{y}^+(\bar{\hat{x}}_{t+1})\|^2]
$$

$$
+ \frac{4}{\gamma_1 - L} \frac{2\gamma_2^2 C_{y_{\hat{x}\hat{y}}^1}^2}{\mu} \mathbb{E}[\|\bar{\hat{x}}_{t+1} - \bar{\hat{x}}_t\|^2] + \frac{4}{\gamma_1 - L} \frac{2\gamma_2^2}{\mu} \left( C_{y_{\hat{x}\hat{y}}^2}^2 + \frac{(1 - \hat{\beta}_y \eta)^2}{\hat{\beta}_y^2 \eta^2} \right) \mathbb{E}[\|\bar{\hat{y}}_t - \hat{y}^+(\bar{\hat{x}}_{t+1})\|^2]
$$

$$
= 20\beta_y^2 \eta^2 C_{x_{y\hat{x}\hat{y}}^1}^2 \mathbb{E}[\|\bar{v}_t - \nabla_y F(\bar{x}_t, \bar{y}_t; \bar{\hat{x}}_t, \bar{\hat{y}}_t)\|^2] + 20\beta_y^2 \eta^2 L^2 C_{x_{y\hat{x}\hat{y}}^1}^2 \mathbb{E}[\|\bar{x}_t - x^*(\bar{y}_t; \bar{\hat{x}}_t, \bar{\hat{y}}_t)\|^2]
$$

$$
+ 10 C_{x_{y\hat{x}\hat{y}}^1}^2 \left( 1 + \frac{1}{\beta_y^2 \eta^2 (\gamma_2 - L)^2} \right) \mathbb{E}[\|y^+(\bar{\hat{x}}_t, \bar{\hat{y}}_t) - \bar{y}_t\|^2]
$$

$$
+ \left( 10 C_{x_{y\hat{x}\hat{y}}^1}^2 C_{y_{\hat{x}\hat{y}}^1}^2 + \frac{4}{\gamma_1 - L} \frac{2\gamma_2^2 C_{y_{\hat{x}\hat{y}}^1}^2}{\mu} \right) \mathbb{E}[\|\bar{\hat{x}}_t - \bar{\hat{x}}_{t+1}\|^2]
$$

$$
+ \left( 10 C_{x_{y\hat{x}\hat{y}}^1}^2 C_{y_{\hat{x}\hat{y}}^2}^2 + \frac{4}{\gamma_1 - L} \frac{2\gamma_2^2}{\mu} \left( C_{y_{\hat{x}\hat{y}}^2}^2 + \frac{(1 - \hat{\beta}_y \eta)^2}{\hat{\beta}_y^2 \eta^2} \right) \right) \mathbb{E}[\|\bar{\hat{y}}_t - \hat{y}^+(\bar{\hat{x}}_{t+1})\|^2] .
$$
(61)

where the second step holds due to Lemma C.4 and Lemma C.5.

By plugging the above inequality into Eq. (60), we have

$$
\mathcal{P}_{t+1} - \mathcal{P}_t \leq -\frac{\beta_x \eta}{2} \mathbb{E}[\|\nabla_x F(\bar{x}_t, \bar{y}_t; \bar{\hat{x}}_t, \bar{\hat{y}}_t)\|^2] - \frac{\beta_y \eta}{2} \mathbb{E}[\|\nabla_y F(\bar{x}_t, \bar{y}_t; \bar{\hat{x}}_t, \bar{\hat{y}}_t)\|^2]
$$

$$
+ \frac{\beta_x \eta}{2} \mathbb{E}[\|\nabla_x F(\bar{x}_t, \bar{y}_t; \bar{\hat{x}}_t, \bar{\hat{y}}_t) - \bar{u}_t\|^2] + \left( \beta_y \eta + 120\gamma_1 \hat{\beta}_x \eta \beta_y^2 \eta^2 C_{x_{y\hat{x}\hat{y}}^1}^2 \right) \mathbb{E}[\|\nabla_y F(\bar{x}_t, \bar{y}_t; \bar{\hat{x}}_t, \bar{\hat{y}}_t) - \bar{v}_t\|^2]
$$

$$
+ \left( 4\beta_y \eta \beta_x^2 \eta^2 L^2 - \frac{\beta_x \eta}{4} \right) \mathbb{E}[\|\bar{u}_t\|^2] + \left( \beta_y^2 \eta^2 L_d + \frac{3\beta_y \eta}{4} + \frac{\beta_y^2 \eta^2 (\gamma_2 + L)}{2} - \frac{7}{8} \beta_y \eta \right) \mathbb{E}[\|\bar{v}_t\|^2]
$$

$$
+ \left( 2\gamma_1 C_{x_{\hat{x}\hat{y}}^1} + \frac{\gamma_1}{6\hat{\beta}_x \eta} + 6\gamma_1 \hat{\beta}_x \eta \left( 10 C_{x_{y\hat{x}\hat{y}}^1}^2 C_{y_{\hat{x}\hat{y}}^1}^2 + \frac{4}{\gamma_1 - L} \frac{2\gamma_2^2 C_{y_{\hat{x}\hat{y}}^1}^2}{\mu} \right) - \frac{\gamma_1 (2 - \hat{\beta}_x \eta)}{2\hat{\beta}_x \eta} \right) \mathbb{E}[\|\bar{\hat{x}}_{t+1} - \bar{\hat{x}}_t\|^2]
$$

$$
+ \left( -\frac{\gamma_2 (2 - \hat{\beta}_y \eta)}{2\hat{\beta}_y \eta} \right) \mathbb{E}[\|\bar{\hat{y}}_{t+1} - \bar{\hat{y}}_t\|^2] + \left( 8\beta_y \eta L^2 + 120\gamma_1 \hat{\beta}_x \eta \beta_y^2 \eta^2 L^2 C_{x_{y\hat{x}\hat{y}}^1}^2 \right) \mathbb{E}[\|x^*(\bar{y}_t; \bar{\hat{x}}_t, \bar{\hat{y}}_t) - \bar{x}_t\|^2]
$$

$$
+ 60\gamma_1 \hat{\beta}_x \eta C_{x_{y\hat{x}\hat{y}}^1}^2 \left( 1 + \frac{1}{\beta_y^2 \eta^2 (\gamma_2 - L)^2} \right) \mathbb{E}[\|y^+(\bar{\hat{x}}_t, \bar{\hat{y}}_t) - \bar{y}_t\|^2]
$$

$$+ 6\gamma_1\hat{\beta}_x\eta\left(10C^2_{x^1_{y\hat{x}\hat{y}}}C^2_{y^2_{\hat{x}\hat{y}}} + \frac{4}{\gamma_1-L}\frac{2\gamma_2^2}{\mu}\left(C^2_{y^2_{\hat{x}\hat{y}}} + \frac{(1-\hat{\beta}_y\eta)^2}{\hat{\beta}_y^2\eta^2}\right)\right)\mathbb{E}[\|\bar{\hat{y}}_t - \hat{y}^+(\bar{\hat{x}}_{t+1})\|^2]. \qquad (62)$$

Furthermore, based on Lemma C.6, we have

$$\mathcal{P}_{t+1} - \mathcal{P}_t \leq -\frac{\beta_x\eta}{2}\mathbb{E}[\|\nabla_x F(\bar{x}_t, \bar{y}_t; \bar{\hat{x}}_t, \bar{\hat{y}}_t)\|^2] - \frac{\beta_y\eta}{2}\mathbb{E}[\|\nabla_y F(\bar{x}_t, \bar{y}_t; \bar{\hat{x}}_t, \bar{\hat{y}}_t)\|^2]$$

$$+ \frac{\beta_x\eta}{2}\mathbb{E}[\|\nabla_x F(\bar{x}_t, \bar{y}_t; \bar{\hat{x}}_t, \bar{\hat{y}}_t) - \bar{u}_t\|^2] + \left(\beta_y\eta + 120\gamma_1\hat{\beta}_x\eta\beta_y^2\eta^2 C^2_{x^1_{y\hat{x}\hat{y}}}\right)\mathbb{E}[\|\nabla_y F(\bar{x}_t, \bar{y}_t; \bar{\hat{x}}_t, \bar{\hat{y}}_t) - \bar{v}_t\|^2]$$

$$+ \left(4\beta_y\eta\beta_x^2\eta^2 L^2 - \frac{\beta_x\eta}{4}\right)\mathbb{E}[\|\bar{u}_t\|^2]$$

$$+ \left(\beta_y^2\eta^2 L_d + \frac{3\beta_y\eta}{4} + \frac{\beta_y^2\eta^2(\gamma_2+L)}{2} + 4A_1\hat{\beta}_y^2\eta^2\beta_y^2\eta^2 - \frac{7}{8}\beta_y\eta\right)\mathbb{E}[\|\bar{v}_t\|^2]$$

$$+ \left(2\gamma_1 C_{x^1_{\hat{x}\hat{y}}} + \frac{\gamma_1}{6\hat{\beta}_x\eta} + 6\gamma_1\hat{\beta}_x\eta\left(10C^2_{x^1_{y\hat{x}\hat{y}}}C^2_{y^1_{\hat{x}\hat{y}}} + \frac{4}{\gamma_1-L}\frac{2\gamma_2^2 C^2_{y^1_{\hat{x}\hat{y}}}}{\mu}\right) - \frac{\gamma_1(2-\hat{\beta}_x\eta)}{2\hat{\beta}_x\eta}\right)\mathbb{E}[\|\bar{\hat{x}}_{t+1} - \bar{\hat{x}}_t\|^2]$$

$$+ \left(2A_1 - \frac{\gamma_2(2-\hat{\beta}_y\eta)}{2\hat{\beta}_y\eta}\right)\mathbb{E}[\|\bar{\hat{y}}_{t+1} - \bar{\hat{y}}_t\|^2]$$

$$+ \left(8\beta_y\eta L^2 + 120\gamma_1\hat{\beta}_x\eta\beta_y^2\eta^2 L^2 C^2_{x^1_{y\hat{x}\hat{y}}}\right)\mathbb{E}[\|x^*(\bar{y}_t; \bar{\hat{x}}_t, \bar{\hat{y}}_t) - \bar{x}_t\|^2]$$

$$+ \left(60\gamma_1\hat{\beta}_x\eta C^2_{x^1_{y\hat{x}\hat{y}}}\left(1 + \frac{1}{\beta_y^2\eta^2(\gamma_2-L)^2}\right) + A_1\frac{4\hat{\beta}_y^2}{\beta_y^2(\gamma_2-L)^2}\right)\mathbb{E}[\|y^+(\bar{\hat{x}}_t, \bar{\hat{y}}_t) - \bar{y}_t\|^2], \qquad (63)$$

where $A_1 = 6\gamma_1\hat{\beta}_x\eta\left(10C^2_{x^1_{y\hat{x}\hat{y}}}C^2_{y^2_{\hat{x}\hat{y}}} + \frac{4}{\gamma_1-L}\frac{2\gamma_2^2}{\mu}\left(C^2_{y^2_{\hat{x}\hat{y}}} + \frac{(1-\hat{\beta}_y\eta)^2}{\hat{\beta}_y^2\eta^2}\right)\right)$.

Moreover, based on Lemma C.7, we have

$$\mathcal{P}_{t+1} - \mathcal{P}_t \leq -\frac{\beta_x\eta}{2}\mathbb{E}[\|\nabla_x F(\bar{x}_t, \bar{y}_t; \bar{\hat{x}}_t, \bar{\hat{y}}_t)\|^2] - \frac{\beta_y\eta}{2}\mathbb{E}[\|\nabla_y F(\bar{x}_t, \bar{y}_t; \bar{\hat{x}}_t, \bar{\hat{y}}_t)\|^2]$$

$$+ \frac{\beta_x\eta}{2}\mathbb{E}[\|\nabla_x F(\bar{x}_t, \bar{y}_t; \bar{\hat{x}}_t, \bar{\hat{y}}_t) - \bar{u}_t\|^2]$$

$$+ \left(\beta_y\eta + 120\gamma_1\hat{\beta}_x\eta\beta_y^2\eta^2 C^2_{x^1_{y\hat{x}\hat{y}}} + 4A_2\beta_y^2\eta^2\right)\mathbb{E}[\|\nabla_y F(\bar{x}_t, \bar{y}_t; \bar{\hat{x}}_t, \bar{\hat{y}}_t) - \bar{v}_t\|^2]$$

$$+ \left(4\beta_y\eta\beta_x^2\eta^2 L^2 - \frac{\beta_x\eta}{4}\right)\mathbb{E}[\|\bar{u}_t\|^2]$$

$$+ \left(\beta_y^2\eta^2 L_d + \frac{3\beta_y\eta}{4} + \frac{\beta_y^2\eta^2(\gamma_2+L)}{2} + 4A_1\hat{\beta}_y^2\eta^2\beta_y^2\eta^2 + 2A_2\beta_y^2\eta^2 - \frac{7}{8}\beta_y\eta\right)\mathbb{E}[\|\bar{v}_t\|^2]$$

$$+ \left(2\gamma_1 C_{x^1_{\hat{x}\hat{y}}} + \frac{\gamma_1}{6\hat{\beta}_x\eta} + 6\gamma_1\hat{\beta}_x\eta\left(10C^2_{x^1_{y\hat{x}\hat{y}}}C^2_{y^1_{\hat{x}\hat{y}}} + \frac{4}{\gamma_1-L}\frac{2\gamma_2^2 C^2_{y^1_{\hat{x}\hat{y}}}}{\mu}\right) - \frac{\gamma_1(2-\hat{\beta}_x\eta)}{2\hat{\beta}_x\eta}\right)\mathbb{E}[\|\bar{\hat{x}}_{t+1} - \bar{\hat{x}}_t\|^2]$$

$$+ \left(2A_1 - \frac{\gamma_2(2-\hat{\beta}_y\eta)}{2\hat{\beta}_y\eta}\right)\mathbb{E}[\|\bar{\hat{y}}_{t+1} - \bar{\hat{y}}_t\|^2]$$

$$+ \left(8\beta_y\eta L^2 + 120\gamma_1\hat{\beta}_x\eta\beta_y^2\eta^2 L^2 C^2_{x^1_{y\hat{x}\hat{y}}} + 4A_2\beta_y^2\eta^2 L^2\right)\mathbb{E}[\|x^*(\bar{y}_t; \bar{\hat{x}}_t, \bar{\hat{y}}_t) - \bar{x}_t\|^2], \qquad (64)$$

where $A_2 = 60\gamma_1\hat{\beta}_x\eta C^2_{x^1_{y\hat{x}\hat{y}}}\left(1 + \frac{1}{\beta_y^2\eta^2(\gamma_2-L)^2}\right) + A_1\frac{4\hat{\beta}_y^2}{\beta_y^2(\gamma_2-L)^2}$.

Finally, based on Lemma B.5, we have

$$\mathcal{P}_{t+1} - \mathcal{P}_t \leq \left(\frac{(A_3 + 7\beta_y\eta)L^2}{(\gamma_1-L)^2} - \frac{\beta_x\eta}{2}\right)\mathbb{E}[\|\nabla_x F(\bar{x}_t, \bar{y}_t; \bar{\hat{x}}_t, \bar{\hat{y}}_t)\|^2] - \frac{\beta_y\eta}{2}\mathbb{E}[\|\nabla_y F(\bar{x}_t, \bar{y}_t; \bar{\hat{x}}_t, \bar{\hat{y}}_t)\|^2]$$

$$+ \frac{\beta_x\eta}{2}\mathbb{E}[\|\nabla_x F(\bar{x}_t, \bar{y}_t; \bar{\hat{x}}_t, \bar{\hat{y}}_t) - \bar{u}_t\|^2] + A_3\mathbb{E}[\|\nabla_y F(\bar{x}_t, \bar{y}_t; \bar{\hat{x}}_t, \bar{\hat{y}}_t) - \bar{v}_t\|^2]$$

$$
+ \left( 4\beta_y \eta \beta_x^2 \eta^2 L^2 - \frac{\beta_x \eta}{4} \right) \mathbb{E}[\|\bar{u}_t\|^2]
$$

$$
+ \left( \beta_y^2 \eta^2 L_d + \frac{3\beta_y \eta}{4} + \frac{\beta_y^2 \eta^2 (\gamma_2 + L)}{2} + 4 A_1 \hat{\beta}_y^2 \eta^2 \beta_y^2 \eta^2 + 2 A_2 \beta_y^2 \eta^2 - \frac{7}{8} \beta_y \eta \right) \mathbb{E}[\|\bar{v}_t\|^2]
$$

$$
+ \left( 2\gamma_1 C_{x_{\hat{x}\hat{y}}^1} + \frac{\gamma_1}{6\hat{\beta}_x \eta} + 6\gamma_1 \hat{\beta}_x \eta \left( 10 C_{x_{y\hat{x}\hat{y}}^1}^2 C_{y_{\hat{x}\hat{y}}^1}^2 + \frac{4}{\gamma_1 - L} \frac{2\gamma_2^2 C_{y_{\hat{x}\hat{y}}^1}^2}{\mu} \right) - \frac{\gamma_1 (2 - \hat{\beta}_x \eta)}{2\hat{\beta}_x \eta} \right) \mathbb{E}[\|\bar{\hat{x}}_{t+1} - \bar{\hat{x}}_t\|^2]
$$

$$
+ \left( 2 A_1 - \frac{\gamma_2 (2 - \hat{\beta}_y \eta)}{2\hat{\beta}_y \eta} \right) \mathbb{E}[\|\bar{\hat{y}}_{t+1} - \bar{\hat{y}}_t\|^2] , \tag{65}
$$

where $A_3 = \beta_y \eta + 120 \gamma_1 \hat{\beta}_x \eta \beta_y^2 \eta^2 C_{x_{y\hat{x}\hat{y}}^1}^2 + 4 A_2 \beta_y^2 \eta^2$.

Then, for $\mathbb{E}[\|\nabla_x F(\bar{x}_t, \bar{y}_t; \bar{\hat{x}}_t, \bar{\hat{y}}_t)\|^2]$, we set

$$
\frac{(A_3 + 7\beta_y \eta) L^2}{(\gamma_1 - L)^2} - \frac{\beta_x \eta}{2}
$$

$$
= \frac{1}{(\gamma_1 - L)^2} \left( 8\beta_y \eta L^2 + 120 \gamma_1 \hat{\beta}_x \eta \beta_y^2 \eta^2 L^2 C_{x_{y\hat{x}\hat{y}}^1}^2 + 4 A_2 \beta_y^2 \eta^2 L^2 \right) - \frac{\beta_x \eta}{2} \le -\frac{\beta_x \eta}{4} . \tag{66}
$$

Specifically, we enforce

$$
\frac{8\beta_y \eta L^2}{(\gamma_1 - L)^2} \le \frac{\beta_x \eta}{8} ,
$$

$$
\frac{120 \gamma_1 \hat{\beta}_x \eta \beta_y^2 \eta^2 L^2 C_{x_{y\hat{x}\hat{y}}^1}^2}{(\gamma_1 - L)^2} \le \frac{\beta_x \eta}{32 \times 16} ,
$$

$$
\frac{4\beta_y^2 \eta^2 L^2}{(\gamma_1 - L)^2} A_2 \le \frac{\beta_x \eta}{32 \times 16} . \tag{67}
$$

For the first inequality in Eq. (67), we set

$$
\beta_y = \beta_x \underbrace{\frac{(\gamma_1 - L)^2}{64 L^2}}_{c_{\beta_y} = O(1)} . \tag{68}
$$

For the last inequality in Eq. (67), from the definition of $A_1$ and $A_2$, we enforce

$$
\frac{4\beta_y^2 \eta^2 L^2}{(\gamma_1 - L)^2} 60 \gamma_1 \hat{\beta}_x \eta C_{x_{y\hat{x}\hat{y}}^1}^2 \le \frac{\beta_x \eta}{32 \times 64} ,
$$

$$
\frac{4 L^2}{(\gamma_1 - L)^2} \left( \frac{60 \gamma_1 \hat{\beta}_x \eta C_{x_{y\hat{x}\hat{y}}^1}^2}{(\gamma_2 - L)^2} + \frac{4 \hat{\beta}_y^2 \eta^2}{(\gamma_2 - L)^2} \frac{24 \gamma_1 \hat{\beta}_x \eta}{\gamma_1 - L} \frac{2\gamma_2^2}{\mu} \frac{(1 - \hat{\beta}_y \eta)^2}{\hat{\beta}_y^2 \eta^2} \right) \le \frac{\beta_x \eta}{32 \times 64} ,
$$

$$
\frac{4 L^2}{(\gamma_1 - L)^2} 60 \gamma_1 \hat{\beta}_x \eta C_{x_{y\hat{x}\hat{y}}^1}^2 C_{y_{\hat{x}\hat{y}}^2}^2 \frac{4 \hat{\beta}_y^2 \eta^2}{(\gamma_2 - L)^2} \le \frac{\beta_x \eta}{32 \times 64} ,
$$

$$
\frac{4 L^2}{(\gamma_1 - L)^2} 6 \gamma_1 \hat{\beta}_x \eta \frac{4}{\gamma_1 - L} \frac{2\gamma_2^2}{\mu} C_{y_{\hat{x}\hat{y}}^2}^2 \frac{4 \hat{\beta}_y^2 \eta^2}{(\gamma_2 - L)^2} \le \frac{\beta_x \eta}{32 \times 64} . \tag{69}
$$

To solve the first inequality in Eq. (69), since $\hat{\beta}_x \eta \le 1$ and $\eta < 1$, from $C_{x_{y\hat{x}\hat{y}}^1} = \frac{\gamma_1}{\gamma_1 - L}$, we obtain

$$
\beta_x \le \frac{L^2}{120 \gamma_1^3} . \tag{70}
$$

Here, we have also shown that the second inequality in Eq. (67) holds.

Then, to address the second inequality in Eq. (69), note that since $\hat{\beta}_y \eta \leq 1$, it follows that $1 - \hat{\beta}_y \eta \leq 1$. Consequently, we obtain

$$\hat{\beta}_x = \beta_x \underbrace{\frac{(\gamma_1 - L)^4 (\gamma_2 - L)^2 \mu}{24 \times 64^2 \gamma_1 L^2 \left(5\gamma_1^2 \mu + 16\gamma_2^2(\gamma_1 - L)\right)}}_{c_{\hat{\beta}_x} = O(1/\kappa)} . \tag{71}$$

Similarly, for the third inequality in Eq. (69), from $C_{y_{\hat{x}\hat{y}}^2} = \frac{\gamma_2}{\gamma_2 - L}$, we obtain

$$\hat{\beta}_y = \beta_x \underbrace{\frac{(\gamma_1 - L)^4 (\gamma_2 - L)^4}{64^2 \times 480 \gamma_1^3 \gamma_2^2 L^2}}_{c_{\hat{\beta}_y} = O(1)} . \tag{72}$$

Moreover, to solve the last inequality in Eq. (69), we obtain

$$\beta_x \leq \frac{\sqrt{\mu(\gamma_1 - L)^3}(\gamma_2 - L)^2}{512\sqrt{6\gamma_1 c_{\hat{\beta}_x}} \gamma_2 c_{\hat{\beta}_y}} = O(1) . \tag{73}$$

Finally, by plugging Eq. (66) into Eq. (65), the proof is complete. $\qquad\square$

# D    KEY LEMMAS RELATED TO THE DECENTRALIZED SETTING

## D.1    CONSENSUS ERRORS

**Lemma D.1.** *Given Assumptions 3.1-3.4, the following inequality holds:*

$$\frac{1}{K}\sum_{k=1}^K \mathbb{E}[\|\bar{p}_{t+1} - p_{t+1}^{(k)}\|^2]$$

$$\leq \lambda \frac{1}{K}\sum_{k=1}^K \mathbb{E}[\|\bar{p}_t - p_t^{(k)}\|^2] + 3\rho_x^2 \eta^4 \frac{1}{1-\lambda} \frac{1}{K}\sum_{k=1}^K \mathbb{E}[\|u_t^{(k)} - \nabla_x F^{(k)}(x_t^{(k)}, y_t^{(k)}; \hat{x}_t^{(k)}, \hat{y}_t^{(k)})\|^2]$$

$$+ \frac{9(L^2 + \gamma_1^2)}{1-\lambda} \frac{1}{K}\sum_{k=1}^K \mathbb{E}[\|x_{t+1}^{(k)} - x_t^{(k)}\|^2] + \frac{9L^2}{1-\lambda} \frac{1}{K}\sum_{k=1}^K \mathbb{E}[\|y_{t+1}^{(k)} - y_t^{(k)}\|^2]$$

$$+ \frac{9\gamma_1^2}{1-\lambda} \frac{1}{K}\sum_{k=1}^K \mathbb{E}[\|\hat{x}_{t+1}^{(k)} - \hat{x}_t^{(k)}\|^2] + 3\rho_x^2 \eta^4 \sigma^2 \frac{1}{1-\lambda} . \tag{74}$$

*Proof.*

$$\frac{1}{K}\sum_{k=1}^K \mathbb{E}[\|\bar{p}_{t+1} - p_{t+1}^{(k)}\|^2]$$

$$= \frac{1}{K}\mathbb{E}[\|\bar{P}_{t+1} - P_{t+1}\|_F^2]$$

$$= \frac{1}{K}\mathbb{E}[\|\bar{P}_t - \bar{U}_t + \bar{U}_{t+1} - P_t W + U_t - U_{t+1}\|_F^2]$$

$$\leq (1+a)\frac{1}{K}\mathbb{E}[\|\bar{P}_t - P_t W\|_F^2] + (1+1/a)\frac{1}{K}\mathbb{E}[\| - \bar{U}_t + \bar{U}_{t+1} + U_t - U_{t+1}\|_F^2]$$

$$\leq (1+a)\lambda^2 \frac{1}{K}\mathbb{E}[\|\bar{P}_t - P_t\|_F^2] + (1+1/a)\frac{1}{K}\mathbb{E}[\|U_t - U_{t+1}\|_F^2]$$

$$\leq \lambda \frac{1}{K}\mathbb{E}[\|\bar{P}_t - P_t\|_F^2] + \frac{1}{1-\lambda}\frac{1}{K}\mathbb{E}[\|U_t - U_{t+1}\|_F^2] , \tag{75}$$

where $a = \frac{1-\lambda}{\lambda}$. Then, we have the following inequality to complete the proof:

$$\frac{1}{K}\mathbb{E}[\|U_t - U_{t+1}\|_F^2]$$

$$= \frac{1}{K}\sum_{k=1}^{K}\mathbb{E}[\|u_{t+1}^{(k)} - u_t^{(k)}\|^2]$$

$$= \frac{1}{K}\sum_{k=1}^{K}\mathbb{E}[\|(1 - \rho_x\eta^2)(u_t^{(k)} - \nabla_x F^{(k)}(x_t^{(k)}, y_t^{(k)}; \hat{x}_t^{(k)}, \hat{y}_t^{(k)}; \xi_{t+1}^{(k)}))$$

$$+ \nabla_x F^{(k)}(x_{t+1}^{(k)}, y_{t+1}^{(k)}; \hat{x}_{t+1}^{(k)}, \hat{y}_{t+1}^{(k)}; \xi_{t+1}^{(k)}) - u_t^{(k)}\|^2]$$

$$\leq 3\frac{1}{K}\sum_{k=1}^{K}\mathbb{E}[\| - \rho_x\eta^2 u_t^{(k)} + \rho_x\eta^2\nabla_x F^{(k)}(x_t^{(k)}, y_t^{(k)}; \hat{x}_t^{(k)}, \hat{y}_t^{(k)})\|^2]$$

$$+ 3\frac{1}{K}\sum_{k=1}^{K}\mathbb{E}[\| - \rho_x\eta^2\nabla_x F^{(k)}(x_t^{(k)}, y_t^{(k)}; \hat{x}_t^{(k)}, \hat{y}_t^{(k)}) + \rho_x\eta^2\nabla_x F^{(k)}(x_t^{(k)}, y_t^{(k)}; \hat{x}_t^{(k)}, \hat{y}_t^{(k)}; \xi_{t+1}^{(k)})\|^2]$$

$$+ 3\frac{1}{K}\sum_{k=1}^{K}\mathbb{E}[\| - \nabla_x F^{(k)}(x_t^{(k)}, y_t^{(k)}; \hat{x}_t^{(k)}, \hat{y}_t^{(k)}; \xi_{t+1}^{(k)}) + \nabla_x F^{(k)}(x_{t+1}^{(k)}, y_{t+1}^{(k)}; \hat{x}_{t+1}^{(k)}, \hat{y}_{t+1}^{(k)}; \xi_{t+1}^{(k)})\|^2]$$

$$\leq 3\rho_x^2\eta^4\frac{1}{K}\sum_{k=1}^{K}\mathbb{E}[\|u_t^{(k)} - \nabla_x F^{(k)}(x_t^{(k)}, y_t^{(k)}; \hat{x}_t^{(k)}, \hat{y}_t^{(k)})\|^2] + 3\rho_x^2\eta^4\sigma^2$$

$$+ 3\frac{1}{K}\sum_{k=1}^{K}\mathbb{E}[\| - \nabla_x F^{(k)}(x_t^{(k)}, y_t^{(k)}; \hat{x}_t^{(k)}, \hat{y}_t^{(k)}; \xi_{t+1}^{(k)}) + \nabla_x F^{(k)}(x_{t+1}^{(k)}, y_{t+1}^{(k)}; \hat{x}_{t+1}^{(k)}, \hat{y}_{t+1}^{(k)}; \xi_{t+1}^{(k)})\|^2]$$

$$\leq 3\rho_x^2\eta^4\frac{1}{K}\sum_{k=1}^{K}\mathbb{E}[\|u_t^{(k)} - \nabla_x F^{(k)}(x_t^{(k)}, y_t^{(k)}; \hat{x}_t^{(k)}, \hat{y}_t^{(k)})\|^2] + 3\rho_x^2\eta^4\sigma^2$$

$$+ 9(L^2 + \gamma_1^2)\frac{1}{K}\sum_{k=1}^{K}\mathbb{E}[\|x_{t+1}^{(k)} - x_t^{(k)}\|^2] + 9L^2\frac{1}{K}\sum_{k=1}^{K}\mathbb{E}[\|y_{t+1}^{(k)} - y_t^{(k)}\|^2]$$

$$+ 9\gamma_1^2\frac{1}{K}\sum_{k=1}^{K}\mathbb{E}[\|\hat{x}_{t+1}^{(k)} - \hat{x}_t^{(k)}\|^2], \tag{76}$$

where the last step holds due to the following inequality:

$$\mathbb{E}[\| - \nabla_x F^{(k)}(x_t^{(k)}, y_t^{(k)}; \hat{x}_t^{(k)}, \hat{y}_t^{(k)}; \xi_{t+1}^{(k)}) + \nabla_x F^{(k)}(x_{t+1}^{(k)}, y_{t+1}^{(k)}; \hat{x}_{t+1}^{(k)}, \hat{y}_{t+1}^{(k)}; \xi_{t+1}^{(k)})\|^2]$$

$$= \mathbb{E}[\| - \nabla_x f^{(k)}(x_t^{(k)}, y_t^{(k)}; \xi_{t+1}^{(k)}) - \gamma_1(x_t^{(k)} - \hat{x}_t^{(k)})$$

$$+ \nabla_x f^{(k)}(x_{t+1}^{(k)}, y_{t+1}^{(k)}; \xi_{t+1}^{(k)}) + \gamma_1(x_{t+1}^{(k)} - \hat{x}_{t+1}^{(k)})\|^2]$$

$$\leq 3\mathbb{E}[\|\nabla_x f^{(k)}(x_{t+1}^{(k)}, y_{t+1}^{(k)}; \xi_{t+1}^{(k)}) - \nabla_x f^{(k)}(x_t^{(k)}, y_t^{(k)}; \xi_{t+1}^{(k)})\|^2]$$

$$+ 3\gamma_1^2\mathbb{E}[\|x_{t+1}^{(k)} - x_t^{(k)}\|^2] + 3\gamma_1^2\mathbb{E}[\|\hat{x}_{t+1}^{(k)} - \hat{x}_t^{(k)}\|^2]$$

$$\leq 3(L^2 + \gamma_1^2)\mathbb{E}[\|x_{t+1}^{(k)} - x_t^{(k)}\|^2] + 3L^2\mathbb{E}[\|y_{t+1}^{(k)} - y_t^{(k)}\|^2] + 3\gamma_1^2\mathbb{E}[\|\hat{x}_{t+1}^{(k)} - \hat{x}_t^{(k)}\|^2]. \tag{77}$$

$\square$

**Lemma D.2.** *Given Assumptions 3.1-3.4, the following inequality holds:*

$$\frac{1}{K}\sum_{k=1}^{K}\mathbb{E}[\|\bar{q}_{t+1} - q_{t+1}^{(k)}\|^2]$$

$$\leq \lambda\frac{1}{K}\sum_{k=1}^{K}\mathbb{E}[\|\bar{q}_t - q_t^{(k)}\|^2] + 3\rho_y^2\eta^4\frac{1}{1-\lambda}\frac{1}{K}\sum_{k=1}^{K}\mathbb{E}[\|v_t^{(k)} - \nabla_y F^{(k)}(x_t^{(k)}, y_t^{(k)}; \hat{x}_t^{(k)}, \hat{y}_t^{(k)})\|^2]$$

$$+ \frac{9L^2}{1-\lambda} \frac{1}{K} \sum_{k=1}^{K} \mathbb{E}[\|x_{t+1}^{(k)} - x_t^{(k)}\|^2] + \frac{9(L^2 + \gamma_2^2)}{1-\lambda} \frac{1}{K} \sum_{k=1}^{K} \mathbb{E}[\|y_{t+1}^{(k)} - y_t^{(k)}\|^2]$$

$$+ \frac{9\gamma_2^2}{1-\lambda} \frac{1}{K} \sum_{k=1}^{K} \mathbb{E}[\|\hat{y}_{t+1}^{(k)} - \hat{y}_t^{(k)}\|^2] + 3\rho_y^2 \eta^4 \sigma^2 \frac{1}{1-\lambda} . \tag{78}$$

This lemma can be proved by following Lemma D.1. Thus, we omit its proof.

**Lemma D.3.** *Given Assumptions 3.1-3.4, when $\hat{\beta}_x \leq \frac{1-\lambda}{4}$, the following inequality holds:*

$$\mathbb{E}[\|\hat{X}_{t+1} - \bar{\hat{X}}_{t+1}\|_F^2] \leq \left(1 - \frac{\eta(1-\lambda^2)}{4}\right) \frac{1}{K} \sum_{k=1}^{K} \mathbb{E}[\|\bar{\hat{x}}_t - \hat{x}_t^{(k)}\|^2]$$

$$+ \frac{4\eta\hat{\beta}_x^2}{1-\lambda^2} \frac{1}{K} \sum_{k=1}^{K} \mathbb{E}[\|\bar{x}_t - x_t^{(k)}\|^2] + \frac{4\eta\hat{\beta}_x^2}{1-\lambda^2} \frac{2\eta\beta_x^2}{1-\lambda^2} \frac{1}{K} \sum_{k=1}^{K} \mathbb{E}[\|\bar{p}_t - p_t^{(k)}\|^2] . \tag{79}$$

*Proof.*

$$\|\hat{X}_{t+1} - \bar{\hat{X}}_{t+1}\|_F^2$$

$$= \|\hat{X}_t + \eta(\tilde{\hat{X}}_{t+1} - \hat{X}_t) - \bar{\hat{X}}_t - \eta\hat{\beta}_x(\bar{X}_{t+1} - \bar{\hat{X}}_t)\|_F^2$$

$$= \|\hat{X}_t + \eta(\hat{X}_t W + \hat{\beta}_x(X_{t+1} - \hat{X}_t) - \hat{X}_t) - \bar{\hat{X}}_t - \eta\hat{\beta}_x(\bar{X}_{t+1} - \bar{\hat{X}}_t)\|_F^2$$

$$= \|(1-\eta)(\hat{X}_t - \bar{\hat{X}}_t) + \eta(\hat{X}_t W - \bar{\hat{X}}_t) + \eta\hat{\beta}_x(X_{t+1} - \hat{X}_t) - \eta\hat{\beta}_x(\bar{X}_{t+1} - \bar{\hat{X}}_t)\|_F^2$$

$$\leq (1-\eta)\|\hat{X}_t - \bar{\hat{X}}_t\|_F^2 + \eta\|\hat{X}_t W - \bar{\hat{X}}_t + \hat{\beta}_x(X_{t+1} - \hat{X}_t) - \hat{\beta}_x(\bar{X}_{t+1} - \bar{\hat{X}}_t)\|_F^2$$

$$\leq (1-\eta)\|\hat{X}_t - \bar{\hat{X}}_t\|_F^2 + (1+c)\eta\|\hat{X}_t W - \bar{\hat{X}}_t\|_F^2 + (1+1/c)\eta\hat{\beta}_x^2\|(X_{t+1} - \hat{X}_t) - (\bar{X}_{t+1} - \bar{\hat{X}}_t)\|_F^2$$

$$\leq (1-\eta)\|\hat{X}_t - \bar{\hat{X}}_t\|_F^2 + (1+c)\eta\lambda^2\|\hat{X}_t - \bar{\hat{X}}_t\|_F^2 + 2(1+1/c)\eta\hat{\beta}_x^2\|X_{t+1} - \bar{X}_{t+1}\|_F^2$$

$$+ 2(1+1/c)\eta\hat{\beta}_x^2\|\hat{X}_t - \bar{\hat{X}}_t\|_F^2$$

$$\leq \left(1 - \frac{\eta(1-\lambda^2)}{4}\right)\|\hat{X}_t - \bar{\hat{X}}_t\|_F^2 + \frac{4\eta\hat{\beta}_x^2}{1-\lambda^2}\|X_{t+1} - \bar{X}_{t+1}\|_F^2$$

$$\leq \left(1 - \frac{\eta(1-\lambda^2)}{4}\right) \frac{1}{K} \sum_{k=1}^{K} \mathbb{E}[\|\bar{\hat{x}}_t - \hat{x}_t^{(k)}\|^2]$$

$$+ \frac{4\eta\hat{\beta}_x^2}{1-\lambda^2} \frac{1}{K} \sum_{k=1}^{K} \mathbb{E}[\|\bar{x}_t - x_t^{(k)}\|^2] + \frac{4\eta\hat{\beta}_x^2}{1-\lambda^2} \frac{2\eta\beta_x^2}{1-\lambda^2} \frac{1}{K} \sum_{k=1}^{K} \mathbb{E}[\|\bar{p}_t - p_t^{(k)}\|^2] , \tag{80}$$

where $c = \frac{1-\lambda^2}{2\lambda^2}$ the second to last inequality holds due to $\hat{\beta}_x \leq \frac{1-\lambda}{4}$, and the last step holds due to Lemma D.5. $\qquad\square$

**Lemma D.4.** *Given Assumptions 3.1-3.4, when $\beta_{\hat{y}} \leq \frac{1-\lambda}{4}$, the following inequality holds:*

$$\mathbb{E}[\|\hat{Y}_{t+1} - \bar{\hat{Y}}_{t+1}\|_F^2] \leq \left(1 - \frac{\eta(1-\lambda^2)}{4}\right) \frac{1}{K} \sum_{k=1}^{K} \mathbb{E}[\|\bar{\hat{y}}_t - \hat{y}_t^{(k)}\|^2]$$

$$+ \frac{4\eta\hat{\beta}_y^2}{1-\lambda^2} \frac{1}{K} \sum_{k=1}^{K} \mathbb{E}[\|\bar{y}_t - y_t^{(k)}\|^2] + \frac{4\eta\hat{\beta}_y^2}{1-\lambda^2} \frac{2\eta\beta_y^2}{1-\lambda^2} \frac{1}{K} \sum_{k=1}^{K} \mathbb{E}[\|\bar{q}_t - q_t^{(k)}\|^2] . \tag{81}$$

This lemma be proved by following Lemma D.3. Thus, we omit its proof.

**Lemma D.5.** *Given Assumptions 3.1-3.4, the following inequality holds:*

$$\frac{1}{K} \sum_{k=1}^{K} \mathbb{E}[\|\bar{x}_{t+1} - x_{t+1}^{(k)}\|^2] = \frac{1}{K} \mathbb{E}[\|\bar{X}_{t+1} - X_{t+1}^{(k)}\|_F^2]$$

$$\leq \left(1 - \frac{\eta(1 - \lambda^2)}{2}\right) \frac{1}{K} \sum_{k=1}^{K} \mathbb{E}[\|\bar{x}_t - x_t^{(k)}\|^2] + \frac{2\eta\beta_x^2}{1 - \lambda^2} \frac{1}{K} \sum_{k=1}^{K} \mathbb{E}[\|\bar{p}_t - p_t^{(k)}\|^2]. \quad (82)$$

This lemma can be proved by following Lemma D.3. Thus, we omit its proof.

**Lemma D.6.** *Given Assumptions 3.1-3.4, the following inequality holds:*

$$\frac{1}{K} \sum_{k=1}^{K} \mathbb{E}[\|\bar{y}_{t+1} - y_{t+1}^{(k)}\|^2] \quad (83)$$

$$\leq \left(1 - \frac{\eta(1 - \lambda^2)}{2}\right) \frac{1}{K} \sum_{k=1}^{K} \mathbb{E}[\|\bar{y}_t - y_t^{(k)}\|^2] + \frac{2\eta\beta_y^2}{1 - \lambda^2} \frac{1}{K} \sum_{k=1}^{K} \mathbb{E}[\|\bar{q}_t - q_t^{(k)}\|^2]. \quad (84)$$

This lemma can be proved by following Lemma D.3. Thus, we omit its proof.

### D.2 GRADIENT ESTIMATION ERRORS

**Lemma D.7.** *Given Assumptions 3.1-3.4, when $\eta \leq \frac{1}{\sqrt{\rho_x}}$, the following inequality holds:*

$$\mathbb{E}[\|\frac{1}{K} \sum_{k=1}^{K} u_{t+1}^{(k)} - \frac{1}{K} \sum_{k=1}^{K} \nabla_x F^{(k)}(x_{t+1}^{(k)}, y_{t+1}^{(k)}; \hat{x}_{t+1}^{(k)}, \hat{y}_{t+1}^{(k)})\|^2]$$

$$\leq (1 - \rho_x\eta^2)\mathbb{E}[\|\frac{1}{K} \sum_{k=1}^{K} u_t^{(k)} - \frac{1}{K} \sum_{k=1}^{K} \nabla_x F^{(k)}(x_t^{(k)}, y_t^{(k)}; \hat{x}_t^{(k)}, \hat{y}_t^{(k)})\|^2] + 2\rho_x^2\eta^4\sigma^2\frac{1}{K}$$

$$+ 4L^2\frac{1}{K^2} \sum_{k=1}^{K} \mathbb{E}[\|x_{t+1}^{(k)} - x_t^{(k)}\|^2] + 4L^2\frac{1}{K^2} \sum_{k=1}^{K} \mathbb{E}[\|y_{t+1}^{(k)} - y_t^{(k)}\|^2]. \quad (85)$$

*Proof.*

$$\mathbb{E}[\|\frac{1}{K} \sum_{k=1}^{K} u_{t+1}^{(k)} - \frac{1}{K} \sum_{k=1}^{K} \nabla_x F^{(k)}(x_{t+1}^{(k)}, y_{t+1}^{(k)}; \hat{x}_{t+1}^{(k)}, \hat{y}_{t+1}^{(k)})\|^2]$$

$$= \mathbb{E}[\|(1 - \rho_x\eta^2)(\frac{1}{K} \sum_{k=1}^{K} u_t^{(k)} - \frac{1}{K} \sum_{k=1}^{K} \nabla_x F^{(k)}(x_t^{(k)}, y_t^{(k)}; \hat{x}_t^{(k)}, \hat{y}_t^{(k)}))\|^2]$$

$$+ \mathbb{E}[\|(1 - \rho_x\eta^2)\frac{1}{K} \sum_{k=1}^{K} (\nabla_x F^{(k)}(x_t^{(k)}, y_t^{(k)}; \hat{x}_t^{(k)}, \hat{y}_t^{(k)}) - \nabla_x F^{(k)}(x_t^{(k)}, y_t^{(k)}; \hat{x}_t^{(k)}, \hat{y}_t^{(k)}; \xi_{t+1}^{(k)})$$

$$+ \nabla_x F^{(k)}(x_{t+1}^{(k)}, y_{t+1}^{(k)}; \hat{x}_{t+1}^{(k)}, \hat{y}_{t+1}^{(k)}; \xi_{t+1}^{(k)}) - \nabla_x F^{(k)}(x_{t+1}^{(k)}, y_{t+1}^{(k)}; \hat{x}_{t+1}^{(k)}, \hat{y}_{t+1}^{(k)}))$$

$$+ \rho_x\eta^2\frac{1}{K} \sum_{k=1}^{K} (\nabla_x F^{(k)}(x_{t+1}^{(k)}, y_{t+1}^{(k)}; \hat{x}_{t+1}^{(k)}, \hat{y}_{t+1}^{(k)}; \xi_{t+1}^{(k)}) - \nabla_x F^{(k)}(x_{t+1}^{(k)}, y_{t+1}^{(k)}; \hat{x}_{t+1}^{(k)}, \hat{y}_{t+1}^{(k)}))\|^2]$$

$$\leq \mathbb{E}[\|(1 - \rho_x\eta^2)(\frac{1}{K} \sum_{k=1}^{K} u_t^{(k)} - \frac{1}{K} \sum_{k=1}^{K} \nabla_x F^{(k)}(x_t^{(k)}, y_t^{(k)}; \hat{x}_t^{(k)}, \hat{y}_t^{(k)}))\|^2]$$

$$+ 2(1 - \rho_x\eta^2)^2\frac{1}{K^2} \sum_{k=1}^{K} \mathbb{E}[\|\nabla_x F^{(k)}(x_t^{(k)}, y_t^{(k)}; \hat{x}_t^{(k)}, \hat{y}_t^{(k)}) - \nabla_x F^{(k)}(x_t^{(k)}, y_t^{(k)}; \hat{x}_t^{(k)}, \hat{y}_t^{(k)}; \xi_{t+1}^{(k)})$$

$$+ \nabla_x F^{(k)}(x_{t+1}^{(k)}, y_{t+1}^{(k)}; \hat{x}_{t+1}^{(k)}, \hat{y}_{t+1}^{(k)}; \xi_{t+1}^{(k)}) - \nabla_x F^{(k)}(x_{t+1}^{(k)}, y_{t+1}^{(k)}; \hat{x}_{t+1}^{(k)}, \hat{y}_{t+1}^{(k)})\|^2]$$

$$+ 2\rho_x^2\eta^4\frac{1}{K^2} \sum_{k=1}^{K} \mathbb{E}[\|\nabla_x F^{(k)}(x_{t+1}^{(k)}, y_{t+1}^{(k)}; \hat{x}_{t+1}^{(k)}, \hat{y}_{t+1}^{(k)}; \xi_{t+1}^{(k)}) - \nabla_x F^{(k)}(x_{t+1}^{(k)}, y_{t+1}^{(k)}; \hat{x}_{t+1}^{(k)}, \hat{y}_{t+1}^{(k)})\|^2]$$

$$\leq (1 - \rho_x\eta^2)\mathbb{E}[\|\frac{1}{K} \sum_{k=1}^{K} u_t^{(k)} - \frac{1}{K} \sum_{k=1}^{K} \nabla_x F^{(k)}(x_t^{(k)}, y_t^{(k)}; \hat{x}_t^{(k)}, \hat{y}_t^{(k)})\|^2] + 2\rho_x^2\eta^4\sigma^2\frac{1}{K}$$

$$+ 2\frac{1}{K^2} \sum_{k=1}^{K} \mathbb{E}[\|\nabla_x F^{(k)}(x_{t+1}^{(k)}, y_{t+1}^{(k)}; \hat{x}_{t+1}^{(k)}, \hat{y}_{t+1}^{(k)}; \xi_{t+1}^{(k)}) - \nabla_x F^{(k)}(x_t^{(k)}, y_t^{(k)}; \hat{x}_t^{(k)}, \hat{y}_t^{(k)}; \xi_{t+1}^{(k)})\|^2]$$

$$\leq (1 - \rho_x \eta^2)\mathbb{E}[\|\frac{1}{K}\sum_{k=1}^{K} u_t^{(k)} - \frac{1}{K}\sum_{k=1}^{K}\nabla_x F^{(k)}(x_t^{(k)}, y_t^{(k)}; \hat{x}_t^{(k)}, \hat{y}_t^{(k)})\|^2] + 2\rho_x^2 \eta^4 \sigma^2 \frac{1}{K}$$

$$+ 4L^2 \frac{1}{K^2} \sum_{k=1}^{K} \mathbb{E}[\|x_{t+1}^{(k)} - x_t^{(k)}\|^2] + 4L^2 \frac{1}{K^2} \sum_{k=1}^{K} \mathbb{E}[\|y_{t+1}^{(k)} - y_t^{(k)}\|^2] , \tag{86}$$

where the last step holds due to the following inequality:

$$\mathbb{E}[\|\nabla_x F^{(k)}(x_t^{(k)}, y_t^{(k)}; \hat{x}_t^{(k)}, \hat{y}_t^{(k)}) - \nabla_x F^{(k)}(x_t^{(k)}, y_t^{(k)}; \hat{x}_t^{(k)}, \hat{y}_t^{(k)}; \xi_{t+1}^{(k)})$$
$$+ \nabla_x F^{(k)}(x_{t+1}^{(k)}, y_{t+1}^{(k)}; \hat{x}_{t+1}^{(k)}, \hat{y}_{t+1}^{(k)}; \xi_{t+1}^{(k)}) - \nabla_x F^{(k)}(x_{t+1}^{(k)}, y_{t+1}^{(k)}; \hat{x}_{t+1}^{(k)}, \hat{y}_{t+1}^{(k)})\|^2]$$
$$= \mathbb{E}[\|\nabla_x f(x_t^{(k)}, y_t^{(k)}) + \gamma_1(x_t^{(k)} - \hat{x}_t^{(k)}) - \nabla_x f(x_t^{(k)}, y_t^{(k)}; \xi_{t+1}^{(k)}) - \gamma_1(x_t^{(k)} - \hat{x}_t^{(k)})$$
$$+ \nabla_x f(x_{t+1}^{(k)}, y_{t+1}^{(k)}; \xi_{t+1}^{(k)}) + \gamma_1(x_{t+1}^{(k)} - \hat{x}_{t+1}^{(k)}) - \nabla_x f(x_{t+1}^{(k)}, y_{t+1}^{(k)}) - \gamma_1(x_{t+1}^{(k)} - \hat{x}_{t+1}^{(k)})\|^2]$$
$$= \mathbb{E}[\|\nabla_x f(x_t^{(k)}, y_t^{(k)}) - \nabla_x f(x_t^{(k)}, y_t^{(k)}; \xi_{t+1}^{(k)})$$
$$+ \nabla_x f(x_{t+1}^{(k)}, y_{t+1}^{(k)}; \xi_{t+1}^{(k)}) - \nabla_x f(x_{t+1}^{(k)}, y_{t+1}^{(k)})\|^2]$$
$$\leq \mathbb{E}[\|\nabla_x f(x_{t+1}^{(k)}, y_{t+1}^{(k)}; \xi_{t+1}^{(k)}) - \nabla_x f(x_t^{(k)}, y_t^{(k)}; \xi_{t+1}^{(k)})\|^2]$$
$$\leq 2L^2 \mathbb{E}[\|x_{t+1}^{(k)} - x_t^{(k)}\|^2] + 2L^2 \mathbb{E}[\|y_{t+1}^{(k)} - y_t^{(k)}\|^2] . \tag{87}$$

$\square$

**Lemma D.8.** *Given Assumptions 3.1-3.4, when $\eta \leq \frac{1}{\sqrt{\rho_x}}$, the following inequality holds:*

$$\frac{1}{K} \sum_{k=1}^{K} \mathbb{E}[\|u_{t+1}^{(k)} - \nabla_x F^{(k)}(x_{t+1}^{(k)}, y_{t+1}^{(k)}; \hat{x}_{t+1}^{(k)}, \hat{y}_{t+1}^{(k)})\|^2]$$

$$\leq (1 - \rho_x \eta^2)\frac{1}{K} \sum_{k=1}^{K} \mathbb{E}[\|u_t^{(k)} - \nabla_x F^{(k)}(x_t^{(k)}, y_t^{(k)}; \hat{x}_t^{(k)}, \hat{y}_t^{(k)})\|^2]$$

$$+ 4L^2 \frac{1}{K} \sum_{k=1}^{K} \mathbb{E}[\|x_{t+1}^{(k)} - x_t^{(k)}\|^2] + 4L^2 \frac{1}{K} \sum_{k=1}^{K} \mathbb{E}[\|y_{t+1}^{(k)} - y_t^{(k)}\|^2] + 2\rho_x^2 \eta^4 \sigma^2 . \tag{88}$$

This lemma can be proved by following Lemma D.7. Thus, we omit its proof.

**Lemma D.9.** *Given Assumptions 3.1-3.4, when $\eta \leq \frac{1}{\sqrt{\rho_y}}$, the following inequality holds:*

$$\mathbb{E}[\|\frac{1}{K} \sum_{k=1}^{K} \nabla_y F^{(k)}(x_{t+1}^{(k)}, y_{t+1}^{(k)}; \hat{x}_{t+1}^{(k)}, \hat{y}_{t+1}^{(k)}) - \frac{1}{K} \sum_{k=1}^{K} v_{t+1}^{(k)}\|^2]$$

$$\leq (1 - \rho_y \eta^2)\mathbb{E}[\|\frac{1}{K} \sum_{k=1}^{K} v_t^{(k)} - \frac{1}{K} \sum_{k=1}^{K} \nabla_y F^{(k)}(x_t^{(k)}, y_t^{(k)}; \hat{x}_t^{(k)}, \hat{y}_t^{(k)})\|^2]$$

$$+ 4L^2 \frac{1}{K^2} \sum_{k=1}^{K} \mathbb{E}[\|x_{t+1}^{(k)} - x_t^{(k)}\|^2] + 4L^2 \frac{1}{K^2} \sum_{k=1}^{K} \mathbb{E}[\|y_{t+1}^{(k)} - y_t^{(k)}\|^2] + 2\rho_y^2 \eta^4 \sigma^2 \frac{1}{K} . \tag{89}$$

This lemma can be proved by following Lemma D.7. Thus, we omit its proof.

**Lemma D.10.** *Given Assumptions 3.1-3.4, when $\eta \leq \frac{1}{\sqrt{\rho_y}}$, the following inequality holds:*

$$\frac{1}{K} \sum_{k=1}^{K} \mathbb{E}[\|\nabla_y F^{(k)}(x_{t+1}^{(k)}, y_{t+1}^{(k)}; \hat{x}_{t+1}^{(k)}, \hat{y}_{t+1}^{(k)}) - v_{t+1}^{(k)}\|^2]$$

$$\leq (1 - \rho_y \eta^2) \frac{1}{K} \sum_{k=1}^{K} \mathbb{E}[\|v_t^{(k)} - \nabla_y F^{(k)}(x_t^{(k)}, y_t^{(k)}; \hat{x}_t^{(k)}, \hat{y}_t^{(k)})\|^2]$$

$$+ 4L^2 \frac{1}{K} \sum_{k=1}^{K} \mathbb{E}[\|x_{t+1}^{(k)} - x_t^{(k)}\|^2] + 4L^2 \frac{1}{K} \sum_{k=1}^{K} \mathbb{E}[\|y_{t+1}^{(k)} - y_t^{(k)}\|^2] + 2\rho_y^2 \eta^4 \sigma^2 . \quad (90)$$

Similarly, this lemma can be proved by following Lemma D.7. Thus, we omit its proof.

### D.3 OTHER AUXILIARY LEMMAS

**Lemma D.11.** *Given Assumptions 3.1-3.4, the following inequality holds:*

$$\frac{1}{K} \sum_{k=1}^{K} \mathbb{E}[\|\hat{x}_{t+1}^{(k)} - \hat{x}_t^{(k)}\|^2] \leq 3\mathbb{E}[\|\bar{\hat{x}}_{t+1} - \bar{\hat{x}}_t\|^2] + 6\frac{1}{K} \sum_{k=1}^{K} \mathbb{E}[\|\bar{\hat{x}}_t - \hat{x}_t^{(k)}\|^2]$$

$$+ \frac{12\eta\hat{\beta}_x^2}{1 - \lambda^2} \frac{1}{K} \sum_{k=1}^{K} \mathbb{E}[\|\bar{x}_t - x_t^{(k)}\|^2] + \frac{12\eta\hat{\beta}_x^2}{1 - \lambda^2} \frac{2\eta\beta_x^2}{1 - \lambda^2} \frac{1}{K} \sum_{k=1}^{K} \mathbb{E}[\|\bar{p}_t - p_t^{(k)}\|^2] . \quad (91)$$

*Proof.*

$$\frac{1}{K} \sum_{k=1}^{K} \mathbb{E}[\|\hat{x}_{t+1}^{(k)} - \hat{x}_t^{(k)}\|^2]$$

$$= \frac{1}{K} \mathbb{E}[\|\hat{X}_{t+1} - \hat{X}_t\|_F^2]$$

$$\leq 3\frac{1}{K} \mathbb{E}[\|\hat{X}_{t+1} - \bar{\hat{X}}_{t+1}\|_F^2] + 3\frac{1}{K} \mathbb{E}[\|\hat{X}_t - \bar{\hat{X}}_t\|_F^2] + 3\frac{1}{K} \mathbb{E}[\|\bar{\hat{X}}_{t+1} - \bar{\hat{X}}_t\|_F^2]$$

$$\leq 3\mathbb{E}[\|\bar{\hat{x}}_{t+1} - \bar{\hat{x}}_t\|^2] + 6\frac{1}{K} \sum_{k=1}^{K} \mathbb{E}[\|\bar{\hat{x}}_t - \hat{x}_t^{(k)}\|^2]$$

$$+ \frac{12\eta\hat{\beta}_x^2}{1 - \lambda^2} \frac{1}{K} \sum_{k=1}^{K} \mathbb{E}[\|\bar{x}_t - x_t^{(k)}\|^2] + \frac{12\eta\hat{\beta}_x^2}{1 - \lambda^2} \frac{2\eta\beta_x^2}{1 - \lambda^2} \frac{1}{K} \sum_{k=1}^{K} \mathbb{E}[\|\bar{p}_t - p_t^{(k)}\|^2] , \quad (92)$$

where the last step holds due to Lemma D.3. $\qquad\square$

**Lemma D.12.** *Given Assumptions 3.1-3.4, the following inequality holds:*

$$\frac{1}{K} \sum_{k=1}^{K} \mathbb{E}[\|\hat{y}_{t+1}^{(k)} - \hat{y}_t^{(k)}\|^2] \leq 3\mathbb{E}[\|\bar{\hat{y}}_{t+1} - \bar{\hat{y}}_t\|^2] + 6\frac{1}{K} \sum_{k=1}^{K} \mathbb{E}[\|\bar{\hat{y}}_t - \hat{y}_t^{(k)}\|^2]$$

$$+ \frac{12\eta\hat{\beta}_y^2}{1 - \lambda^2} \frac{1}{K} \sum_{k=1}^{K} \mathbb{E}[\|\bar{y}_t - y_t^{(k)}\|^2] + \frac{12\eta\hat{\beta}_y^2}{1 - \lambda^2} \frac{2\eta\beta_y^2}{1 - \lambda^2} \frac{1}{K} \sum_{k=1}^{K} \mathbb{E}[\|\bar{q}_t - q_t^{(k)}\|^2] . \quad (93)$$

This lemma can be proved by following Lemma D.11. Thus, we omit its proof.

**Lemma D.13.** *Given Assumptions 3.1-3.4, the following inequality holds:*

$$\frac{1}{K} \sum_{k=1}^{K} \mathbb{E}[\|x_{t+1}^{(k)} - x_t^{(k)}\|^2] \leq 12\eta^2 \frac{1}{K} \sum_{k=1}^{K} \mathbb{E}[\|\bar{x}_t - x_t^{(k)}\|^2]$$

$$+ 3\beta_x^2 \eta^2 \frac{1}{K} \sum_{k=1}^{K} \mathbb{E}[\|\bar{p}_t - p_t^{(k)}\|^2] + 3\beta_x^2 \eta^2 \mathbb{E}[\|\bar{u}_t\|^2] . \quad (94)$$

**Lemma D.14.** *Given Assumptions 3.1-3.4, the following inequality holds:*

$$\frac{1}{K} \sum_{k=1}^{K} \mathbb{E}[\|y_{t+1}^{(k)} - y_t^{(k)}\|^2] \leq 12\eta^2 \frac{1}{K} \sum_{k=1}^{K} \mathbb{E}[\|\bar{y}_t - y_t^{(k)}\|^2]$$

$$+ 3\beta_y^2\eta^2 \frac{1}{K}\sum_{k=1}^{K}\mathbb{E}[\|\bar{q}_t - q_t^{(k)}\|^2] + 3\beta_y^2\eta^2\mathbb{E}[\|\bar{v}_t\|^2] . \tag{95}$$

Lemmas D.13, D.14 can be proved by following (Gao, 2022).

## E PROOF OF THEOREM 4.2

We first propose a novel potential function as follows:

$$\mathcal{L}_t = \mathcal{P}_t + c_1\mathbb{E}[\|\frac{1}{K}\sum_{k=1}^{K}u_t^{(k)} - \frac{1}{K}\sum_{k=1}^{K}\nabla_x F^{(k)}(x_t^{(k)}, y_t^{(k)}; \hat{x}_t^{(k)}, \hat{y}_t^{(k)})\|^2]$$

$$+ c_2\mathbb{E}[\|\frac{1}{K}\sum_{k=1}^{K}v_t^{(k)} - \frac{1}{K}\sum_{k=1}^{K}\nabla_y F^{(k)}(x_t^{(k)}, y_t^{(k)}; \hat{x}_t^{(k)}, \hat{y}_t^{(k)})\|^2]$$

$$+ c_3\frac{1}{K}\sum_{k=1}^{K}\mathbb{E}[\|\bar{x}_t - x_t^{(k)}\|^2] + c_4\frac{1}{K}\sum_{k=1}^{K}\mathbb{E}[\|\bar{y}_t - y_t^{(k)}\|^2] + c_5\frac{1}{K}\sum_{k=1}^{K}\mathbb{E}[\|\bar{\hat{x}}_t - \hat{x}_t^{(k)}\|^2]$$

$$+ c_{10}\frac{1}{K}\sum_{k=1}^{K}\mathbb{E}[\|\bar{\hat{y}}_t - \hat{y}_t^{(k)}\|^2] + c_6\frac{1}{K}\sum_{k=1}^{K}\mathbb{E}[\|\bar{p}_t - p_t^{(k)}\|^2] + c_7\frac{1}{K}\sum_{k=1}^{K}\mathbb{E}[\|\bar{q}_t - q_t^{(k)}\|^2]$$

$$+ c_8\frac{1}{K}\sum_{k=1}^{K}\mathbb{E}[\|u_t^{(k)} - \nabla_x F^{(k)}(x_t^{(k)}, y_t^{(k)}; \hat{x}_t^{(k)}, \hat{y}_t^{(k)})\|^2]$$

$$+ c_9\frac{1}{K}\sum_{k=1}^{K}\mathbb{E}[\|v_t^{(k)} - \nabla_y F^{(k)}(x_t^{(k)}, y_t^{(k)}; \hat{x}_t^{(k)}, \hat{y}_t^{(k)})\|^2] , \tag{96}$$

where the coefficient $\{c_i\}_{i=1}^{9}$ are positive.

Since

$$\mathbb{E}[\|\nabla_x F(\bar{x}_t, \bar{y}_t; \bar{\hat{x}}_t, \bar{\hat{y}}_t) - \bar{u}_t\|^2]$$

$$\leq 2\mathbb{E}[\|\nabla_x F(\bar{x}_t, \bar{y}_t; \bar{\hat{x}}_t, \bar{\hat{y}}_t) - \frac{1}{K}\sum_{k=1}^{K}\nabla_x F^{(k)}(x_t^{(k)}, y_t^{(k)}; \hat{x}_t^{(k)}, \hat{y}_t^{(k)})\|^2]$$

$$+ 2\mathbb{E}[\|\frac{1}{K}\sum_{k=1}^{K}\nabla_x F^{(k)}(x_t^{(k)}, y_t^{(k)}; \hat{x}_t^{(k)}, \hat{y}_t^{(k)}) - \bar{u}_t\|^2]$$

$$\leq 2L^2\frac{1}{K}\sum_{k=1}^{K}\mathbb{E}[\|\bar{x}_t - x_t^{(k)}\|^2] + 2L^2\frac{1}{K}\sum_{k=1}^{K}\mathbb{E}[\|\bar{y}_t - y_t^{(k)}\|^2]$$

$$+ 2\mathbb{E}[\|\frac{1}{K}\sum_{k=1}^{K}\nabla_x F^{(k)}(x_t^{(k)}, y_t^{(k)}; \hat{x}_t^{(k)}, \hat{y}_t^{(k)}) - \frac{1}{K}\sum_{k=1}^{K}u_t^{(k)}\|^2] , \tag{97}$$

and

$$\mathbb{E}[\|\nabla_y F(\bar{x}_t, \bar{y}_t; \bar{\hat{x}}_t, \bar{\hat{y}}_t) - \bar{v}_t\|^2] \leq 2L^2\frac{1}{K}\sum_{k=1}^{K}\mathbb{E}[\|\bar{x}_t - x_t^{(k)}\|^2] + 2L^2\frac{1}{K}\sum_{k=1}^{K}\mathbb{E}[\|\bar{y}_t - y_t^{(k)}\|^2]$$

$$+ 2\mathbb{E}[\|\frac{1}{K}\sum_{k=1}^{K}\nabla_y F^{(k)}(x_t^{(k)}, y_t^{(k)}; \hat{x}_t^{(k)}, \hat{y}_t^{(k)}) - \bar{v}_t\|^2] , \tag{98}$$

we obtain

$$\mathcal{L}_{t+1} - \mathcal{L}_t \leq -\frac{\beta_x\eta}{4}\mathbb{E}[\|\nabla_x F(\bar{x}_t, \bar{y}_t; \bar{\hat{x}}_t, \bar{\hat{y}}_t)\|^2] - \frac{\beta_y\eta}{2}\mathbb{E}[\|\nabla_y F(\bar{x}_t, \bar{y}_t; \bar{\hat{x}}_t, \bar{\hat{y}}_t)\|^2]$$

$$+ (\beta_x \eta - c_1 \rho_x \eta^2) \mathbb{E}[\| \frac{1}{K} \sum_{k=1}^{K} \nabla_x F^{(k)}(x_t^{(k)}, y_t^{(k)}; \hat{x}_t^{(k)}, \hat{y}_t^{(k)}) - \frac{1}{K} \sum_{k=1}^{K} u_t^{(k)} \|^2]$$

$$+ \left(2A_3 - \rho_y \eta^2 c_2\right) \mathbb{E}[\| \frac{1}{K} \sum_{k=1}^{K} \nabla_y f^{(k)}(x_t^{(k)}, y_t^{(k)}) - \frac{1}{K} \sum_{k=1}^{K} v_t^{(k)} \|^2]$$

$$+ \left(4\beta_y \eta \beta_x^2 \eta^2 L^2 - \frac{\beta_x \eta}{4}\right) \mathbb{E}[\|\bar{u}_t\|^2]$$

$$+ \left(\beta_y^2 \eta^2 L_d + \frac{3\beta_y \eta}{4} + \frac{\beta_y^2 \eta^2 (\gamma_2 + L)}{2} + 4A_1 \hat{\beta}_y^2 \eta^2 \beta_y^2 \eta^2 + 2A_2 \beta_y^2 \eta^2 - \frac{7}{8}\beta_y \eta\right) \mathbb{E}[\|\bar{v}_t\|^2]$$

$$+ \left(2\gamma_1 C_{x_{\hat{x}\hat{y}}^1} + \frac{\gamma_1}{6\hat{\beta}_x \eta} + 6\gamma_1 \hat{\beta}_x \eta \left(10 C_{x_{y\hat{x}\hat{y}}^1}^2 C_{y_{\hat{x}\hat{y}}^1}^2 + \frac{4}{\gamma_1 - L} \frac{2\gamma_2^2 C_{y_{\hat{x}\hat{y}}^1}^2}{\mu}\right) - \frac{\gamma_1(2 - \hat{\beta}_x \eta)}{2\hat{\beta}_x \eta}\right) \mathbb{E}[\|\bar{\hat{x}}_{t+1} - \bar{\hat{x}}_t\|^2]$$

$$+ \left(2A_1 - \frac{\gamma_2(2 - \hat{\beta}_y \eta)}{2\hat{\beta}_y \eta}\right) \mathbb{E}[\|\bar{\hat{y}}_{t+1} - \bar{\hat{y}}_t\|^2]$$

$$+ \left(\frac{4L^2 c_1}{K} + \frac{4L^2 c_2}{K} + \frac{9(L^2 + \gamma_1^2)c_6}{1 - \lambda} + \frac{9L^2 c_7}{1 - \lambda} + 4L^2 c_8 + 4L^2 c_9\right) \frac{1}{K} \sum_{k=1}^{K} \mathbb{E}[\|x_{t+1}^{(k)} - x_t^{(k)}\|^2]$$

$$+ \left(\frac{4L^2 c_1}{K} + \frac{4L^2 c_2}{K} + \frac{9L^2 c_6}{1 - \lambda} + \frac{9(L^2 + \gamma_2^2)c_7}{1 - \lambda} + 4L^2 c_8 + 4L^2 c_9\right) \frac{1}{K} \sum_{k=1}^{K} \mathbb{E}[\|y_{t+1}^{(k)} - y_t^{(k)}\|^2]$$

$$+ \left(\frac{9\gamma_1^2}{1 - \lambda} c_6\right) \frac{1}{K} \sum_{k=1}^{K} \mathbb{E}[\|\hat{x}_{t+1}^{(k)} - \hat{x}_t^{(k)}\|^2] + \left(\frac{9\gamma_2^2}{1 - \lambda} c_7\right) \frac{1}{K} \sum_{k=1}^{K} \mathbb{E}[\|\hat{y}_{t+1}^{(k)} - \hat{y}_t^{(k)}\|^2]$$

$$+ \left(\beta_x \eta L^2 + 2L^2 A_3 + \frac{4\eta \hat{\beta}_x^2}{1 - \lambda^2} c_5 - \frac{\eta(1 - \lambda^2)}{2} c_3\right) \frac{1}{K} \sum_{k=1}^{K} \mathbb{E}[\|\bar{x}_t - x_t^{(k)}\|^2]$$

$$+ \left(\beta_x \eta L^2 + 2L^2 A_3 + \frac{4\eta \hat{\beta}_y^2}{1 - \lambda^2} c_{10} - \frac{\eta(1 - \lambda^2)}{2} c_4\right) \frac{1}{K} \sum_{k=1}^{K} \mathbb{E}[\|\bar{y}_t - y_t^{(k)}\|^2]$$

$$+ \left(\frac{2\eta \beta_x^2}{1 - \lambda^2} c_3 + \frac{8\eta^2 \beta_x^2 \hat{\beta}_x^2}{(1 - \lambda^2)^2} c_5 - (1 - \lambda)c_6\right) \frac{1}{K} \sum_{k=1}^{K} \mathbb{E}[\|\bar{p}_t - p_t^{(k)}\|^2]$$

$$+ \left(\frac{2\eta \beta_y^2}{1 - \lambda^2} c_4 + \frac{8\eta^2 \beta_y^2 \hat{\beta}_y^2}{(1 - \lambda^2)^2} c_{10} - (1 - \lambda)c_7\right) \frac{1}{K} \sum_{k=1}^{K} \mathbb{E}[\|\bar{q}_t - q_t^{(k)}\|^2]$$

$$+ \left(-\frac{\eta(1 - \lambda^2)}{4} c_5\right) \frac{1}{K} \sum_{k=1}^{K} \mathbb{E}[\|\bar{\hat{x}}_t - \hat{x}_t^{(k)}\|^2] + \left(-\frac{\eta(1 - \lambda^2)}{4} c_{10}\right) \frac{1}{K} \sum_{k=1}^{K} \mathbb{E}[\|\bar{\hat{y}}_t - \hat{y}_t^{(k)}\|^2]$$

$$+ \left(3\rho_x^2 \eta^4 \frac{1}{1 - \lambda} c_6 - \rho_x \eta^2 c_8\right) \frac{1}{K} \sum_{k=1}^{K} \mathbb{E}[\|u_t^{(k)} - \nabla_x F^{(k)}(x_t^{(k)}, y_t^{(k)}; \hat{x}_t^{(k)}, \hat{y}_t^{(k)})\|^2]$$

$$+ \left(3\rho_y^2 \eta^4 \frac{1}{1 - \lambda} c_7 - \rho_y \eta^2 c_9\right) \frac{1}{K} \sum_{k=1}^{K} \mathbb{E}[\|v_t^{(k)} - \nabla_y F^{(k)}(x_t^{(k)}, y_t^{(k)}; \hat{x}_t^{(k)}, \hat{y}_t^{(k)})\|^2]$$

$$+ 2c_1 \rho_x^2 \eta^4 \sigma^2 \frac{1}{K} + 2c_2 \rho_y^2 \eta^4 \sigma^2 \frac{1}{K} + 3c_6 \rho_x^2 \eta^4 \sigma^2 \frac{1}{1 - \lambda} + 3c_7 \rho_y^2 \eta^4 \sigma^2 \frac{1}{1 - \lambda} + 2c_8 \rho_x^2 \eta^4 \sigma^2 + 2c_9 \rho_y^2 \eta^4 \sigma^2 .$$

$$\tag{99}$$

By setting

$$\mathcal{X} = \frac{4L^2 c_1}{K} + \frac{4L^2 c_2}{K} + \frac{9(L^2 + \gamma_1^2)c_6}{1 - \lambda} + \frac{9L^2 c_7}{1 - \lambda} + 4L^2 c_8 + 4L^2 c_9 ,$$

$$\mathcal{Y} = \frac{4L^2 c_1}{K} + \frac{4L^2 c_2}{K} + \frac{9L^2 c_6}{1-\lambda} + \frac{9(L^2 + \gamma_2^2)c_7}{1-\lambda} + 4L^2 c_8 + 4L^2 c_9 \,, \tag{100}$$

and due to $\lambda < 1$, we obtain $\frac{1}{1-\lambda^2} \le \frac{1}{1-\lambda}$, and further derive

$$\mathcal{L}_{t+1} - \mathcal{L}_t \le -\frac{\beta_x \eta}{4} \mathbb{E}[\|\nabla_x F(\bar{x}_t, \bar{y}_t; \bar{\hat{x}}_t, \bar{\hat{y}}_t)\|^2] - \frac{\beta_y \eta}{2} \mathbb{E}[\|\nabla_y F(\bar{x}_t, \bar{y}_t; \bar{\hat{x}}_t, \bar{\hat{y}}_t)\|^2]$$

$$+ (\beta_x \eta - c_1 \rho_x \eta^2) \mathbb{E}[\|\frac{1}{K} \sum_{k=1}^{K} \nabla_x F^{(k)}(x_t^{(k)}, y_t^{(k)}; \hat{x}_t^{(k)}, \hat{y}_t^{(k)}) - \frac{1}{K} \sum_{k=1}^{K} u_t^{(k)}\|^2]$$

$$+ \left(2\beta_y \eta + 240\gamma_1 \hat{\beta}_x \eta \beta_y^2 \eta^2 C_{x_{y\hat{x}\hat{y}}^1}^2 + 8A_2 \beta_y^2 \eta^2 - \rho_y \eta^2 c_2\right) \mathbb{E}[\|\frac{1}{K} \sum_{k=1}^{K} \nabla_y f^{(k)}(x_t^{(k)}, y_t^{(k)}) - \frac{1}{K} \sum_{k=1}^{K} v_t^{(k)}\|^2]$$

$$+ \left(4\beta_y \eta \beta_x^2 \eta^2 L^2 + 3\beta_x^2 \eta^2 \mathcal{X} - \frac{\beta_x \eta}{4}\right) \mathbb{E}[\|\bar{u}_t\|^2]$$

$$+ \left(\beta_y^2 \eta^2 L_d + \frac{3\beta_y \eta}{4} + \frac{\beta_y^2 \eta^2 (\gamma_2 + L)}{2} + 4A_1 \hat{\beta}_y^2 \eta^2 \beta_y^2 \eta^2 + 2A_2 \beta_y^2 \eta^2 + 3\beta_y^2 \eta^2 \mathcal{Y} - \frac{7}{8}\beta_y \eta\right) \mathbb{E}[\|\bar{v}_t\|^2]$$

$$+ \left(2\gamma_1 C_{x_{\hat{x}\hat{y}}^1} + \frac{\gamma_1}{6\hat{\beta}_x \eta} + 6\gamma_1 \hat{\beta}_x \eta \left(10 C_{x_{y\hat{x}\hat{y}}^1}^2 C_{y_{\hat{x}\hat{y}}^1}^2 + \frac{4}{\gamma_1 - L} \frac{2\gamma_2^2 C_{y_{\hat{x}\hat{y}}^1}^2}{\mu}\right) + \frac{27\gamma_1^2}{1-\lambda} c_6\right.$$

$$\left. - \frac{\gamma_1(2 - \hat{\beta}_x \eta)}{2\hat{\beta}_x \eta}\right) \mathbb{E}[\|\bar{\hat{x}}_{t+1} - \bar{\hat{x}}_t\|^2]$$

$$+ \left(2A_1 + \frac{27\gamma_2^2}{1-\lambda} c_7 - \frac{\gamma_2(2 - \hat{\beta}_y \eta)}{2\hat{\beta}_y \eta}\right) \mathbb{E}[\|\bar{\hat{y}}_{t+1} - \bar{\hat{y}}_t\|^2]$$

$$+ \left(\beta_x \eta L^2 + 2L^2 A_3 + \frac{4\eta \hat{\beta}_x^2}{1-\lambda} c_5 + \frac{108\eta \hat{\beta}_x^2 \gamma_1^2}{(1-\lambda)^2} c_6 + 12\eta^2 \mathcal{X} - \frac{\eta(1-\lambda^2)}{2} c_3\right) \frac{1}{K} \sum_{k=1}^{K} \mathbb{E}[\|\bar{x}_t - x_t^{(k)}\|^2]$$

$$+ \left(\beta_x \eta L^2 + 2L^2 A_3 + \frac{4\eta \hat{\beta}_y^2}{1-\lambda} c_{10} + \frac{108\eta \hat{\beta}_y^2 \gamma_2^2}{(1-\lambda)^2} c_7 + 12\eta^2 \mathcal{Y} - \frac{\eta(1-\lambda^2)}{2} c_4\right) \frac{1}{K} \sum_{k=1}^{K} \mathbb{E}[\|\bar{y}_t - y_t^{(k)}\|^2]$$

$$+ \left(\frac{2\eta \beta_x^2}{1-\lambda} c_3 + \frac{8\eta^2 \beta_x^2 \hat{\beta}_x^2}{(1-\lambda)^2} c_5 + \frac{216\eta^2 \beta_x^2 \hat{\beta}_x^2 \gamma_1^2}{(1-\lambda)^3} c_6 + 3\beta_x^2 \eta^2 \mathcal{X} - (1-\lambda)c_6\right) \frac{1}{K} \sum_{k=1}^{K} \mathbb{E}[\|\bar{p}_t - p_t^{(k)}\|^2]$$

$$+ \left(\frac{2\eta \beta_y^2}{1-\lambda} c_4 + \frac{8\eta^2 \beta_y^2 \hat{\beta}_y^2}{(1-\lambda)^2} c_{10} + \frac{216\eta^2 \beta_y^2 \hat{\beta}_y^2 \gamma_2^2}{(1-\lambda)^3} c_7 + 3\beta_y^2 \eta^2 \mathcal{Y} - (1-\lambda)c_7\right) \frac{1}{K} \sum_{k=1}^{K} \mathbb{E}[\|\bar{q}_t - q_t^{(k)}\|^2]$$

$$+ \left(\frac{54\gamma_1^2}{1-\lambda} c_6 - \frac{\eta(1-\lambda^2)}{4} c_5\right) \frac{1}{K} \sum_{k=1}^{K} \mathbb{E}[\|\bar{\hat{x}}_t - \hat{x}_t^{(k)}\|^2] + \left(\frac{54\gamma_2^2}{1-\lambda} c_7 - \frac{\eta(1-\lambda^2)}{4} c_{10}\right) \frac{1}{K} \sum_{k=1}^{K} \mathbb{E}[\|\bar{\hat{y}}_t - \hat{y}_t^{(k)}\|^2]$$

$$+ \left(\frac{3\rho_x^2 \eta^4}{1-\lambda} c_6 - \rho_x \eta^2 c_8\right) \frac{1}{K} \sum_{k=1}^{K} \mathbb{E}[\|u_t^{(k)} - \nabla_x F^{(k)}(x_t^{(k)}, y_t^{(k)}; \hat{x}_t^{(k)}, \hat{y}_t^{(k)})\|^2]$$

$$+ \left(\frac{3\rho_y^2 \eta^4}{1-\lambda} c_7 - \rho_y \eta^2 c_9\right) \frac{1}{K} \sum_{k=1}^{K} \mathbb{E}[\|v_t^{(k)} - \nabla_y F^{(k)}(x_t^{(k)}, y_t^{(k)}; \hat{x}_t^{(k)}, \hat{y}_t^{(k)})\|^2]$$

$$+ 2c_1 \rho_x^2 \eta^4 \sigma^2 \frac{1}{K} + 2c_2 \rho_y^2 \eta^4 \sigma^2 \frac{1}{K} + 3c_6 \rho_x^2 \eta^4 \sigma^2 \frac{1}{1-\lambda} + 3c_7 \rho_y^2 \eta^4 \sigma^2 \frac{1}{1-\lambda} + 2c_8 \rho_x^2 \eta^4 \sigma^2 + 2c_9 \rho_y^2 \eta^4 \sigma^2 \,. \tag{101}$$

To cancel out $\mathbb{E}[\|\frac{1}{K} \sum_{k=1}^{K} \nabla_x F^{(k)}(x_t^{(k)}, y_t^{(k)}; \hat{x}_t^{(k)}, \hat{y}_t^{(k)}) - \frac{1}{K} \sum_{k=1}^{K} u_t^{(k)}\|^2]$, i.e.,

$$\beta_x \eta - \rho_x \eta^2 c_1 \le 0 \,. \tag{102}$$

Then, we set

$$c_1 = \frac{\beta_x}{\rho_x \eta} \ . \tag{103}$$

To cancel out $\mathbb{E}[\|\frac{1}{K}\sum_{k=1}^{K}\nabla_y f^{(k)}(x_t^{(k)}, y_t^{(k)}) - \frac{1}{K}\sum_{k=1}^{K}v_t^{(k)}\|^2]$, i.e.,

$$2\beta_y\eta + 240\gamma_1\hat{\beta}_x\eta\beta_y^2\eta^2 C_{x_{y\hat{x}\hat{y}}^1}^2 + 8A_2\beta_y^2\eta^2 - \rho_y\eta^2 c_2 \leq 0 \ . \tag{104}$$

Specifically, since the second and last inequality in Eq. (67) holds, we have

$$240\gamma_1\hat{\beta}_x\eta\beta_y^2\eta^2 C_{x_{y\hat{x}\hat{y}}^1}^2 \leq \frac{2\beta_x\eta}{32\times 16}\frac{(\gamma_1-L)^2}{L^2} \ ,$$

$$8A_2\beta_y^2\eta^2 \leq \frac{2\beta_x\eta}{32\times 16}\frac{(\gamma_1-L)^2}{L^2} \ . \tag{105}$$

Then, by the definition of $c_{\beta_y}$, i.e., $c_{\beta_y} = \frac{(\gamma_1-L)^2}{64L^2}$, we set

$$240\gamma_1\hat{\beta}_x\eta\beta_y^2\eta^2 C_{x_{y\hat{x}\hat{y}}^1}^2 \leq \frac{2\beta_x\eta}{32\times 16}64c_{\beta_y} = \frac{1}{4}\beta_y\eta \ ,$$

$$8A_2\beta_y^2\eta^2 \leq \frac{2\beta_x\eta}{32\times 16}64c_{\beta_y} = \frac{\beta_y\eta}{4} \ . \tag{106}$$

Therefore, we obtain

$$c_2 = \frac{5\beta_y}{2\rho_y\eta} \ . \tag{107}$$

To cancel out $\frac{1}{K}\sum_{k=1}^{K}\mathbb{E}[\|u_t^{(k)} - \nabla_x F^{(k)}(x_t^{(k)}, y_t^{(k)}; \hat{x}_t^{(k)}, \hat{y}_t^{(k)})\|^2]$, i.e.,

$$\frac{3\rho_x^2\eta^4}{1-\lambda}c_6 - \rho_x\eta^2 c_8 \leq 0 \ . \tag{108}$$

Here, because $\rho_x\eta^2 < 1$, we set

$$c_6 = \beta_x\eta(1-\lambda) \ , \quad c_8 = 3\beta_x\eta \ . \tag{109}$$

Similarly, to cancel out $\frac{1}{K}\sum_{k=1}^{K}\mathbb{E}[\|v_t^{(k)} - \nabla_y F^{(k)}(x_t^{(k)}, y_t^{(k)}; \hat{x}_t^{(k)}, \hat{y}_t^{(k)})\|^2]$, i.e.,

$$\frac{3\rho_y^2\eta^4}{1-\lambda}c_7 - \rho_y\eta^2 c_9 \leq 0 \ . \tag{110}$$

Because $\rho_y\eta^2 < 1$, we set

$$c_7 = \beta_y\eta(1-\lambda) \ , \quad c_9 = 3\beta_y\eta \ . \tag{111}$$

To cancel out $\frac{1}{K}\sum_{k=1}^{K}\mathbb{E}[\|\bar{\hat{x}}_t - \hat{x}_t^{(k)}\|^2]$, i.e.,

$$\frac{54\gamma_1^2}{1-\lambda}c_6 - \frac{\eta(1-\lambda^2)}{4}c_5 \leq 0 \ , \tag{112}$$

we set

$$c_5 = \frac{216\beta_x\gamma_1^2}{(1-\lambda)} \ . \tag{113}$$

To cancel out $\frac{1}{K}\sum_{k=1}^{K}\mathbb{E}[\|\bar{\hat{y}}_t - \hat{y}_t^{(k)}\|^2]$, i.e.,

$$\frac{54\gamma_2^2}{1-\lambda}c_7 - \frac{\eta(1-\lambda^2)}{4}c_{10} \leq 0 \ , \tag{114}$$

we set

$$c_{10} = \frac{216\beta_y\gamma_2^2}{(1-\lambda)} \ . \tag{115}$$

To cancel out $\frac{1}{K}\sum_{k=1}^{K}\mathbb{E}[\|\bar{x}_t - x_t^{(k)}\|^2]$, i.e.,

$$\beta_x\eta L^2 + 2L^2 A_3 + \frac{4\eta\hat{\beta}_x^2}{1-\lambda}c_5 + \frac{108\eta\hat{\beta}_x^2\gamma_1^2}{(1-\lambda)^2}c_6 + 12\eta^2\mathcal{X} - \frac{\eta(1-\lambda^2)}{2}c_3 \le 0 \ . \tag{116}$$

Firstly, from the definition of $\mathcal{X}$, we have

$$\begin{aligned}
\mathcal{X} &= \frac{4L^2 c_1}{K} + \frac{4L^2 c_2}{K} + \frac{9(L^2+\gamma_1^2)c_6}{1-\lambda} + \frac{9L^2 c_7}{1-\lambda} + 4L^2 c_8 + 4L^2 c_9 \ , \\
&= \frac{4L^2}{K}\frac{\beta_x}{\rho_x\eta} + \frac{4L^2}{K}\frac{5\beta_y}{2\rho_y\eta} + 9(L^2+\gamma_1^2)\beta_x\eta + 9L^2\beta_y\eta + 12L^2\beta_x\eta + 12L^2\beta_y\eta \ , \\
&= \frac{4L^2}{K}\frac{\beta_x}{\rho_x\eta} + \frac{10L^2}{K}\frac{\beta_y}{\rho_y\eta} + (21L^2+9\gamma_1^2)\beta_x\eta + 21L^2\beta_y\eta \ . \tag{117}
\end{aligned}$$

Moreover, from the definition of $A_3$ and Eq. (104), we have

$$\beta_y\eta + 120\gamma_1\hat{\beta}_x\eta\beta_y^2\eta^2 C_{x_y^1\hat{x}\hat{y}}^2 + 4A_2\beta_y^2\eta^2 \le \frac{5}{4}\beta_y\eta \tag{118}$$

Therefore, we set

$$\begin{aligned}
&\beta_x\eta L^2 + 2L^2 A_3 + \frac{4\eta\hat{\beta}_x^2}{1-\lambda}c_5 + \frac{108\eta\hat{\beta}_x^2\gamma_1^2}{(1-\lambda)^2}c_6 + 12\eta^2\mathcal{X} \\
&\le \beta_x\eta L^2 + \frac{5}{2}\beta_y\eta L^2 + \frac{4\eta\hat{\beta}_x^2}{1-\lambda}\frac{216\beta_x\gamma_1^2}{(1-\lambda)} + \frac{108\gamma_1^2\eta\hat{\beta}_x^2}{(1-\lambda)}\beta_x\eta \\
&\quad + 12\eta^2\left(\frac{4L^2}{K}\frac{\beta_x}{\rho_x\eta} + \frac{10L^2}{K}\frac{\beta_y}{\rho_y\eta} + (21L^2+9\gamma_1^2)\beta_x\eta + 21L^2\beta_y\eta\right) \\
&\le \beta_x\eta L^2 + \beta_x\eta L^2\frac{5\beta_y}{2\beta_x} + \beta_x\eta\hat{\beta}_x^2\frac{864\gamma_1^2}{(1-\lambda)^2} + \beta_x\eta\hat{\beta}_x\frac{108\gamma_1^2}{(1-\lambda)} \\
&\quad + 12\eta\left(\frac{4L^2}{\rho_x K}\beta_x + \frac{10L^2}{\rho_y K}\beta_y + (21L^2+9\gamma_1^2)\beta_x + 21L^2\beta_y\right) \\
&= \beta_x\eta L^2 + \beta_x\eta L^2\frac{5}{2}c_{\beta_y} + \beta_x^3\eta c_{\hat{\beta}_x}^2\frac{864\gamma_1^2}{(1-\lambda)^2} + \beta_x^2\eta c_{\hat{\beta}_x}\frac{108\gamma_1^2}{(1-\lambda)} \\
&\quad + 12\beta_x\eta\left(\frac{4L^2}{\rho_x K} + \frac{10L^2}{\rho_y K}c_{\beta_y} + (21L^2+9\gamma_1^2) + 21L^2 c_{\beta_y}\right) \\
&\le \frac{\eta(1-\lambda)}{2}c_3 \ , \tag{119}
\end{aligned}$$

where the second step holds due to $\hat{\beta}_x\eta < 1$ and $\eta < 1$, the fourth step holds due to Eq. (55) . By solving this inequality, we obtain

$$c_3 \ge \frac{2\beta_x}{(1-\lambda)}\left(\frac{48L^2}{\rho_x K} + \frac{120L^2}{\rho_y K}c_{\beta_y} + 253L^2 + 108\gamma_1^2 + 255L^2 c_{\beta_y} + \beta_x^2 c_{\hat{\beta}_x}^2\frac{864\gamma_1^2}{(1-\lambda)^2} + \beta_x c_{\hat{\beta}_x}\frac{108\gamma_1^2}{(1-\lambda)}\right) \ . \tag{120}$$

Then, we set

$$c_3 = \frac{2\beta_x}{(1-\lambda)}\left(\underbrace{\frac{48L^2}{\rho_x K} + \frac{120L^2}{\rho_y K}c_{\beta_y} + 253L^2 + 108\gamma_1^2 + 302L^2 c_{\beta_y}}_{c_{3,1}}\right)$$

$$+ \frac{2\beta_x^3}{(1-\lambda)^3} \underbrace{864\gamma_1^2 c_{\hat{\beta}_x}^2}_{c_{3,2}} + \frac{2\beta_x^2}{(1-\lambda)^2} \underbrace{108\gamma_1^2 c_{\hat{\beta}_x}}_{c_{3,3}} . \tag{121}$$

Here, it is easy to know that $c_{3,1} = O(1)$ when $\rho_x = O(1/K)$ and $\rho_y = O(1/K)$, $c_{3,2} = O(1/\kappa^2)$ and $c_{3,3} = O(1/\kappa)$ due to $c_{\hat{\beta}_x} = O(1/\kappa)$.

To cancel out $\frac{1}{K}\sum_{k=1}^{K} \mathbb{E}[\|\bar{y}_t - y_t^{(k)}\|^2]$, i.e.,

$$\beta_x \eta L^2 + 2L^2 A_3 + \frac{4\eta\hat{\beta}_y^2}{1-\lambda}c_{10} + \frac{108\eta\hat{\beta}_y^2\gamma_2^2}{(1-\lambda)^2}c_7 + 12\eta^2\mathcal{Y} - \frac{\eta(1-\lambda^2)}{2}c_4 \le 0 . \tag{122}$$

Firstly, from the definition of $\mathcal{Y}$, we have

$$\mathcal{Y} = \frac{4L^2 c_1}{K} + \frac{4L^2 c_2}{K} + \frac{9L^2 c_6}{1-\lambda} + \frac{9(L^2+\gamma_2^2)c_7}{1-\lambda} + 4L^2 c_8 + 4L^2 c_9$$

$$= \frac{4L^2}{K}\frac{\beta_x}{\rho_x\eta} + \frac{10L^2}{K}\frac{\beta_y}{\rho_y\eta} + 21L^2\beta_x\eta + (9L^2 + 21\gamma_2^2)\beta_y\eta . \tag{123}$$

Therefore, we set

$$\beta_x\eta L^2 + 2L^2 A_3 + \frac{4\eta\hat{\beta}_y^2}{1-\lambda}c_{10} + \frac{108\eta\hat{\beta}_y^2\gamma_2^2}{(1-\lambda)^2}c_7 + 12\eta^2\mathcal{Y}$$

$$\le \beta_x\eta L^2 + \beta_x\eta L^2\frac{5}{2}c_{\beta_y} + \beta_x^3\eta c_{\beta_y}c_{\hat{\beta}_x}^2\frac{864\gamma_2^2}{(1-\lambda)^2} + \beta_x^2\eta c_{\beta_y}c_{\hat{\beta}_x}\frac{108\gamma_2^2}{(1-\lambda)}$$

$$+ 12\beta_x\eta\left(\frac{4L^2}{\rho_x K} + \frac{10L^2}{\rho_y K}c_{\beta_y} + 21L^2 + (21L^2 + 9\gamma_2^2)c_{\beta_y}\right)$$

$$\le \frac{\eta(1-\lambda)}{2}c_4 , \tag{124}$$

where the second step holds due to $\hat{\beta}_y\eta < 1$ and $\eta < 1$, the fourth step holds due to Eq. (55). By solving this inequality, we set

$$c_4 = \frac{2\beta_x}{(1-\lambda)}\left(\underbrace{\frac{48L^2}{\rho_x K} + \frac{120L^2}{\rho_y K}c_{\beta_y} + 253L^2 + 255L^2 c_{\beta_y} + 108\gamma_2^2 c_{\beta_y}}_{c_{4,1}}\right)$$

$$+ \frac{2\beta_x^3}{(1-\lambda)^3}\underbrace{864\gamma_2^2 c_{\beta_y}c_{\hat{\beta}_x}^2}_{c_{4,2}} + \frac{2\beta_x^2}{(1-\lambda)^2}\underbrace{108\gamma_2^2 c_{\beta_y}c_{\hat{\beta}_x}}_{c_{4,3}} . \tag{125}$$

Similarly, it is easy to know that $c_{4,1} = O(1)$ when $\rho_x = O(1/K)$ and $\rho_y = O(1/K)$, $c_{4,2} = O(1/\kappa^2)$ and $c_{4,3} = O(1/\kappa)$ due to $c_{\hat{\beta}_x} = O(1/\kappa)$.

To cancel out $\frac{1}{K}\sum_{k=1}^{K} \mathbb{E}[\|\bar{p}_t - p_t^{(k)}\|^2]$, i.e.,

$$\frac{2\eta\beta_x^2}{1-\lambda}c_3 + \frac{8\eta^2\beta_x^2\hat{\beta}_x^2}{(1-\lambda)^2}c_5 + \frac{216\eta^2\beta_x^2\hat{\beta}_x^2\gamma_1^2}{(1-\lambda)^3}c_6 + 3\beta_x^2\eta^2\mathcal{X} - (1-\lambda)c_6 \le 0 . \tag{126}$$

Firstly, we enforce

$$\frac{216\eta^2\beta_x^2\hat{\beta}_x^2\gamma_1^2}{(1-\lambda)^3}c_6 \le \frac{(1-\lambda)}{4}c_6 . \tag{127}$$

Then, based on Eq. (55) , we obtain

$$\beta_x \le \frac{(1-\lambda)}{6\sqrt{\gamma_1 c_{\hat{\beta}_x}}} . \tag{128}$$

Then, we enforce

$$c_3 \frac{2\eta\beta_x^2}{1-\lambda} \leq \frac{\beta_x\eta}{4}(1-\lambda)^2 \,,$$

$$c_5 \frac{8\eta^2\beta_x^2\hat{\beta}_x^2}{(1-\lambda)^2} \leq \frac{\beta_x\eta}{4}(1-\lambda)^2 \,, \tag{129}$$

$$3\beta_x^2\eta^2\mathcal{X} \leq \frac{\beta_x\eta}{16}(1-\lambda)^2 \,.$$

To solve the first inequality in Eq. (129), we enforce

$$\frac{2\eta\beta_x^2}{1-\lambda}\frac{2\beta_x}{(1-\lambda)}c_{3,1} \leq \frac{\beta_x\eta}{12}(1-\lambda)^2 \,,$$

$$\frac{2\eta\beta_x^2}{1-\lambda}\frac{2\beta_x^3}{(1-\lambda)^3}c_{3,2} \leq \frac{\beta_x\eta}{12}(1-\lambda)^2 \,, \tag{130}$$

$$\frac{2\eta\beta_x^2}{1-\lambda}\frac{2\beta_x^2}{(1-\lambda)^2}c_{3,3} \leq \frac{\beta_x\eta}{12}(1-\lambda)^2 \,.$$

Therefore, we obtain

$$\beta_x \leq \min\left\{\frac{(1-\lambda)^2}{4\sqrt{3c_{3,1}}} \,, \frac{(1-\lambda)^{3/2}}{2(3c_{3,2})^{1/4}} \,, \frac{(1-\lambda)^{5/3}}{2(6c_{3,3})^{1/3}}\right\} \,. \tag{131}$$

To solve the second inequality in Eq. (129), we obtain

$$\beta_x \leq \frac{(1-\lambda)^{5/4}}{12\sqrt{\gamma_1 c_{\hat{\beta}_x}}} \,. \tag{132}$$

To solve the last inequality in Eq. (129), we enforce

$$3\beta_x\eta\frac{4L^2}{K}\frac{\beta_x}{\rho_x\eta} \leq \frac{1}{16\times4}(1-\lambda)^2 \,, \quad 3\beta_x\eta\frac{10L^2}{K}\frac{\beta_y}{\rho_y\eta} \leq \frac{1}{16\times4}(1-\lambda)^2 \,,$$

$$3\beta_x\eta(21L^2+9\gamma_1^2)\beta_x\eta \leq \frac{1}{16\times4}(1-\lambda)^2 \,, \quad 3\beta_x\eta21L^2\beta_y\eta \leq \frac{1}{16\times4}(1-\lambda)^2 \,. \tag{133}$$

We obtain

$$\beta_x \leq \left\{\frac{\sqrt{\rho_x K}(1-\lambda)}{16\sqrt{3}L} \,, \frac{\sqrt{\rho_y K}(1-\lambda)}{8L\sqrt{30c_{\beta_y}}} \,, \frac{(1-\lambda)}{8\sqrt{3(21L^2+9\gamma_1^2)}} \,, \frac{(1-\lambda)}{24L\sqrt{7c_{\beta_y}}}\right\} \,. \tag{134}$$

To cancel out $\frac{1}{K}\sum_{k=1}^K \mathbb{E}[\|\bar{q}_t - q_t^{(k)}\|^2]$, i.e.,

$$\frac{2\eta\beta_y^2}{1-\lambda}c_4 + \frac{8\eta^2\beta_y^2\hat{\beta}_y^2}{(1-\lambda)^2}c_{10} + \frac{216\eta^2\beta_y^2\hat{\beta}_y^2\gamma_2^2}{(1-\lambda)^3}c_7 + 3\beta_y^2\eta^2\mathcal{Y} - (1-\lambda)c_7 \leq 0 \,. \tag{135}$$

Firstly, we enforce

$$\frac{216\eta^2\beta_y^2\hat{\beta}_y^2\gamma_2^2}{(1-\lambda)^3}c_7 \leq \frac{(1-\lambda)}{6}c_7 \,. \tag{136}$$

Then, based on Eq. (55), we obtain

$$\beta_x \leq \frac{(1-\lambda)}{6\sqrt{\gamma_2 c_{\beta_y} c_{\hat{\beta}_y}}} \,. \tag{137}$$

Then, we enforce

$$c_4 \frac{2\eta\beta_y^2}{1-\lambda} \leq \frac{\beta_y\eta}{4}(1-\lambda)^2 \,,$$

$$c_{10} \frac{8\eta^2 \beta_y^2 \hat{\beta}_y^2}{(1-\lambda)^2} \le \frac{\beta_y \eta}{4}(1-\lambda)^2 \ , \tag{138}$$

$$3\beta_y^2 \eta^2 \mathcal{Y} \le \frac{\beta_y \eta}{96}(1-\lambda)^2 \ .$$

To solve the first inequality in Eq. (138), we enforce

$$\frac{2\eta \beta_y^2}{1-\lambda} \frac{2\beta_x}{(1-\lambda)} c_{4,1} \le \frac{\beta_y \eta}{12}(1-\lambda)^2 \ ,$$

$$\frac{2\eta \beta_y^2}{1-\lambda} \frac{2\beta_x^3}{(1-\lambda)^3} c_{4,2} \le \frac{\beta_y \eta}{12}(1-\lambda)^2 \ , \tag{139}$$

$$\frac{2\eta \beta_y^2}{1-\lambda} \frac{2\beta_x^2}{(1-\lambda)^2} c_{4,3} \le \frac{\beta_y \eta}{12}(1-\lambda)^2 \ .$$

Therefore, we obtain

$$\beta_x \le \min\left\{ \frac{(1-\lambda)^2}{4\sqrt{3c_{\beta_y} c_{4,1}}} \ , \ \frac{(1-\lambda)^{3/2}}{2(3c_{\beta_y} c_{4,2})^{1/4}} \ , \ \frac{(1-\lambda)^{5/3}}{2(6c_{\beta_y} c_{4,3})^{1/3}} \right\} \ . \tag{140}$$

To solve the second inequality in Eq. (138), we obtain

$$\beta_x \le \frac{(1-\lambda)^{5/4}}{12\sqrt{\gamma_2 c_{\beta_y} c_{\hat{\beta}_y}}} \ . \tag{141}$$

To solve the last inequality in Eq. (138), we enforce

$$3\beta_y \eta \frac{4L^2}{K} \frac{\beta_x}{\rho_x \eta} \le \frac{1}{96 \times 4}(1-\lambda)^2 \ , \quad 3\beta_y \eta \frac{10L^2}{K} \frac{\beta_y}{\rho_y \eta} \le \frac{1}{96 \times 4}(1-\lambda)^2 \ ,$$

$$3\beta_y \eta 21L^2 \beta_x \eta \le \frac{1}{96 \times 4}(1-\lambda)^2 \ , \quad 3\beta_y \eta(21L^2 + 9\gamma_2^2)\beta_y \eta \le \frac{1}{96 \times 4}(1-\lambda)^2 \ . \tag{142}$$

We obtain

$$\beta_x \le \left\{ \frac{\sqrt{\rho_x K}(1-\lambda)}{48\sqrt{2c_{\beta_y}}L} \ , \ \frac{\sqrt{\rho_y K}(1-\lambda)}{48c_{\beta_y} L\sqrt{5}} \ , \ \frac{(1-\lambda)}{24L\sqrt{42c_{\beta_y}}} \ , \ \frac{(1-\lambda)}{24c_{\beta_y}\sqrt{2(21L^2+9\gamma_2^2)}} \right\} \ . \tag{143}$$

For $\mathbb{E}[\|\bar{\hat{x}}_{t+1} - \bar{\hat{x}}_t\|^2]$, by setting

$$2\gamma_1 C_{x_{\hat{x}\hat{y}}^1} + 6\gamma_1 \hat{\beta}_x \eta \left( 10 C_{x_{y\hat{x}\hat{y}}^1}^2 C_{y_{\hat{x}\hat{y}}^1}^2 + \frac{4}{\gamma_1 - L} \frac{2\gamma_2^2 C_{y_{\hat{x}\hat{y}}^1}^2}{\mu} \right) + \frac{27\gamma_1^2}{1-\lambda} c_6 - \frac{\gamma_1}{3\hat{\beta}_x \eta} \le -\frac{\gamma_1}{4\hat{\beta}_x \eta} \ . \tag{144}$$

Specifically, we enforce

$$2\gamma_1 C_{x_{\hat{x}\hat{y}}^1} \le \frac{\gamma_1}{36\hat{\beta}_x \eta} \ ,$$

$$6\gamma_1 \hat{\beta}_x \eta \left( 10 C_{x_{y\hat{x}\hat{y}}^1}^2 C_{y_{\hat{x}\hat{y}}^1}^2 + \frac{4}{\gamma_1 - L} \frac{2\gamma_2^2 C_{y_{\hat{x}\hat{y}}^1}^2}{\mu} \right) \le \frac{\gamma_1}{36\hat{\beta}_x \eta} \ ,$$

$$\frac{27\gamma_1^2}{1-\lambda} c_6 \le \frac{\gamma_1}{36\hat{\beta}_x \eta} \ . \tag{145}$$

Since $C_{x_{\hat{x}\hat{y}}^1} = \frac{\gamma_1}{\gamma_1 - L}$, $C_{x_{y\hat{x}\hat{y}}^1} = \frac{\gamma_1}{\gamma_1 - L}$, and $C_{y_{\hat{x}\hat{y}}^1} = \frac{\gamma_1}{\gamma_2 - L}$, we obtain

$$\beta_x \le \min\left\{ \frac{\gamma_1 - L}{72c_{\hat{\beta}_x}\gamma_1} \ , \ \frac{(\gamma_1 - L)(\gamma_2 - L)\sqrt{\mu}}{6\gamma_1 c_{\hat{\beta}_x}\sqrt{6(10\gamma_1^2 \mu + 8\gamma_2^2(\gamma_1 - L))}} \ , \ \frac{1}{18\sqrt{3c_{\hat{\beta}_x}\gamma_1}} \right\} \ . \tag{146}$$

For $\mathbb{E}[\|\bar{\hat{y}}_{t+1} - \bar{\hat{y}}_t\|^2]$, by setting

$$2A_1 + \frac{27\gamma_2^2}{1-\lambda}c_7 - \frac{\gamma_2}{2\hat{\beta}_y\eta} \leq -\frac{\gamma_2}{4\hat{\beta}_y\eta} \tag{147}$$

Specifically, from the definition of $A_1$, we enforce

$$12\gamma_1\hat{\beta}_x\eta C_{y_{\hat{x}\hat{y}}^2}^2 \left(10C_{x_{y\hat{x}\hat{y}}^1}^2 + \frac{8\gamma_2^2}{(\gamma_1 - L)\mu}\right) \leq \frac{\gamma_2}{16\hat{\beta}_y\eta} \, ,$$

$$12\gamma_1\hat{\beta}_x\eta \frac{8\gamma_2^2}{(\gamma_1 - L)\mu} \frac{(1 - \hat{\beta}_y\eta)^2}{\hat{\beta}_y^2\eta^2} \leq \frac{\gamma_2}{16\hat{\beta}_y\eta} \, ,$$

$$\frac{27\gamma_2^2}{1-\lambda}c_7 \leq \frac{\gamma_2}{8\hat{\beta}_y\eta} \, . \tag{148}$$

To solve the second inequality, we use the second inequality in Eq. (69) to obtain the following:

$$\frac{4}{(\gamma_2 - L)^2} \frac{24\gamma_1\hat{\beta}_x\eta}{\gamma_1 - L} \frac{2\gamma_2^2}{\mu} \frac{(1 - \hat{\beta}_y\eta)^2}{\hat{\beta}_y^2\eta^2} \leq \frac{\beta_x\eta}{32 \times 64} \frac{(\gamma_1 - L)^2}{4L^2} \, . \tag{149}$$

Then, it is easy to derive

$$12\gamma_1\hat{\beta}_x\eta \frac{8\gamma_2^2}{(\gamma_1 - L)\mu} \frac{(1 - \hat{\beta}_y\eta)^2}{\hat{\beta}_y^2\eta^2} \leq \frac{\beta_x\eta}{32 \times 64} \frac{(\gamma_1 - L)^2}{4L^2} \frac{(\gamma_2 - L)^2}{2} \tag{150}$$

Therefore, it leads us to solve

$$\frac{\beta_x\eta}{32 \times 64} \frac{(\gamma_1 - L)^2}{4L^2} \frac{(\gamma_2 - L)^2}{2} \leq \frac{\gamma_2}{16\hat{\beta}_y\eta} \tag{151}$$

and we obtain

$$\beta_x \leq \frac{32L}{\sqrt{c_{\hat{\beta}_y}}(\gamma_1 - L)(\gamma_2 - L)} \tag{152}$$

Finally, to solve the first and last inequality in Eq. (148), from $C_{x_{y\hat{x}\hat{y}}^1} = \frac{\gamma_1}{\gamma_1 - L}$ and $C_{y_{\hat{x}\hat{y}}^2} = \frac{\gamma_2}{\gamma_2 - L}$, we obtain

$$\beta_x \leq \min\left\{\frac{\sqrt{\mu}(\gamma_1 - L)(\gamma_2 - L)}{8\sqrt{3c_{\hat{\beta}_x}c_{\hat{\beta}_y}\gamma_1\gamma_2(10\gamma_1^2\mu + 8\gamma_2^2(\gamma_1 - L))}} \, , \frac{1}{6\sqrt{6\gamma_2 c_{\beta_y}c_{\hat{\beta}_y}}}\right\} \tag{153}$$

By setting

$$c_1 = \frac{\beta_x}{\rho_x\eta} \, , \quad c_2 = \frac{5\beta_y}{2\rho_y\eta} \, ,$$

$$c_3 \triangleq \frac{2\beta_x}{(1-\lambda)}c_{3,1} + \frac{2\beta_x^3}{(1-\lambda)^3}c_{3,2} + \frac{2\beta_x^2}{(1-\lambda)^2}c_{3,3} \, ,$$

$$c_4 \triangleq \frac{2\beta_x}{(1-\lambda)}c_{4,1} + \frac{2\beta_x^3}{(1-\lambda)^3}c_{4,2} + \frac{2\beta_x^2}{(1-\lambda)^2}c_{4,3} \, ,$$

$$c_5 = \frac{216\beta_x\gamma_1^2}{(1-\lambda)} \, , \quad c_{10} = \frac{216\beta_y\gamma_2^2}{(1-\lambda)}$$

$$c_6 = \beta_x\eta(1-\lambda) \, , \quad c_7 = \beta_y\eta(1-\lambda) \, , \quad c_8 = 3\beta_x\eta \, , \quad c_9 = 3\beta_y\eta \, , \tag{154}$$

we obtain

$$\mathcal{L}_{t+1} - \mathcal{L}_t \leq -\frac{\beta_x\eta}{4}\mathbb{E}[\|\nabla_x F(\bar{x}_t, \bar{y}_t; \bar{\hat{x}}_t, \bar{\hat{y}}_t)\|^2] - \frac{\beta_y\eta}{2}\mathbb{E}[\|\nabla_y F(\bar{x}_t, \bar{y}_t; \bar{\hat{x}}_t, \bar{\hat{y}}_t)\|^2]$$

$$+ \left( 4\beta_y \eta \beta_x^2 \eta^2 L^2 + 3\beta_x^2 \eta^2 \mathcal{X} - \frac{\beta_x \eta}{4} \right) \mathbb{E}[\|\bar{u}_t\|^2]$$

$$+ \left( \beta_y^2 \eta^2 L_d + \frac{3\beta_y \eta}{4} + \frac{\beta_y^2 \eta^2 (\gamma_2 + L)}{2} + 4A_1 \hat{\beta}_y^2 \eta^2 \beta_y^2 \eta^2 + 2A_2 \beta_y^2 \eta^2 + 3\beta_y^2 \eta^2 \mathcal{Y} - \frac{7}{8} \beta_y \eta \right) \mathbb{E}[\|\bar{v}_t\|^2]$$

$$- \frac{\gamma_1}{4\hat{\beta}_x \eta} \mathbb{E}[\|\bar{\hat{x}}_{t+1} - \bar{\hat{x}}_t\|^2] - \frac{\gamma_2}{4\hat{\beta}_y \eta} \mathbb{E}[\|\bar{\hat{y}}_{t+1} - \bar{\hat{y}}_t\|^2]$$

$$+ 2c_1 \rho_x^2 \eta^4 \sigma^2 \frac{1}{K} + 2c_2 \rho_y^2 \eta^4 \sigma^2 \frac{1}{K} + 3c_6 \rho_x^2 \eta^4 \sigma^2 \frac{1}{1 - \lambda} + 3c_7 \rho_y^2 \eta^4 \sigma^2 \frac{1}{1 - \lambda} + 2c_8 \rho_x^2 \eta^4 \sigma^2 + 2c_9 \rho_y^2 \eta^4 \sigma^2 \ . \tag{155}$$

For $\mathbb{E}[\|\bar{u}_t\|^2]$, we enforce

$$4\beta_y \eta \beta_x^2 \eta^2 L^2 + 3\beta_x^2 \eta^2 \mathcal{X} - \frac{\beta_x \eta}{4} \le -\frac{\beta_x \eta}{8} \ . \tag{156}$$

Specifically, we enforce

$$4\beta_y \eta \beta_x^2 \eta^2 L^2 \le \frac{\beta_x \eta}{16}$$

$$3\beta_x^2 \eta^2 \mathcal{X} \le \frac{\beta_x \eta}{16} \ . \tag{157}$$

To solve the first inequality, we obtain

$$\beta_x \le \frac{1}{8L\sqrt{c_{\beta_y}}} \tag{158}$$

To solve the last inequality, we use the last inequality in Eq. (129) along with the fact that $1 - \lambda < 1$, from which it is straightforward to show that the inequality holds.

For $\mathbb{E}[\|\bar{v}_t\|^2]$, we enforce

$$\beta_y^2 \eta^2 L_d + \frac{3\beta_y \eta}{4} + \frac{\beta_y^2 \eta^2 (\gamma_2 + L)}{2} + 4A_1 \hat{\beta}_y^2 \eta^2 \beta_y^2 \eta^2 + 2A_2 \beta_y^2 \eta^2 + 3\beta_y^2 \eta^2 \mathcal{Y} - \frac{7}{8} \beta_y \eta \le -\frac{1}{32} \beta_y \eta \ . \tag{159}$$

Firstly, from Eq. (106) and the definition of $A_2$, we obtain $2A_2 \beta_y^2 \eta^2 \le \frac{\beta_y \eta}{4 \times 4}$, and

$$2\beta_y^2 \eta^2 A_1 \frac{4\hat{\beta}_y^2}{\beta_y^2 (\gamma_2 - L)^2} \le \frac{\beta_y \eta}{4 \times 4} \tag{160}$$

By reformulating the above inequality, we obtain

$$4A_1 \hat{\beta}_y^2 \eta^2 \beta_y^2 \eta^2 \le \frac{\beta_y^3 \eta^3 (\gamma_2 - L)^2}{4 \times 8} \tag{161}$$

Therefore, we enforce

$$\beta_y^2 \eta^2 L_d + \frac{\beta_y^2 \eta^2 (\gamma_2 + L)}{2} + \frac{\beta_y^3 \eta^3 (\gamma_2 - L)^2}{32} + 3\beta_y^2 \eta^2 \mathcal{Y} \le \frac{1}{32} \beta_y \eta \tag{162}$$

Specifically, we enforce

$$\beta_y^2 \eta^2 L_d + \frac{\beta_y^2 \eta^2 (\gamma_2 + L)}{2} \le \frac{1}{96} \beta_y \eta \ ,$$

$$\frac{\beta_y^3 \eta^3 (\gamma_2 - L)^2}{32} \le \frac{1}{96} \beta_y \eta \ ,$$

$$3\beta_y^2 \eta^2 \mathcal{Y} \le \frac{1}{96} \beta_y \eta \ ,$$

$$\tag{163}$$

To solve the first and second inequality, we obtain

$$\beta_x \leq \min\left\{\frac{1}{48c_{\beta_y}(2L_d + \gamma_2 + L)}, \frac{1}{\sqrt{3}c_{\beta_y}(\gamma_2 - L)}\right\} \tag{164}$$

To solve the last inequality, we use the last inequality in Eq. (138) along with the fact that $1 - \lambda < 1$, from which it is straightforward to show that the inequality holds.

In summary, by setting

$$\beta_x \leq \min\Big\{\frac{(1-\lambda)}{6\sqrt{\gamma_1 c_{\hat{\beta}_x}}}, \frac{(1-\lambda)^2}{4\sqrt{3c_{3,1}}}, \frac{(1-\lambda)^{3/2}}{2(3c_{3,2})^{1/4}}, \frac{(1-\lambda)^{5/3}}{2(6c_{3,3})^{1/3}}, \frac{(1-\lambda)^{5/4}}{12\sqrt{\gamma_1 c_{\hat{\beta}_x}}}, \frac{\sqrt{\rho_x K}(1-\lambda)}{16\sqrt{3}L},$$

$$\frac{\sqrt{\rho_y K}(1-\lambda)}{8L\sqrt{30c_{\beta_y}}}, \frac{(1-\lambda)}{8\sqrt{3(21L^2 + 9\gamma_1^2)}}, \frac{(1-\lambda)}{6\sqrt{\gamma_2 c_{\beta_y} c_{\hat{\beta}_y}}}, \frac{(1-\lambda)^2}{4\sqrt{3c_{\beta_y} c_{4,1}}}, \frac{(1-\lambda)^{3/2}}{2(3c_{\beta_y} c_{4,2})^{1/4}},$$

$$\frac{(1-\lambda)^{5/3}}{2(6c_{\beta_y} c_{4,3})^{1/3}}, \frac{(1-\lambda)^{5/4}}{12\sqrt{\gamma_2 c_{\beta_y} c_{\hat{\beta}_y}}}, \frac{\sqrt{\rho_x K}(1-\lambda)}{48\sqrt{2c_{\beta_y}}L}, \frac{\sqrt{\rho_y K}(1-\lambda)}{48c_{\beta_y}L\sqrt{5}}, \frac{(1-\lambda)}{24L\sqrt{42c_{\beta_y}}},$$

$$\frac{(1-\lambda)}{24c_{\beta_y}\sqrt{2(21L^2 + 9\gamma_2^2)}}, \frac{\gamma_1 - L}{72c_{\hat{\beta}_x}\gamma_1}, \frac{(\gamma_1 - L)(\gamma_2 - L)\sqrt{\mu}}{6\gamma_1 c_{\hat{\beta}_x}\sqrt{6(10\gamma_1^2\mu + 8\gamma_2^2(\gamma_1 - L))}}, \frac{1}{18\sqrt{3c_{\hat{\beta}_x}\gamma_1}},$$

$$\frac{4L}{\sqrt{c_{\hat{\beta}_y}}(\gamma_1 - L)(\gamma_2 - L)}, \frac{\sqrt{\mu}(\gamma_1 - L)(\gamma_2 - L)}{8\sqrt{3c_{\hat{\beta}_x}c_{\hat{\beta}_y}\gamma_1\gamma_2(10\gamma_1^2\mu + 8\gamma_2^2(\gamma_1 - L))}}, \frac{1}{6\sqrt{6\gamma_2 c_{\beta_y} c_{\hat{\beta}_y}}},$$

$$\frac{L^2}{120\gamma_1^3}, \frac{\sqrt{\mu(\gamma_1 - L)^3}(\gamma_2 - L)^2}{512\sqrt{6\gamma_1 c_{\hat{\beta}_x}}\gamma_2 c_{\hat{\beta}_y}}, \frac{1}{8L\sqrt{c_{\beta_y}}}, \frac{1}{48c_{\beta_y}(2L_d + \gamma_2 + L)}, \frac{1}{\sqrt{3}c_{\beta_y}(\gamma_2 - L)}\Big\}$$

$$\eta \leq \min\left\{\frac{1}{\sqrt{\rho_x}}, \frac{1}{\sqrt{\rho_y}}, \frac{1}{\hat{\beta}_x}, \frac{1}{\hat{\beta}_y}, \frac{1}{2\beta_x(\gamma_1 + L)}\right\}, \tag{165}$$

we obtain

$$\mathcal{L}_{t+1} - \mathcal{L}_t \leq -\frac{\beta_x\eta}{4}\mathbb{E}[\|\nabla_x F(\bar{x}_t, \bar{y}_t; \bar{\hat{x}}_t, \bar{\hat{y}}_t)\|^2] - \frac{\beta_x c_{\beta_y}\eta}{2}\mathbb{E}[\|\nabla_y F(\bar{x}_t, \bar{y}_t; \bar{\hat{x}}_t, \bar{\hat{y}}_t)\|^2]$$

$$- \gamma_1 c_{\hat{\beta}_x}\frac{\beta_x\eta}{4}\mathbb{E}[\|\bar{x}_{t+1} - \bar{\hat{x}}_t\|^2] - \gamma_2 c_{\hat{\beta}_y}\frac{\beta_x\eta}{4}\mathbb{E}[\|\bar{y}_{t+1} - \bar{\hat{y}}_t\|^2]$$

$$+ 2\beta_x\rho_x\eta^3\sigma^2\frac{1}{K} + 5\beta_x c_{\beta_y}\rho_y\eta^3\sigma^2\frac{1}{K} + 9\beta_x\rho_x^2\eta^5\sigma^2 + 9c_{\beta_y}\beta_x\rho_y^2\eta^5\sigma^2. \tag{166}$$

Because

$$\|\nabla_x f(\bar{x}_t, \bar{y}_t)\|^2 \leq 2\|\nabla_x F(\bar{x}_t, \bar{y}_t; \bar{\hat{x}}_t, \bar{\hat{y}}_t)\|^2 + 2\gamma_1^2\|\bar{x}_{t+1} - \bar{\hat{x}}_t\|^2,$$

$$\|\nabla_y f(\bar{x}_t, \bar{y}_t)\|^2 \leq 2\|\nabla_y F(\bar{x}_t, \bar{y}_t; \bar{\hat{x}}_t, \bar{\hat{y}}_t)\|^2 + 2\gamma_2^2\|\bar{y}_{t+1} - \bar{\hat{y}}_t\|^2, \tag{167}$$

we obtain

$$\frac{1}{T}\sum_{t=0}^{T-1}\left(\mathbb{E}[\|\nabla_x f(\bar{x}_t, \bar{y}_t)\|^2] + \kappa\mathbb{E}[\|\nabla_y f(\bar{x}_t, \bar{y}_t)\|^2]\right)$$

$$\leq \frac{1}{T}\sum_{t=0}^{T-1}\Big(2\mathbb{E}[\|\nabla_x F(\bar{x}_t, \bar{y}_t; \bar{\hat{x}}_t, \bar{\hat{y}}_t)\|^2] + 2\kappa\mathbb{E}[\|\nabla_y F(\bar{x}_t, \bar{y}_t; \bar{\hat{x}}_t, \bar{\hat{y}}_t)\|^2] + 2\gamma_1^2\mathbb{E}[\|\bar{x}_{t+1} - \bar{\hat{x}}_t\|^2]$$

$$+ 2\kappa\gamma_2^2\mathbb{E}[\|\bar{y}_{t+1} - \bar{\hat{y}}_t\|^2]\Big)$$

$$\leq \max\left\{\frac{8}{\beta_x\eta}, \frac{8\kappa}{\beta_x\eta c_{\beta_y}}, \frac{8\gamma_1}{\beta_x\eta c_{\hat{\beta}_x}}, \frac{8\kappa\gamma_2}{\beta_x\eta c_{\hat{\beta}_y}}\right\}\left(\frac{\mathcal{L}_0 - \mathcal{L}_T}{T} + 2\beta_x\rho_x\eta^3\sigma^2\frac{1}{K} + 5\beta_x c_{\beta_y}\rho_y\eta^3\sigma^2\frac{1}{K}\right.$$

$$\left. + 9\beta_x\rho_x^2\eta^5\sigma^2 + 9c_{\beta_y}\beta_x\rho_y^2\eta^5\sigma^2\right). \tag{168}$$

By setting $\gamma_1 = O(L)$, $\gamma_2 = O(L)$, we obtain

$$c_{\beta_y} = O(1)\,, \quad c_{\hat{\beta}_x} = O\left(\frac{1}{L^2\kappa}\right)\,, \quad c_{\hat{\beta}_y} = O(1)\,. \tag{169}$$

Because

$$\frac{1}{K}\sum_{k=1}^{K}\mathbb{E}[\|\bar{p}_0 - p_0^{(k)}\|^2]$$

$$= \frac{1}{K}\sum_{k=1}^{K}\mathbb{E}[\|\frac{1}{K}\sum_{j=1}^{K}\nabla_x F^{(j)}(x_0, y_0; \hat{x}_0, \hat{y}_0; \xi_0^{(j)}) - \nabla_x F^{(k)}(x_0, y_0; \hat{x}_0, \hat{y}_0; \xi_0^{(k)})\|^2]$$

$$\leq 3\frac{1}{K}\sum_{k=1}^{K}\mathbb{E}[\|\frac{1}{K}\sum_{j=1}^{K}\nabla_x F^{(j)}(x_0, y_0; \hat{x}_0, \hat{y}_0; \xi_0^{(j)}) - \frac{1}{K}\sum_{j=1}^{K}\nabla_x F^{(j)}(x_0, y_0; \hat{x}_0, \hat{y}_0)\|^2]$$

$$+ 3\frac{1}{K}\sum_{k=1}^{K}\mathbb{E}[\|\frac{1}{K}\sum_{j=1}^{K}\nabla_x F^{(j)}(x_0, y_0; \hat{x}_0, \hat{y}_0) - \nabla_x F^{(k)}(x_0, y_0; \hat{x}_0, \hat{y}_0)\|^2]$$

$$+ 3\frac{1}{K}\sum_{k=1}^{K}\mathbb{E}[\|\nabla_x F^{(k)}(x_0, y_0; \hat{x}_0, \hat{y}_0) - \nabla_x F^{(k)}(x_0, y_0; \hat{x}_0, \hat{y}_0; \xi_0^{(k)})\|^2]$$

$$\leq 6\sigma^2 + 6\frac{1}{K}\sum_{k=1}^{K}\mathbb{E}[\|\nabla_x f^{(k)}(x_0, y_0)\|^2]\,, \tag{170}$$

and

$$\frac{1}{K}\sum_{k=1}^{K}\mathbb{E}[\|\bar{q}_0 - q_0^{(k)}\|^2] \leq 6\sigma^2 + 6\frac{1}{K}\sum_{k=1}^{K}\mathbb{E}[\|\nabla_y f^{(k)}(x_0, y_0)\|^2]\,, \tag{171}$$

we have

$$\mathcal{L}_0 = \mathcal{P}_0 + \frac{\beta_x}{\rho_x\eta}\mathbb{E}[\|\frac{1}{K}\sum_{k=1}^{K}u_0^{(k)} - \frac{1}{K}\sum_{k=1}^{K}\nabla_x F^{(k)}(x_0^{(k)}, y_0^{(k)}; \hat{x}_0^{(k)}, \hat{y}_0^{(k)})\|^2]$$

$$+ \frac{5\beta_y}{2\rho_y\eta}\mathbb{E}[\|\frac{1}{K}\sum_{k=1}^{K}v_0^{(k)} - \frac{1}{K}\sum_{k=1}^{K}\nabla_y F^{(k)}(x_0^{(k)}, y_0^{(k)}; \hat{x}_0^{(k)}, \hat{y}_0^{(k)})\|^2]$$

$$+ \beta_x\eta(1-\lambda)\frac{1}{K}\sum_{k=1}^{K}\mathbb{E}[\|\bar{p}_0 - p_0^{(k)}\|^2] + \beta_y\eta(1-\lambda)\frac{1}{K}\sum_{k=1}^{K}\mathbb{E}[\|\bar{q}_0 - q_0^{(k)}\|^2]$$

$$+ 3\beta_x\eta\frac{1}{K}\sum_{k=1}^{K}\mathbb{E}[\|u_0^{(k)} - \nabla_x F^{(k)}(x_0^{(k)}, y_0^{(k)}; \hat{x}_0^{(k)}, \hat{y}_0^{(k)})\|^2]$$

$$+ 3\beta_y\eta\frac{1}{K}\sum_{k=1}^{K}\mathbb{E}[\|v_0^{(k)} - \nabla_y F^{(k)}(x_0^{(k)}, y_0^{(k)}; \hat{x}_0^{(k)}, \hat{y}_0^{(k)})\|^2]$$

$$\leq \mathcal{P}_0 + \frac{\beta_x}{\rho_x\eta}\frac{\sigma^2}{B} + \frac{5\beta_y}{2\rho_y\eta}\frac{\sigma^2}{B} + 9\beta_x\eta\sigma^2 + 9\beta_y\eta\sigma^2 + 6\beta_x\eta\frac{1}{K}\sum_{k=1}^{K}\mathbb{E}[\|\nabla_x f^{(k)}(x_0, y_0)\|^2]$$

$$+ 6\beta_y\eta\frac{1}{K}\sum_{k=1}^{K}\mathbb{E}[\|\nabla_y f^{(k)}(x_0, y_0)\|^2]\,. \tag{172}$$

Then, we have

$$\frac{1}{T}\sum_{t=0}^{T-1}\left(\mathbb{E}[\|\nabla_x f(\bar{x}_t, \bar{y}_t)\|^2] + \kappa\mathbb{E}[\|\nabla_y f(\bar{x}_t, \bar{y}_t)\|^2]\right)$$

$$\leq O\left(\frac{\kappa \mathcal{P}_0}{\beta_x \eta T}\right) + O\left(\frac{\kappa}{T}\frac{1}{K}\sum_{k=1}^{K}\mathbb{E}[\|\nabla_x f^{(k)}(x_0, y_0)\|^2]\right) + O\left(\frac{\kappa}{T}\frac{1}{K}\sum_{k=1}^{K}\mathbb{E}[\|\nabla_y f^{(k)}(x_0, y_0)\|^2]\right)$$

$$+ O\left(\frac{\kappa \sigma^2}{\rho_x \eta^2 TB}\right) + O\left(\frac{\kappa \sigma^2}{\rho_y \eta^2 TB}\right) + O\left(\frac{\kappa \sigma^2}{T}\right) + O\left(\frac{\kappa \rho_x \eta^2 \sigma^2}{K}\right) + O\left(\frac{\kappa \rho_y \eta^2 \sigma^2}{K}\right)$$

$$+ O(\kappa \rho_x^2 \eta^4 \sigma^2) + O(\kappa \rho_y^2 \eta^4 \sigma^2) \,. \tag{173}$$

By setting $\beta_x = O((1-\lambda)^2)$, $\eta = O(\frac{K\epsilon}{\kappa^{1/2}})$, $\rho_x = O(\frac{1}{K})$, $\rho_y = O(\frac{1}{K})$, $B = O(\frac{\kappa^{1/2}}{\epsilon})$, $T = O(\frac{\kappa^{3/2}}{K(1-\lambda)^2\epsilon^3})$, we have

$$\frac{1}{T}\sum_{t=0}^{T-1}\left(\mathbb{E}[\|\nabla_x f(\bar{x}_t, \bar{y}_t)\|^2] + \kappa\mathbb{E}[\|\nabla_y f(\bar{x}_t, \bar{y}_t)\|^2]\right) \leq O(\epsilon^2)\,. \tag{174}$$

