# OpenReview forum: "Doubly Smoothed Decentralized Stochastic Minimax Optimization Algorithm"
_ICLR.cc/2026/Conference — Submitted to ICLR 2026_

### Official Review · Reviewer_5FPy · 2025-10-25

**Soundness:** 2
**Presentation:** 2
**Contribution:** 2
**Rating:** 4
**Confidence:** 3

**Summary:**

This paper addressed the nonconvex-PL min-max optimization problem on a decentralized network. The proposed algorithm incorporates gradient tracking as the decentralized optimization framework, STORM variance reduced gradient estimator to improve the iteration complexity and the smoothing technique to control the progress of outer minimization and inner maximization.

**Strengths:**

- A better dependence on the condition number $\kappa$ is guaranteed.

**Weaknesses:**

### Algorithm Design
- Additional 2 smoothing variables, 2 variance reduction variables and 2 gradient tracking variables incur significant memory overhead.
- Each iteration requires the gossip communication of 6 variables, incurring significant communication overhead.

### Technical Contribution
- The proposed convergence guarantee depends on the notion of $\mathcal{O}(\epsilon_1, \epsilon_2)$-stationarity (Definition 4.1), which deviates from the standard notion of $\mathcal{O}(\epsilon)$-stationarity (also Definition 4.1). In Remark 4.5, the authors suggested applying a centralized algorithm (Yang et al., 2022) to transfer $\mathcal{O}(\epsilon_1, \epsilon_2)$-stationary solution to the standard $\mathcal{O}(\epsilon)$-stationary solution, possibly implying that the proposed algorithm cannot achieve $\mathcal{O}(\epsilon)$-stationary solutions in a decentralized setup.

### Experiment
- The presented numerical experiments are limited to small scale setting. Given the conference's emphasis on neural network training, larger scale experiments such as larger decentralized network, larger dataset and complex models optimization are preferred to demonstrate the practical performance of the proposed algorithm.
- It is not mentioned what kind of classifier is employed in the experiment.


### Writing
 - In line 055, "that" is redundant.

**Questions:**

- Can the authors explain / give intuition to why the proposed decentralized algorithm can achieve $\mathcal{O}(\kappa^{3/2})$ dependence, which is better than the state-of-the-art dependence of $\mathcal{O}(\kappa^2)$ in centralized setting? Can the same $\mathcal{O}(\kappa^{3/2})$ dependence be achieved in the centralized setting, or is it only achievable by decentralized algorithms? Furthermore, do the authors have any clue regarding the optimal dependence to $\kappa$, for instance, if there exists lower bound guarantees to the dependence of $\kappa$?

- Can the authors explain why in Remark 4.5, the optimization problem in line 372 satisfies PL condition in both $x$ and $y$?

- Have the authors considered whether variance reduction on the $y$ variable is necessary or not? Intuitively, the nonconvex minimization over $x$ (at a rate of $\mathcal{O}(1/\sqrt{T})$ [a] or variance-reduced acceleration of $\mathcal{O}(1/T^{2/3})$  (Cutkosky & Orabona, 2019)) should converge slower than the PL stochastic gradient maximization over $y$ (at a rate of $\mathcal{O}(1 / T)$ [b]). Therefore, I suspect that the variance-reduction over $y$ does not contribute to the acceleration of the overall convergence.

[a] Ghadimi, Saeed, and Guanghui Lan. "Stochastic first-and zeroth-order methods for nonconvex stochastic programming." SIAM journal on optimization 23.4 (2013): 2341-2368.

[b] Karimi, Hamed, Julie Nutini, and Mark Schmidt. "Linear convergence of gradient and proximal-gradient methods under the polyak-łojasiewicz condition." Joint European conference on machine learning and knowledge discovery in databases. Cham: Springer International Publishing, 2016.

---

> ### Author Response · Authors · 2025-11-20
>
> **We sincerely appreciate the reviewers’ efforts and insightful comments. Below, we respond to all concerns in detail.**
>
> ---
>
> **W1:** “Algorithm Design: … incur significant memory overhead; … incurring significant communication overhead.”
>
> **Answer:** On the one hand, **the increased memory and communication cost in each round are minimal**. In fact, all competing baselines (DM-HSGD, DREAM, DGDA-VR, DM-GDA) require maintaining 2 momentum variables (variance reduction) and 2 gradient tracking variables, and preform gossip communication on 4 variables. The **only additional elements in our method are the auxiliary variables**, which have the same dimension as the primal and dual variables. Thus, the increased communication cost in each round is minimal and the increased memory overhead is also minimal.  On the other hand, **our method requires fewer communication rounds.** As shown in Table 1,  our communication complexity is better than all existing methods in terms of $\kappa$. Since $\kappa$ is typically a large number, especially for ill-conditioned problems, which are very common for practical machine learning models. This represents a significant advantage. For example, our experimental results show that our method converges faster than competing baselines, indicating that our algorithm requires fewer communication rounds to achieve convergence. In summary, **our method does NOT incur significant communication and memory overhead**.
>
> ---
>
> **W2:** “Theoretical contribution: … cannot achieve O(\epsilon)-stationary solutions in a decentralized setup”
>
> **Answer:** Our analysis does guarantee an $O(\epsilon)$-stationary solutions in a decentralized setup. Extensive studies, such as [1,2, 3, 4], have already shown that the **decentralized minimax algorithms can achieve the same order of convergence rate** as their centralized counterparts. Therefore, following these existing studies, it is straightforward to adapt existing centralized minimax algorithms, such as [5], for the two-sided PL minimax problem to the decentralized setting and use Proposition 2.1 in Yang et al.2022 to complete the translation.  We will cite these papers and provide more discussions in our paper.
>
> [1] Decentralized Stochastic Variance Reduced Extragradient Method.
>
> [2] Near-Optimal Decentralized Algorithms for Saddle Point Problems over Time-Varying Networks.
>
> [3] Optimal Algorithms for Decentralized Stochastic Variational Inequalities.
>
> [4] Decentralized Convex Optimization over Time-Varying Graphs: a Survey.
>
> [5] Global Convergence and Variance Reduction for a Class of Nonconvex-Nonconcave Minimax Problems.
>
> ---
>
> **W3:** “Experiment: The presented numerical experiments are limited to small scale setting…”; “It is not mentioned what kind of classifier is employed in the experiment.”
>
> **Answer:** In response to the reviewer’s suggestion, we provide **an additional experiments on a real-world computer vision task: fair classification on CIFAR10 using ResNet18**. Fair classification is a standard benchmark for reweighting classes to improve worst-class performance and has been widely evaluated in distributed learning algorithms.
>
> In Section 5.2 of the revised version of our paper, we report results under three communication graphs: random, ring, and torus. In **Figure 4**, across all settings, our algorithm achieves the best test accuracy and exhibits more efficient convergence than the competing baselines, further validating its effectiveness on computer vision tasks. For the AUC Maximization, we employ linear classifier to train AUC loss.
>
> ---

---

> > ### Author Response · Authors · 2025-11-20
> >
> > **Q1:** “… why the proposed decentralized algorithm can achieve $O(\kappa^{3/2})$ dependence, which is better than … $O(\kappa^{2})$ in centralized setting? Can the same $O(\kappa^{3/2})$ dependence be achieved in the centralized setting …?”;  “Furthermore, do the authors have any clue regarding the optimal dependence to $\kappa$ … exists lower bound guarantees…? ”
> >
> > **Answer:** The improved $O(\kappa^{3/2})$ dependence comes from the **variance reduction technique**. As discussed in Remark 4.4, since the gradient variance is better controlled by using variance reduction technique, we can use a larger learning rate, while the existing state-of-the-art method (Yang et al., 2022) uses the standard stochastic gradient, which requires a smaller learning rate to mitigate the large gradient variance. Therefore, with a large learning rate, we can get a better dependence on $\kappa$.
> >
> > Our result can be easily extended to the centralized setting. Specifically, when using a complete graph, which is equivalent to the centralized setting and the spectral gap $1-\lambda$ becomes 1, we can conclude that **the centralized setting can achieve the same $O(\kappa^{3/2})$ dependence**.
> >
> > Regarding the optimal dependence on $\kappa$, to the best of our knowledge there is currently no lower bound for this class of decentralized minimax problems. Establishing such lower bounds (and thereby clarifying whether $O(\kappa^{3/2})$ is optimal) remains an interesting direction for future work.
> >
> > ---
> >
> > **Q2:** “… why in Remark 4.5, the optimization problem in line 372 satisfies PL condition in both x and y?”
> >
> > **Answer:** In Remark 4.5, the function $f(x,y)$ refers to the original objective defined in Eq(1). By assumption, it already satisfies the PL condition in dual variable $y$. After adding the quadratic term $L\\|x – x’\\|^2$ to the primal variable, i.e., setting $\gamma_1=2L$ in Eq(2), the resulting objective becomes strongly convex in x, and thus satisfies the PL condition. Therefore, it satisfies PL condition in both x and y.
> >
> > ---
> >
> > **Q3:** “… suspect that the variance-reduction over y does not contribute to the acceleration of the overall convergence.”
> >
> > **Answer:** We agree that the PL condition can **enjoy a better convergence rate for a single variable optimization problem**. However, this intuition **does not fully extend to the minimax setting**, where the primal and dual updates are tightly coupled and both partial gradients needed to be bounded simultaneously. More specifically, in our analysis (Lemma D.9 and D.10), **omitting variance reduction on y** leaves an term of $O(\eta/K)$ in the gradient estimation error, which in turn **leads to a slower $O(1/\sqrt{T})$ rate**. Thus, variance-reduction on y is essential for achieving a convergence rate of $O(1/T^{2/3})$ for minimax problems. This necessity is also consistent with existing variance-reduced minimax algorithms such as DM-HSGD and DM-GDA (which apply STORM to both variables) and DGDA-VR and DREAM (which apply SPIDER to both variables).

---

> > > ### Author Response · Authors · 2025-11-25
> > > **Follow-up**
> > >
> > > Dear Reviewer #5FPy,
> > >
> > > Thank you again for reviewing our paper and reading our rebuttal. If you have any further questions, we are happy to answer them. If our rebuttal has addressed all of your concerns, we would really appreciate it if you could consider raising your score. Thank you so much!
> > >
> > > Best regards,
> > >
> > > Author of 22361

---

### Official Review · Reviewer_z5PY · 2025-11-01

**Soundness:** 3
**Presentation:** 3
**Contribution:** 3
**Rating:** 6
**Confidence:** 3

**Summary:**

This paper proposes \textbf{Smoothed$^2$-DSGDAM}, a doubly smoothed decentralized stochastic gradient descent–ascent with momentum for nonconvex–PL minimax problems.  The algorithm applies the smoothing technique to both primal and dual variables and incorporates variance reduction to reduce gradient variance. It eliminates the need for setting different learning rate scales between primal and dual updates, achieving a unified learning rate and improving condition-number dependence from $\mathcal{O}(\kappa^3)$ to $\mathcal{O}(\kappa^{3/2})$. Theoretical analysis shows convergence rate $\mathcal{O}(1/\epsilon^3)$ and communication complexity $\mathcal{O}(\kappa^{3/2}/((1-\lambda)^2\epsilon^3))$.

**Strengths:**

1. First decentralized minimax optimization algorithm that applies smoothing to both primal and dual variables.

2. Achieves same-order learning rates, facilitating hyperparameter tuning and improving the stability of the training process.

3. Improves dependence on the condition number $\kappa$ and achieves a faster convergence rate compared to previous works.

4. Introduces an effective combination of smoothing and variance reduction tailored for decentralized settings.

5. Provides clear theoretical comparisons with existing algorithms (e.g., DREAM, DGDA-VR, DM-HSGD).

**Weaknesses:**

1. Communication overhead may increase due to auxiliary variable exchange in both primal and dual sides. I hope the author can have more discussions about it.

2. Experiments are somewhat limited. The experiment part includes datasets such as a9a,w8a,and ijcnn1. I hope authors can add more real-world datasets from computer vision areas into the experiments to further validate the advantages of the proposed algorithms.

3.  This paper is restricted to nonconvex–PL settings, which may limit generality to broader minimax classes.

**Questions:**

1. How does auxiliary-variable communication affect scalability with increasing number of workers $K$?

---

> ### Author Response · Authors · 2025-11-20
>
> **We sincerely appreciate the reviewers’ efforts and insightful comments. Below, we respond to all concerns in detail.**
>
> ---
>
> **W1:** “Communication overhead may increase due to auxiliary variable exchange in both primal and dual sides.”
>
> **Answer:** On the one hand, **the increased communication cost in each round is minimal**. The auxiliary variables have the same dimension as the primal and dual variables, meaning that each communication round transmits vectors of the same size as those already required for exchanging model parameters. Thus, the increased communication cost in each round is minimal. On the other hand, **our method requires fewer communication rounds.** As shown in Table 1,  our communication complexity is better than all existing methods in terms of $\kappa$. Since $\kappa$ is typically a large number, especially for ill-conditioned problems, which are very common for practical machine learning models. This represents a significant advantage. For example, our experimental results show that our method converges faster than competing baselines, indicating that our algorithm requires fewer communication rounds to achieve convergence. In summary, the overall communication overhead is modest.
>
> ---
>
> **W2:** “…add more real-world datasets from computer vision areas…”
>
> **Answer:** In response to the reviewer’s suggestion, we provide **an additional experiments on a real-world computer vision task: fair classification on CIFAR10 using ResNet18**. Fair classification is a standard benchmark for reweighting classes to improve worst-class performance in distributed training and has been widely evaluated in federated learning algorithms. However, this more challenging task has not been examined under decentralized minimax optimization in any existing work, not even by the baselines we compare against.
>
> In the revised version, we report results under three communication graphs: random, ring, and torus. In **Figure 4**, across all settings, our algorithm achieves the best test accuracy and exhibits more efficient convergence than the competing baselines, further validating its effectiveness on computer vision tasks.
>
> ---
>
> **W3:** “This paper is restricted to nonconvex–PL settings, which may limit generality to broader minimax classes.”
>
> **Answer:** The nonconvex–PL setting already **represents a broad and meaningful class of minimax problems**, as it includes the nonconvex–strongly-concave case as a special instance. Importantly, recent studies have shown that overparameterized deep neural networks satisfy the PL condition, making the nonconvex–PL framework particularly suitable for analyzing **modern machine learning models**, such as generative adversarial networks, distributionally robust optimization, policy evaluation, and imbalanced data classification. Therefore, studying decentralized minimax optimization under the nonconvex–PL condition provides both theoretical generality and wide practical applicability.
>
> We are also open to extending this doubly smoothed variance-reduced algorithm to other minimax classes, including the nonconvex–nonconcave and nonconvex-concave setting, which is an interesting direction for future research.
>
> ---
>
> **Q1:** “How does auxiliary-variable communication affect scalability with increasing number of workers?”
>
> **Answer:** Our algorithm achieves a convergence rate at $T=O\left(\frac{\kappa^{3/2}}{K(1-\lambda)^{2}\epsilon^3}\right)$, which indicates linear speedup in terms of number of workers $K$. Since the auxiliary variables have the same dimension as the primal ($d_1$) and dual ($d_2$) variables, the overall communication cost $O\left(\frac{(d\_1+d\_2)\kappa^{3/2}}{K(1-\lambda)^{2}\epsilon^3}\right)$.  Therefore, the auxiliary-variable does not hinder the scalability in terms of the order of $d_1$, $d_2$, and $K$. Instead, it improves the dependence on $\kappa$, indicating that the auxiliary-variable benefits the scalability.

---

> ### Author Response · Authors · 2025-11-25
> **Follow-up**
>
> Dear Reviewer #z5PY,
>
> Thank you again for reviewing our paper and reading our rebuttal. If you have any further questions, we are happy to answer them. If our rebuttal has addressed all of your concerns, we would really appreciate it if you could consider raising your score. Thank you so much!
>
> Best regards,
>
> Author of 22361

---

### Official Review · Reviewer_JUDL · 2025-11-01

**Soundness:** 3
**Presentation:** 2
**Contribution:** 2
**Rating:** 4
**Confidence:** 3

**Summary:**

The paper presents a novel decentralized algorithm for stochastic nonconvex-PL saddle-point problems. It improves the best known complexity dependence on the condition number of the loss, and also has a favorable (experimentally observed) property that only one hyperparameter needs tuning.

**Strengths:**

The work makes a significant theoretical advance in this specific research direction by improving complexity by objective's condition number $\kappa$ from $O(\kappa^2)$ to $O(\kappa^{3/2})$.

Authors state that all hyperparameters except one do not require fine-tuning

Structure of the proof is clearly visualized in Figure 4.

**Weaknesses:**

The communication complexity is inferior to DREAM algorithm in terms of $\epsilon$ and the spectral gap.

There is no theoretical comparison for computational complexity with competing algorithms (e.g. in Table 1) only an experimental comparison of gradient‑evaluation counts is provided.

I cannot see theoretically, what is the benefit from having stepsizes of the same scale for primal and dual variables if the algorithm still has many parameters. It was stated in Section 5.2 that "method is robust to all hyperparameters except $\beta$", but the only justification is through a numerical experiment on a single problem instance, which is not a solid evidence.

The analysis is overwhelmingly technical, but this is, unfortunately, standard for the field.

**Questions:**

- Could communication acceleration techniques such as multi-round communication used in DREAM  be incorporated in your algorithm to match the optimal dependence on the spectral gap?
- Why is the nonconvex-PL setting important?

---

> ### Author Response · Authors · 2025-11-20
>
> **We sincerely appreciate the reviewers’ efforts and insightful comments. Below, we respond to all concerns in detail.**
>
> ---
> **W1:** “The communication complexity is inferior to DREAM algorithm in terms of $\epsilon$ and the spectral gap.”
>
> **Answer:** We have discussed this comparison in Table 1. The communication complexity of DREAM achieves a better dependence on the spectral gap, but **this comes at the cost of performing multi-round communication in each iteration**. Specifically, DREAM **relies on a large batch size $O(\frac{\kappa}{\epsilon})$**.
>
> In contrast, our algorithm use only $O(1)$ batch size. If we adopt the same large batch size and multi-round communication as DREAM, our method can also achieve the same communication complexity.
>
> ---
>
> **W2:** “… no theoretical comparison for computational complexity with competing algorithms (e.g. in Table 1) …”
>
> **Answer:** In Table 1, we report both the communication (iteration) complexity, and the batch size used by each algorithm. Since the computation complexity is given by the iteration complexity multiplied by the batch size per iteration, the computation complexity of all methods can be directly obtained from the existing results. We present the results **in the fourth column of Table 1** in the updated version of our paper.
>
> ---
> **W3:** “…theoretically, what is the benefit from having stepsizes of the same scale for primal and dual variables if the algorithm still has many parameters…”; “..a numerical experiment…”
>
> **Answer:** Theoretically, as discussed in Remark 4.4, using primal and dual stepsize of the same scale **brings two key benefits**. First, our algorithm **allows larger stepsize** than the baselines, which leads to a faster convergence rate. Secondly, it **simplifies hyperparameter tuning**: one can set the dual step size equal to the primal stepsize, with one grid search, whereas existing baselines require search over two separate ranges with different magnitudes.
>
> Moreover, in our experiments, we have shown that our algorithm is robust to most hyperparameters.  In addition, beyond the task of auc maximization in Section 5.1, we provide **an additional hyperparameter tuning study on a more challenging image classification task using the CNN model** in Section 5.2 in the updated version of our paper. These new experimental results in **Figure 5** further demonstrate the robustness of our method to most hyperparameters.
>
> ---
>
> **W4:** “The analysis … unfortunately, standard for the field.”
>
> **Answer:** Our theoretical analysis goes **substantially beyond standard arguments** and **introduces several novel results and techniques** that, to the best of our knowledge, have not appeared in prior work.
>
> First, our analysis is built upon a **novel structural framework**, presented in Appendix B. In particular, lemmas characterizing the optimization error (Lemmas C.1–C.7) are novel, as they rely on a doubly smoothed formulation. Determining the dependencies on $\beta_{y}, \hat{\beta}\_{x}, \hat{\beta}\_{y}$ with $\beta_{x}$ in Lemma C.8 is **not trivial and has not been addressed in previous literature**. And these dependencies highlight why our algorithm can use same-scale stepsize, which is not achieved by the existing baselines.
>
> Second, we introduce a **new potential function**, which jointly tracks the **optimization error, gradient estimation error, and consensus error**. Each component requires a rigorous bounding argument. Critically, the choice of $\{c\}\_{1}^{10}$ must satisfy several intertwined constraints. For example, in Eq. (116), bounding the consensus error for $x$,  $\frac{1}{K}\sum\_{k=1}^{K}\mathbb{E}[\| \bar{x}\_{t} - {x}^{(k)}\_{t}\|^2]$ depends simultaneously on the values of $c_5$ and $\hat{\beta}\_{x}$. Such **unique interdependencies involving variance reduction, doubly smoothed, and decentralized consensus error has never been analyzed before**, either in decentralized stochastic minimax optimization or even in the single-machine minimax literature. Similar interdependencies also arise when bounding the consensus errors for other variables, as shown in Eqs. (122), (126), and (135).
>
> Overall, **our proof introduces new techniques, resolves new dependencies, and establishes results that have not been previously studied in any existing work**. These contributions go well beyond the standard analyses typically seen in this area.
>
> ---

---

> > ### Author Response · Authors · 2025-11-20
> >
> > ---
> >
> > **Q1:** “Could communication acceleration techniques such as multi-round communication used in DREAM be incorporated in your algorithm to match the optimal dependence on the spectral gap?”
> >
> > **Answer:** This is an interesting question and indeed a potential direction for future research. Intuitively, incorporating multi-round communication into our doubly smoothed and variance-reduced algorithm could improve the dependence on the spectral gap, and we do not anticipate any challenges in applying DREAM's communication method to our algorithm.
> >
> > ---
> >
> > **Q2:** “Why is the nonconvex-PL setting important?"
> >
> > **Answer:** The nonconvex-PL problem is important because it captures **a broad class of problems**, including the nonconvex–strongly-concave setting and going beyond the standard convex–concave formulation.  Recent studies have shown that overparameterized deep neural networks satisfy the PL condition, making the nonconvex–PL framework particularly suitable for analyzing **modern machine learning models**, such as generative adversarial networks, distributionally robust optimization, policy evaluation, and imbalanced data classification. Therefore, studying decentralized minimax optimization under the nonconvex–PL condition provides both theoretical generality and wide practical applicability.

---

> > > ### Author Response · Authors · 2025-11-25
> > > **Follow-up**
> > >
> > > Dear Reviewer #JUDL,
> > >
> > > Thank you again for reviewing our paper and reading our rebuttal. If you have any further questions, we are happy to answer them. If our rebuttal has addressed all of your concerns, we would really appreciate it if you could consider raising your score. Thank you so much!
> > >
> > > Best regards,
> > >
> > > Author of 22361

---

> > ### Comment · Reviewer_JUDL · 2025-11-25
> >
> > I thank the authors for the response
> >
> > **W1** and **Q1**
> >
> > What are the benefits of using single communication round per iteration and $O(1)$ batch size if this increases total communication and computation theoretical complexities?
> >
> > **W2**
> >
> > It seems strange that the computation complexity of DREAM depends on the spectral gap despite DREAM applies multiround communications
> >
> > **W3**
> >
> > Thank you for doing this work! Although I am still sceptical about this evidence since it is still a single problem instance and a very dense graph.
> >
> > **W4**
> >
> > Thank you for clarifying technical difficulties behind your proofs! I believe this information is quite valuable
> >
> > **Q2**
> >
> > Thank you, my concern is addressed

---

> ### Author Response · Authors · 2025-11-26
>
> Thanks for the reviewer’s follow-up questions. We address them in detail below.
>
> ----
> **W1 and Q1:** What are the benefits of using single communication round per iteration and batch size $O(1)$ if this increases total communication and computation theoretical complexities?
>
> **Answer:**  The key point is that using single-round communication  per iteration and batch size $O(1)$ leads to much lower per-iteration cost. In addition, single-round communication is more robust to the communication latency when the communication is not stable, as the multi-round communication could suffer from large latency in every round.
>
> **Theoretical computation efficiency**:  Large-batch methods (e.g., DGDA-VR and DREAM) incur significantly higher per-iteration computation costs because each iteration requires computing large batches $O(\kappa/\epsilon)$. As shown in Table 1, for methods with a large batch size (DGDA-VR, DREAM), the computation complexity is higher than the communication complexity, whereas for methods using batch size $O(1)$, the computation and communication complexities are of the same order. Notably, the large batch-size $O(\kappa/\epsilon)$ is NOT practical when the desired solution accuracy $\epsilon$ is small. For example, when $\epsilon=10^{-7}$, we should use at least $10^{7}$ training samples in every iteration. Considering the ImageNet dataset, such a large batch size would require using almost all images in the dataset in every iteration, which is clearly impractical.
>
> **Empirical gradient-evaluation efficiency**: Across both numerical experiments and deep neural networks, our method consistently requires far fewer gradient evaluations compared to large-batch baselines. For instance, in Figure 4, DGDA-VR and DREAM typically require around $40×10^3$ gradient evaluations to converge, whereas our method converges within only about $10×10^3$ evaluations, roughly 25% of the baseline cost. Similarly, in Figures 1 and 2, our method achieves convergence using approximately 40% of the gradient evaluations required by the baslines. These practical benefits further demonstrate the advantage of using an $O(1)$ batch size.
>
> ---
>
> **W2:** It seems strange that the computation complexity of DREAM depends on the spectral gap despite DREAM applies multiround communications.
>
> **Answer:** We follow the theoretical result given in Theorem 3.1 of (Chen et al., 2022). In DREAM, the communication complexity explicitly depends on the spectral gap of the network. Since the computation complexity is obtained by multiplying this communication complexity by the batch size used by DREAM, the dependence on the spectral gap naturally appears in the computation complexity as well.
>
> In fact, the dependence on the spectral gap is inherent in the decentralized algorithms. While multi-round communication in DREAM improve the spectral gap factor from $(1 - \lambda)^2$ to $\sqrt{1-\lambda}$, they do not remove this dependence.
> Only for some special graphs, such as [1], the multi-round communication can guarantee the exact average, which can then remove the  dependence on the spectral gap.
>
> [1] https://arxiv.org/abs/2305.11420
>
> ---
>
> **W3:** Although I am still sceptical about this evidence since it is still a single problem instance and a very dense graph.
>
> **Answer:** We believe the presented results are already sufficiently diverse to support the claim: the experiments cover multiple tasks (auc maximization and fair classication), multiple datasets, and different neural architectures. Therefore, this is not a single-problem instance. Instead, the performance of our algorithm is confirmed by multiple instances, including instances in both the original submission and the revised version.
>
> Moreover, we would like to clarify that our aim is to develop a novel algorithm and provide a provable convergence analysis.
> All empirical comparison and ablation studies are sufficient to confirm the effectiveness of our novel method.
>
>
> ---
>
> We hope that our detailed responses have addressed all remaining concerns. If there are further questions, we would be happy to provide additional clarification. We would also be grateful if the reviewer could consider raising the score. Thank you so much!

---

> > ### Comment · Reviewer_JUDL · 2025-11-26
> >
> > Unfortunately, some of my follow-up questions were not addressed
> >
> > **W1** and **Q1**
> >
> > > For example, when $\epsilon=10^{-7}$, we should use at least $10^{7}$ training samples in every iteration. Considering the ImageNet dataset, such a large batch size would require using almost all images in the dataset in every iteration, which is clearly impractical.
> >
> > Applying this logic to your algorithm which has computation complexity of $O(\kappa^2 / \epsilon^4)$, it would require to use $\sim 10^{28}$ computations when $\epsilon = 10^{-7}$ which is even more impractical
> >
> > **W2**
> >
> > Multiconsensus efficiently reduce spectral gap to $O(1)$ for the whole algorithm except the communication part. E.g., computation complexity of algotirhm proposed in [1] does not depend on network properties, while communication complexity, of course, does, but in an accelerated way (the spectral gap is reduced to the square root of the spectral gap). Similarly, I as far as see, in Theorem 3.1 in the DREAM paper shows no computation (SFO) complexity dependence on spectral gap $\delta$.
> >
> > **W3**
> >
> > I recognize additional experimental validation, but my opinion is that stating that "method is robust to all hyperparameters except $\beta$" is not scientifically honest, since the only evidence is experimants where hyperparameters were changed one-by-one on two problem instances (AUC maximization, a9a and Fair classification, CIFAR-10) both tested on dense random graphs with large spectral gap.
> >
> > [1]  Scaman, Kevin, et al. "Optimal algorithms for smooth and strongly convex distributed optimization in networks." international conference on machine learning. PMLR, 2017.

---

> ### Author Response · Authors · 2025-11-26
>
> Thanks for the reviewer's additional questions. We address them below.
>
> ----
>
> **W1 and Q1**
>
> **Answer**: We respectfully disagree with the reviewer's computation complexity, as **the reviewer has some $\textcolor{red}{\text{misunderstanding}}$** and made some $\textcolor{red}{\text{mistakes}}$ about the **per-iteration** computation complexity, the **total** computation complexity, and our **Table 1**.  The details are shown below.
>
> * ($\textcolor{red}{\text{Misunderstanding}}$) **Per-iteration v.s. Total computation complexity:** In our previous response, we have clearly pointed out that the **per-iteration** computation complexity of DREAM is $10^{7}$ when $\epsilon=10^{-7}$. However, in the reviewer's follow-up comments, **the reviewer compares the per-iteration computation complexity with the total computation complexity**.  That said, **$10^{28}$ is the total computation complexity, NOT the per-iteration complexity**.  This comparison is **unfair**.
>
>
> * ($\textcolor{red}{\text{Mistake}}$) **$10^{28}$ is the baseline's computation complexity, NOT ours.** As shown in Table 1, $O(\kappa^2/\epsilon^{\color{red}{4}})$ is the computation complexity of Smoothed-SAGDA (Yang et al., 2022), **NOT ours**. In fact, according to the dependence on $\epsilon$, both our method and DREAM have the same order, i.e., $O(1/\epsilon^{\color{red}{3}})$. This indicates that our method and DREAM have the same **total** computation complexity (i.e., $10^{3\times 7}$), while our method has a much smaller **per-iteration** complexity than DREAM (i.e., $O(1)$ v.s. $10^{7}$).
>
> * Finally, our experiments further support this observation: large-batch methods require substantially more gradient evaluations to converge, consistent with the theoretical analysis.
>
> ----
>
> **W2**
>
> **Answer:** Thank you for pointing out the computation complexity in DREAM. After further checking it, we agree that its computation complexity does not rely on $\delta$, as it uses the multi-round communication. We have revised this in Table 1 in the  new version of the paper.
>
> However, we would like to emphasize that our method can also use DREAM's multi-round communication. In fact, its communication method has been well-studied and applied to different algorithms. Just as the reviewer mentioned, it is well recognized that multiconsensus can improve the spectral gap dependence. **Directly using multiconsensus to our method does NOT help too much for the development of decentralized optimization**, as **it does NOT address the unique challenges in decentralized minimax optimzation** from our perspective.
>
> Therefore, we don't want to spend our efforts to this straightforward extension. Instead,  we focus on the more challenging goals in our paper: **improving the dependence on the condition number and better coordinating the update of two variables.** These two have not been explored for decentralized minimax optimization and much more challenging than using the multi-round communication. Our paper has successfully addressed these challenges and achieved better theoretical guarantees and empirical performance. Therefore, our contribution is significant for decentralized minimax optimization and should not be diminished merely due to the use of a basic communication method.
>
> Finally, we agree that multiconsensus is an important and effective method for improving the spectral gap dependence. We appreciate the reviewer for sharing that paper with us. We will definitely cite that paper and provide discussions about multiconsensus communication.
>
> ---
>
> **W3**
>
> **Answer**: In response to the reviewer’s concern, we **add an ablation study under the line graph topology** in the appendix, which corresponds to a **small spectral gap** (approximately $1/K^2$). As shown in the revised paper (Page 13, **Figure 6**), varying the hyperparameter values under a sparce graph still does not significantly affect performance, further confirming the robustness of our method.
>
> ----
>
> We appreciate the reviewer’s feedback. Please let us know if any concerns remain and we are glad to clarify further.

---

### Official Review · Reviewer_pTZp · 2025-11-07

**Soundness:** 3
**Presentation:** 3
**Contribution:** 3
**Rating:** 6
**Confidence:** 2

**Summary:**

The motivation of this paper is to develop a decentralized smoothed minimax optimization method that achieves a better convergence rate while using same-scale learning rates for the primal and dual variables. This paper achieves this goal by introducing their method, Double Smoothed Decentralized Stochastic Gradient Descent Ascent (Smoothed$^2$-DSGDAM), which applies a smoothing technique to both the primal and dual variables.
While such double smoothing is challenging in incorporating the variance reduction technique and handling communication complexity, this paper overcomes these issues and improves the convergence complexity to $O(\kappa^{3/2})$. Finally, they present experimental results showing that their algorithm outperforms previous algorithms.

**Strengths:**

- This paper provides an algorithm based on a new design technique (double smoothing), which addresses the goal of developing a decentralized smoothed minimax optimization algorithm that uses same-scale learning rates for the primal and dual variables.

- Its novelty is not only based on its idea but also justified by its technical difficulty. This smoothing technique faces two challenges. The algorithm is carefully designed to apply the variance reduction technique and to bound the consensus error regarding their auxiliary variable, thereby achieving a faster convergence rate. Their high-level idea is introduced on page 6.

- The improvement of convergence complexity has its own merit as a contribution.

- The overall story of the paper is presented clearly, starting from the motivation to the novel aspects of the algorithm’s design, the challenges they faced, and how they overcame them.

- This reviewer personally finds the well-structured appendix very impressive. It seems that the authors put a lot of effort into helping readers understand the structure of the proofs and the appendix.

- Not only for the idea of overcoming challenges, this reviewer thinks that the fact they went through the heavy calculations required for the proof is worthy of applause.

**Weaknesses:**

While this reviewer didn’t find any significant weaknesses in the paper, this reviewer felt there are a few points that could be improved in terms of formatting.

- In line 219, instead of writing $>0$ repeatedly for every parameter like $a>0, b>0, c>0$, writing $a, b, c>0$ seems better. The same applies for $<1$.

- In lines 230–233 (update rule of $u$ and $v$), this reviewer thinks it would be better to have a line break.

- In the appendix, there are too many unused equation numbers. Since an equation number provides information about which equations will be used again and may be important, not only are the extra numbers distracting, but this reviewer also thinks they may make it harder to identify the important equations.

- The formal introduction of the algorithm name is on page 4, after going through motivation and challenges. But this reviewer personally thinks it may be better to introduce it earlier in the introduction.
This paper puts a lot of emphasis on the challenges they needed to overcome, using bold text several times, but seems to place less emphasis on how they actually overcame them. For example, to the best of this reviewer's understanding, lines 303–321 seem to detail how they overcame the challenges. This reviewer wonders whether this part could be more structured, indicating which section corresponds to which challenge, and highlighting the key ideas.

**Questions:**

- **Q1.** Could the authors provide a high-level explanation of the core ideas they used to overcome the challenges, and share their insights on why those ideas work?

- **Q2.** The potential function in equation (95) seems quite complicated. Could the authors provide the motivation for how they came up with such a potential function?

---

> ### Author Response · Authors · 2025-11-20
>
> **We sincerely appreciate the reviewers’ efforts and insightful comments. Below, we respond to all concerns in detail.**
>
> ---
>
> **W1, W2:** “… line 219, … writing…”; “… line 230-233, … line break…”
>
> **Answer:** We have updated the algorithm’s formatting as suggested to improve readability.
>
> ---
> **W3:** “… appendix, … too many unused equation numbers”
>
> **Answer:** Thank you for the suggestion. We included equation numbers throughout the appendix to help reviewers easily reference any equation during the review and rebuttal process. We will refine the numbering in the camera-ready version, if accepted, to remove unnecessary labels and highlight only the essential equations.
>
> ---
> **W4:** “… The formal introduction of the algorithm name is on page 4 …, better to introduce it earlier in the introduction”; “… less emphasis on how they actually overcame them …, … how they overcame the challenges, … more structured”
>
> **Answer:** We have formally introduced our algorithm earlier in the **Introduction** (page 3, line 110). The key ideas for **addressing the challenges are summarized** afterward in lines 111–131, where we highlight our contributions to how to overcome the challenges from two perspectives: algorithmic design and theoretical analysis.
>
> The subsequent sections then provide the detailed development of these ideas. In the **Method** section, lines 267–269 present the specific challenges in algorithmic design, and lines 270–321 elaborate on how we address them step by step. For the theoretical challenges, although space limitations prevent a full presentation in the main text, the **Appendix** includes a structured proof outline that explains in detail how each difficulty is resolved.
>
> We hope this clarifies the intended structure: **the Introduction offers a high-level summary, and the main sections and Appendix provide the corresponding detailed resolutions.**
>
> ---
>
> **Q1:** “… high-level explanation of the core ideas they used to overcome the challenges, and share their insights on why those ideas work …”
>
> **Answer:** The main challenges arise in two aspects of algorithmic design as shown in lines 267–269: (1) how to incorporate variance reduction technique into smoothing technique, and (2) how to compute and communicate the auxiliary variable efficiently.
>
> To address (1), we analyze two possible choices: applying variance reduction to the original gradients or to the smoothed gradients. As discussed in lines 270–303, applying variance reduction directly to the original gradients significantly complicates the analysis and makes hyperparameter selection more difficult. Therefore, we apply variance reduction to the smoothed gradients, which leads to a more stable algorithmic structure and a cleaner proof. Here, **the key takeaway is that variance reduction should be applied to the entire gradient** rather than to its subcomponents.
>
> To address (2), we also consider two options: communicating or not communicating the auxiliary variables. Lines 304-321 show that if the auxiliary variables are not communicated, the resulting consensus error becomes difficult to control and may even diverge. Therefore, we propose communicating the auxiliary variables to avoid this issue and ensure stability.  Here, **the key takeaway is that auxiliary variables should be communicated to guarantee convergence**, as they can otherwise introduce inconsistencies across different workers.
>
> ---
> **Q2:** “ … The potential function … seems quite complicated. … provide the motivation for how they came up with such a potential function?”
>
> **Answer:** An explanation of this construction, **including high-level idea and motivation,** has already been included in Appendix B. In particular, in the equation of Page 14,  the potential function $\mathcal{L}_t$ involves three types of errors: optimization error, consensus error, and gradient estimation error. Each is bounded through lemmas in Appendix C and D. Since these errors share a common recursive structure, we combine them into a unified potential function to track their evolution throughout the iteration.

---

> > ### Author Response · Authors · 2025-11-25
> > **Follow-up**
> >
> > Dear Reviewer #pTZp,
> >
> > Thank you again for reviewing our paper and reading our rebuttal. If you have any further questions, we are happy to answer them. If our rebuttal has addressed all of your concerns, we would really appreciate it if you could consider raising your score. Thank you so much!
> >
> > Best regards,
> >
> > Author of 22361

---

### Author Response · Authors · 2025-11-30
**Summary of Rebuttal**

Dear AC,

We thank you and all reviewers for your efforts in reviewing our paper and for providing insightful comments. In this message, we would like to summarize the initial reviews, our responses, and the follow-up discussions.

---
 **Reviewer pTZp**

(+) This reviewer acknowledges the **novelty** of our paper and our efforts in **clearly presenting the motivation and solutions**.

(-) This reviewer raised only **minor** concerns about **certain details of our paper**, which we have thoroughly clarified in our rebuttal.

Therefore, it should be evident that we have **fully resolved the reviewer’s concerns**.

---
**Reviewer JUDL**

(+) This reviewer acknowledges the **significant theoretical advance** in improving the complexity and our efforts in **clearly presenting the proof structure**.

(-) This reviewer's concern is also **minor** and mainly lies in the **comparison with the theoretical computational complexity** and asking for **additional experiments regarding hyperparameters** even though we already have such experiments in the initial submission.
For the first concern, we updated Table 1 by providing all methods' computational complexity.
For the second one, we have conducted two additional experiments (see Figure 5 and 6) to further demonstrate how the hyperparameter affects convergence.

In the follow-up discussion, this reviewer raised additional questions about the baseline method DREAM, which uses multi-round communication. To address these questions, we provided an example to better illustrate the advantages of our method. However, **the reviewer’s response contained misunderstandings and inaccuracies**. Then, we clearly pointed out these errors, but the reviewer did not respond further.

In summary, **it is clear that these concerns have been fully addressed.**

---
**Reviewer z5PY**

(+) This reviewer recognizes the **novelty of our algorithm design** and the **improvements in convergence rate and hyperparameter selection**.

(-) This reviewer's concerns are **minor**, which includes the **clarification of some details** and the **verification of our algorithm on image dataset**. We provide detailed explanation and add a new experiment in Section 5.2 to address these concerns.

Clearly, **these concerns have been fully addressed.**

---

**Reviewer 5FPy**

(+) This reviewer acknowledges **the improvement in our convergence rate, particularly in its dependence on the condition number**.

(-) This reviewer's concerns are also **minor**. In particular, he/she asked for the **clarification regarding certain details** of our paper and the **performance of our algorithm on complicated applications**. To address these concerns, we provided more detailed explanation and conducted an additional experiment in Section 5.2.


Obviously, **all concerns have been fully addressed.**


---
**Summary**

In summary, all reviewers acknowledge the **novelty of our algorithm** and the **theoretical contributions** of our paper. The concerns raised mainly involve **clarifying certain details** and **verifying our algorithm on additional applications**, both of which have been **fully addressed** in our rebuttal.

Thank you again for taking the time to read this message. We sincerely hope the AC will take our responses into account when making the final decision.

Best regards,

Author of 22361

---

### Meta-Review · Area_Chair_d5Sk · 2025-12-24

**Summary:**

This paper considers the NC-PL minimax optimization problem. In particular, instead of the (local) KL condition that generally holds for tame functions, a strong global PL assumption is made for y variable. The paper considers the distributed setting of this problem and establishes an improved dependence on condition number.

One criticism on the paper is whether the global PL assumption is too restricted. Several reviewers pointed this out. The authors justified by mentioning NC-SC problems and the PL condition for neural nets. However, strong-convexity is significantly stronger than global PL, and this can be observed from the known lower bound results (known in minimization, unknown for min-max). In addition, neural nets only satisfy (local) KL condition, not global PL, that is, the authors' justification is not correct. The AC does not think the authors well justify this concern.

Another criticism on the paper is about the (\epsilon,\epsilon/\sqrt{\kappa})-solution and the \epsilon-solution. The authors do provide some justification. However, the AC thinks that the provided justification is hand waving and lacks sufficient rigor. In addition, (Yang et al. 2022) considers NC-SC, while this paper is NC-PL. Additional justification is needed.

The AC thinks the paper is below the acceptance threshold and would like to decide a reject.

**Reviewer Concerns:**

Addressed:
(1) pTZp: Several presentation issues.
(2) JUDL: Lacking motivation to use the same step size.
(3) JUDL: The communication complexity is inferior to DREAM algorithm.

Not Addressed:
(1) JUDL & z5PY: PL is too restrictive and lacks sufficient importance

Let us explain why the AC thinks the justification is for this is bad. (1). The authors say the PL contains a wide class of functions. This is wrong. Only KL is widely satisfied, and it is a local property. The paper considers globally defined PL condition, it is very narrow. (2) The authors say NN satisfies PL. Unfortunately, this is still wrong. NN only satisfies the (local) KL condition, it does not satisfy the global PL condition. (3) The authors use NC-SC as an example. Yes, SC implies PL. However, SC is much stronger, and it allows acceleration, but PL does not. Then the questions is: "why not consider SC?" Overall, the AC thinks this is a fatal issue of the paper: it considers a too narrow problem setting (PL but not SC).

**Reviewer Scores:**

The AC thinks the authors' justification is not sufficient to change the reviewer JUDL's mind. Therefore, the paper's overall rating is below the acceptance threshold of the conference.

---

### Decision · Program_Chairs · 2026-01-26

Reject